

# The compact Earth system model OSCAR v2.2: description and first results

Thomas Gasser[1,2], Philippe Ciais[1], Olivier Boucher[3], Yann Quilcaille[1,2], Maxime Tortora[2], Laurent Bopp[1], and Didier Hauglustaine[1]

[1]Laboratoire des Sciences du Climat et de l'Environnement, LSCE/IPSL, Université Paris-Saclay, CEA – CNRS – UVSQ, 91191 Gif-sur-Yvette, France
[2]Centre International de Recherche sur l'Environnement et le Développement (CIRED), CNRS – PontsParisTech – EHESS – AgroParisTech – CIRAD, 94736 Nogent-sur-Marne, France
[3]Laboratoire de Météorologie Dynamique, LMD/IPSL, CNRS – UPMC, 75252 Paris, France

*Correspondence to:* Thomas Gasser (tgasser@lsce.ipsl.fr)

**Abstract.** This paper provides a comprehensive description of OSCAR v2.2, a simple Earth system model. The general philosophy of development is first explained, it is then followed by a complete description of the model's drivers and various modules. All components of the Earth system necessary to simulate future climate change are represented in the model: the oceanic and terrestrial carbon-cycles – including a book-keeping module to endogenously estimate land-use change emissions – so as to simulate the change in atmospheric carbon dioxide; the tropospheric OH chemistry and the natural wetlands, to simulate that of methane; the stratospheric chemistry, for nitrous oxide; thirty-seven halogenated compounds; changing tropospheric and stratospheric ozone; the direct and indirect effects of aerosols; changes in surface albedo caused by black carbon deposition on snow and land-cover change; and the global and regional response of climate – in terms of temperatures and precipitations – to all these climate forcers. Following the probabilistic framework of the model, an ensemble of simulations is made over the historical period (1750–2010). We show that the model performs well in reproducing observed past changes in the Earth system such as increased atmospheric concentration of greenhouse gases or increased global mean surface temperature.

## 1 Introduction

Simple biogeochemistry-climate models, also qualified as compact or reduced-form models, are widely used in the climate change research community. These models share several features. First, they are not spatially resolved and as such they can be refered to as box models, although the number of boxes – and therefore of state variables – may vary greatly: from a couple to several hundreds. This limited number of state variables make them relatively intelligible, when compared to complex three-dimensional models. Second, the time-step of analysis and of numerical solving is about one year, which implies they usually cannot include representations of seasonal processes. One consequence of these two features is their very high computing efficiency: one simulation typically takes about one minute on a laptop. Compact models can therefore be used in a variety of setups, for instance: to translate a large number of pathways of greenhouse gases emissions into projected climate change (e.g. Clarke et al., 2014); to complement a study by a process-based model (e.g. Schneider von Deimling et al., 2012) or an





economic model (e.g. Rogelj et al., 2013); to extend a given experiment or assess its uncertainty with a Monte-Carlo analysis (e.g. Gasser et al., 2015); to attribute changes in a variable of the climate system to physical processes (e.g. Raupach et al., 2014) or to emitting countries (e.g. Höhne et al., 2011); or to easily discuss theoretical frameworks (e.g. Raupach, 2013) or policy-relevant indicators (e.g. Huntingford et al., 2012). Because they are simple models of complex phenomena, these models

can hardly be process-based. Quite the opposite, they usually consist of *ad hoc* parametric laws which require to be calibrated or optimized using either observations (e.g. Ricciuto et al., 2008) or more complex models (e.g. Meinshausen et al., 2011).

Here, we present an important update of a simple Earth system model that has already been used for some time. The model is named OSCAR, and this paper provides a comprehensive description of version 2.2. This model has a relatively large number of parameters which are almost all calibrated on complex models. Section 2 provides the details of the mathematical formulation

of the model, and it describes how the parameters are calibrated or derived. The first subsection provides general information about the model. The second subsection describes the forcings of the model: anthropogenic emissions of greenhouse gases, ozone precursors, aerosols and aerosols precursors, land-use and land-cover change, and some additional anthropogenic and natural radiative forcings. The next subsections describe the model's various modules dedicated to the prediction of carbon dioxide, methane, nitrous oxide, halogenated compounds, ozone, aerosols, surface albedo, and the climate reponse. The last

subsection describes the numerical solving method. Section 3 then provides first results from OSCAR v2.2 in the case of a simulation over the historical period from 1750 to 2010, as well as a brief discussion of these results. Section 4 provides some preliminary conclusions regarding the model, its performance and interest, and future potential development.

## 2 Description

### 2.1 General points

Since version 2.0 (see appendix A for an overview of the model's history), the development of OSCAR is driven by three principles. First, we aim to embed in OSCAR as many components and processes of the actual Earth system as possible, with the overall idea of favoring the amount of processes, interactions and feedbacks implemented over the modelling complexity of each of these elements. Second, OSCAR is built as a meta-model capable of emulating the sensitivities of models of higher resolution or superior complexity. To do so, in order to calibrate those sensitivities, we use outputs from complex models that

participated to intercomparison projects whenever possible. Consequently, OSCAR is designed to be used in a probabilistic framework. Third, the model is developed as a dynamic model of the Earth system with an assumed equilibrium corresponding to the preindustrial era. The reason for this is the original purpose of the model to perform studies attributing the anthropogenic component of climate change (e.g. Ciais et al., 2013a; Gasser, 2014; Li et al., 2016).

This last point also is the reason why all the following equations are expressed as a difference to our so-called preindustrial

equilibrium. In the model, all variables $Z$ are therefore formulated as a constant term $Z_0$, being the preindustrial value of the variable, to which a time-varying perturbation term $\Delta Z$ is added, so that we always have: $Z = Z_0 + \Delta Z$. Only the forcings, described in the following subsection, are expressed without this $\Delta$-term, since per construction their constant term is zero. This $\Delta$-term is the actual state variable of the model, and it is consequently the one used to discuss the performance of OSCAR





in section 3. Also, all these state variables are nil at the beginning of the simulation (i.e. at $t = 0$). Because of this modelling approach, one may better describe OSCAR as being a "climate *change*" or "Earth system *change*" model. A diagram of the model's simplified structure is shown in figure 1.

Let us now introduce some of the main notations that are used throughout the description sections. The variables of the model are written with Roman letters, whereas its parameters are with Greek letters. Among the variables, some letters are consistently dedicated to a specific kind of variable: $F$ for fluxes of matter; $E$ for emissions, i.e. positive fluxes towards the atmosphere; $C$ for carbon pools; $A$ for surface areas; $T$ for temperature; $P$ for precipitations. Similarly, among the parameters: $\alpha$ for proportionality factors, and therefore also conversion factors; $\gamma$ for relative sensitivities to a climate variable; $\Gamma$ for absolute sensitivities to a climate variable; $\xi$ for chemical sensitivities; $\tau$ for time constants; $\nu$ for rate constants, i.e. the inverse of time constants; $\kappa$ for dimensionless constants; $\pi$ for fractions, i.e. dimensionless constants within $[0,1]$; $\omega$ for weights, i.e. dimensionless constants whose reference value is 1. A few variables or parameters, however, do not follow these general rules. Additionally, we use the notation $\mathcal{F}$ for mathematical functions whose arguments are given in square brackets. Superscripts are used either to refer to a given atmospheric species or to denote the subdivision of a given variable or parameter along a given axis (e.g. a regional axis). The time variable $t$ is kept implicit for legibility, unless it is required.

## 2.2 Forcings

### 2.2.1 Anthropogenic emissions

Anthropogenic emissions of various active gases are the main drivers of climate change, and thus of OSCAR. In this version of the model, the anthropogenically emitted species directly prescribed as emissions over the 1750–2010 period are, for greenhouse gases: carbon dioxide from fossil-fuel burning and cement production ($E_{FF}$); methane ($E_{CH4}$); nitrous oxide ($E_{N2O}$); many halogenated compounds ($E_X$; see list in section 2.7); for ozone precursors: nitrogen oxides ($E_{NOx}$); carbon monoxide ($E_{CO}$); non-methane volatile organic compounds ($E_{VOC}$); for aerosols and aerosol precursors: sulfur dioxide ($E_{SO2}$); ammonia ($E_{NH3}$); organic carbon ($E_{OC}$); black carbon ($E_{BC}$).

Because most of these anthropogenic drivers are actually poorly known, especially when going backward in time, we use various established inventories in order to introduce variation in our model's results, and thus explore the uncertainty in climate change reconstruction and projection. The historical emissions of fossil-based $CO_2$ can be based on the CDIAC dataset (Boden et al., 2013) or on EDGAR v4.2 (EC-JRC/PBL, 2011). Those of $CH_4$ can be based on EDGAR, ACCMIP (Lamarque et al., 2010) or EPA (2012). Those of $N_2O$ can be based on EDGAR or EPA. Those of all hydrofluorocarbons (HFCs), perfluorocarbons (PFCs), sulfur hexafluoride ($SF_6$) and nitrogen trifluoride ($NF_3$) are based on EDGAR, while those of ozone depleting substances (ODSs) are taken from Meinshausen et al. (2011). The emissions of $NO_x$, CO, VOCs, $SO_2$ and $NH_3$ can be based on EDGAR or ACCMIP – this being set independently for each species. Those of OC and BC are based on ACCMIP. Note that the biomass burning emissions are removed from these datasets, since those emissions are endogenous to the OSCAR model (see section 2.4.1), but all other sectors provided by the inventories are accounted for, included notably agricultural waste burning.





Similarly to what is done by Li et al. (2016, supplementary information section B.1), the time-series of anthropogenic emissions are constructed as follows. First, we choose one of the datasets above as the reference dataset whose absolute values are taken as they are over its period of definition. Second, if needed, we extend the reference dataset beyond its period of definition by following the relative year-to-year variation of other datasets. The extension forward is needed in two cases: with

the EDGAR dataset as reference, in which case the extension covers 2008–2010 and is made with the EDGAR-FT v4.2 dataset (EC-JRC/PBL, 2013) for greenhouse gases and the EDGAR-HTAP v2 dataset (EC-JRC/PBL, 2014) for other species; and with the ACCMIP dataset, in which case the extension covers 2000–2010 and is made with EDGAR first and then EDGAR-FT or EDGAR-HTAP. The extension backward is also needed in almost all cases. It is done with CDIAC for fossil-fuel $CO_2$ based on EDGAR; with EDGAR-HYDE v1.4 (van Aardenne et al., 2001) for $N_2O$ based on EDGAR or EPA; and with ACCMIP

for any other species based on EDGAR or EPA. For $N_2O$, however, given that the EDGAR-HYDE dataset has global values that differ significantly from more recent estimates, the dataset is rescaled to the global estimates by Davidson (2009) before being used for extension. For non-$CO_2$ species, the obtained time-series are then linearly extrapolated from 1850, or 1860 in the case of $N_2O$: to be zero in 1500 for biogenic emissions, and in 1750 for fossil-related emissions. The year 1500 is taken to be consistent with the dataset we use for land-use change (see section 2.2.2). For the HFCs and PFCs whose emissions are not

zero in 1970 (first year of the EDGAR inventory) the backward extrapolation is slightly different. From 1970, their emissions are extrapolated backward following a quadratic function of the time, to reach zero in a year taken from Meinshausen et al. (2011): 1930 for HFC-23, 1950 for $SF_6$, 1922 for $CF_4$ and 1889 for $C_2F_6$.

Finally, because of our assumption of a preindustrial equilibrium occuring before 1750, we have to offset the full time-series of anthropogenic emissions obtained thus far by the level of emissions of 1750, thus assuming that everything before that point

in time is part of the natural cycle – or at least negligible as an anthropogenic perturbation of this natural cycle. The obtained time-series of anthropogenic emissions for all species are shown in figure 2, except for the halogenated compounds which are shown in figure S1 in the supplement.

### 2.2.2 Land-use and land-cover change

As OSCAR embeds a book-keeping module to endogenously calculate $CO_2$ emissions from land-use change, it needs a specific

type of drivers related to land-use and land-cover change (LULCC). These are threefold. The first driver is for land-cover change itself ($\delta A^{b_1 \rightarrow b_2}$; an area per unit time), it is the human-induced transitions from one biome $b_1$ to another $b_2$. The second driver is for wood harvest ($\delta H^b$; a mass of carbon per unit time), it is the extraction of woody biomass from a given biome $b$ but without changing the land-cover, and it can be seen as a coarse forestry driver. The third driver is for shifting cultivation ($\delta S^{b_1 \leftrightarrow b_2}$; an area per unit time), it is the transitions from one natural biome $b_1$ to another anthropogenic $b_2$ which occurs

simultaneously with the reciprocal transitions. The latter driver is therefore a triangular matrix on the $(b_1, b_2)$ axes, and it is typical of – but not exclusive to – the agricultural practice happening in the tropics and known as "slash-and-burn".

In this version of OSCAR, only one LULCC dataset is available: the LUH v1.1 dataset (Hurtt et al., 2011) updated for the last TRENDY exercise (Sitch et al., 2015). Given that this dataset provides information only for an aggregated "natural vegetation" biome, whereas OSCAR considers different natural biomes, we need to combine the dataset with the natural vegetation maps





used for the terrestrial carbon-cycle in section 2.3.2, so as to disaggregate further the natural vegetation provided by the LUH project. Other than that, the dataset is used as is over the 1750–2010 period; it is shown in figure S2.

### 2.2.3 Radiative forcings

Finally, some remaining known climate forcings are prescribed to OSCAR directly as radiative forcings. This is the case of one

anthropogenic forcing: aviation contrails and induced cirrus ($\mathrm{RF_{con}}$); and of two natural forcings: volcanic aerosols ($\mathrm{RF_{volc}}$) and solar irradiance ($\mathrm{RF_{solar}}$). Those drivers are directly taken from IPCC (2013), except that we offset the volcanic forcing by its value averaged over 1750–2011, thus assuming this value to be representative of volcanic preindustrial conditions.

## 2.3 Carbon dioxide

### 2.3.1 Ocean carbon-cycle

The ocean carbon-cycle module is based on the mixed-layer impulse response developed by Joos et al. (1996), widely used among compact models (e.g. Meinshausen et al., 2011; Raupach et al., 2011), albeit with three important modifications. First, the convolution with the impulse response function is written as its equivalent box-model, similarly to what Harman et al. (2011) did. Second, the *ad hoc* function used to emulate the carbonate chemistry is updated (Harman et al., 2011) and it now includes a dependency on sea surface temperature. Third, we extend the initial formulation to include a varying mixed layer

depth, assumed to vary with global sea surface temperature in order to represent the stratification of the upper ocean induced by global warming (e.g. Capotondi et al., 2012). With the last two modifications, key mechanisms of the global ocean uptake – such as carbonate saturation, warming-driven changes in solubility and impact of ocean stratification (Ciais et al., 2013b) – are better accounted for in OSCAR.

Following Joos et al. (1996), we explicitly represent only one oceanic carbon pool which corresponds to the surface ocean

($C_{\mathrm{surf}}$); other oceanic carbon pools are only implicitly considered. The two carbon fluxes – in and out – between this surface ocean and the atmosphere are then calculated separately. They both are proportional to the gaz exchange rate ($\nu_{\mathrm{fg}}$) and to the atmospheric conversion factor for $CO_2$ ($\alpha_{\mathrm{atm}}^{\mathrm{CO2}}$). The latter is only used to express a partial pressure of $CO_2$ – in ppm – into a quantity of carbon – in GtC. The in-going flux ($F_{\mathrm{in}}$) is a linear function of the atmospheric partial pressure in $CO_2$ (CO2):

$$\Delta F_{\mathrm{in}} = \nu_{\mathrm{fg}}\, \alpha_{\mathrm{atm}}^{\mathrm{CO2}}\, \Delta\mathrm{CO2}; \tag{1}$$

and the out-going flux ($F_{\mathrm{out}}$) is also a linear function of the sea surface partial pressure in $CO_2$. This partial pressure is calculated thanks to an *ad-hoc* function ($\mathcal{F}_{\mathrm{pCO_2}}$) designed to emulate the non-linear oceanic carbonate chemistry. It depends on dissolved inorganic carbon concentration in the surface ocean (dic) and sea surface temperature ($T_S$):

$$\Delta F_{\mathrm{out}} = \nu_{\mathrm{fg}}\, \alpha_{\mathrm{atm}}^{\mathrm{CO2}}\, \mathcal{F}_{\mathrm{pCO2}}\left[\Delta\mathrm{dic}, T_{S,0} + \Delta T_S\right]. \tag{2}$$





The dissolved inorganic carbon is deduced from the total carbon stored in the surface layer *via* a conversion factor ($\alpha_{\text{sol}}$), the global area of the ocean ($A_{\text{ocean}}$), and the mixed layer depth ($h_{\text{mld}}$):

$$\Delta\text{dic} = \frac{\alpha_{\text{sol}}}{\alpha_{\text{atm}}^{\text{CO2}}} \frac{h_{\text{mld},0}^{-1}}{A_{\text{ocean}}} \left(1 + \frac{\Delta h_{\text{mld}}}{h_{\text{mld},0}}\right)^{-1} \Delta C_{\text{surf}}; \tag{3}$$

where the mixed layer depth is assumed to vary according to the following law, parameterized by the maximum relative intensity of the stratification ($\pi_{\text{mld}} \in [0,1]$) and its sensitivity to sea surface temperature change ($\gamma_{\text{mld}} < 0$):

$$\Delta h_{\text{mld}} = h_{\text{mld},0} \, \pi_{\text{mld}} \left(\exp\left[\gamma_{\text{mld}} \, \Delta T_S\right] - 1\right). \tag{4}$$

We then represent the net effect of the oceanic circulation and mixing fluxes as a unique flux of carbon that goes from the surface ocean to an implict deep ocean ($F_{\text{circ}}$). To do so, the surface ocean is subdivided into several boxes (superscript $^o$). Each box contains a fraction of the total surface carbon and is assigned an areal fraction ($\pi_{\text{circ}}^o$; $\sum_o \pi_{\text{circ}}^o = 1$) and a turnover time ($\tau_{\text{circ}}^o$), so that it works as a first order model. So we have:

$$\tau_{\text{circ}}^o \, \Delta F_{\text{circ}}^o = \Delta C_{\text{surf}}^o. \tag{5}$$

Note that this subdivision of the surface ocean is not geographical: it only corresponds to the different turnover times of mixing between the surface and deep oceans, accounted for by the response function of Joos et al. (1996); and the boxes are not distinguished otherwise as they e.g. share the same carbonate chemistry. Finally, the global perturbation of the ocean carbon-cycle is obtained by summing over the $o$-boxes:

$$\Delta C_{\text{surf}} = \sum_o \Delta C_{\text{surf}}^o; \tag{6}$$

and by solving the carbon budget in each of the boxes:

$$\frac{\text{d}}{\text{d}t}\Delta C_{\text{surf}}^o = \pi_{\text{circ}}^o \, \Delta F_{\text{in}} - \pi_{\text{circ}}^o \, \Delta F_{\text{out}} - \Delta F_{\text{circ}}^o. \tag{7}$$

Note that this model implictly assumes no change in the biological pump – change that could be induced e.g. by changes in climate or nutrient availability (Ciais et al., 2013b).

The atmospheric conversion factor is calculated following Prather et al. (2012, table S2): a value of 0.1765 Tmol ppb$^{-1}$ of dry air is assumed, and it is multiplied by the molecular mass of any species X to obtain $\alpha_{\text{atm}}^{\text{X}}$. The conversion factor $\alpha_{\text{sol}}$ is set to 1.722 $10^{17}$ µmol m$^3$ ppm$^{-1}$ kg$^{-1}$ (Joos et al., 1996). The function $\mathcal{F}_{\text{pCO2}}$ can be either one of the two formulations (Pade-approximant or Power-law fits) given by Harman et al. (2011). The parameters $\nu_{\text{fg}}$, $A_{\text{ocean}}$, $h_{\text{mld},0}$, $\pi_{\text{circ}}$ and $\tau_{\text{circ}}$, as well as the preindustrial sea surface temperature $T_{S,0}$, are taken from Joos et al. (1996) who provide four sets of values derived from four ocean models of various complexity.

We use the latter study in a way very close to what is done by Harman et al. (2011). We take the long-term response functions of Joos et al. (1996) which, analytically, are a weighted sum of exponential terms: $\pi_{\text{circ}}$ are taken as the weights and $\tau_{\text{circ}}$ as the time constants. The number of $o$-boxes is thus equal to the number of exponential terms. To ensure that the response is



consistent in the short-term, however, we add to these another box whose $\pi_{\mathrm{circ}}$ is taken as the complementary fraction so that the sum of all fractions gives one, and whose $\tau_{\mathrm{circ}}$ is arbitrarily set to 1/3 year in the case of the "HILDA" and "Box-diffusion" models, and 1/2 year in the case of the "2D-Princeton" model. In the case of the "3D-Princeton" model, however, because the sum of the fractions provided by Joos et al. (1996) is greater than one, we do not add that other box, we simply reduce the
fraction of the fastest box so that the sum of the fractions is one.

The two parameters related to the mixed layer depth, i.e. $\pi_{\mathrm{mld}}$ and $\gamma_{\mathrm{mld}}$, can be calibrated on three CMIP5 Earth system models (see appendix B for a list of the models used to calibrate OSCAR). To do so, we use the CMIP5 output variable named "omlmax" which corresponds to the maximum depth of the mixed layer over a given period of time. We then fit the parameters, on the basis of equation (4), using the relative variation of the "omlmax" variable over the historical period and the RCP8.5 up
to 2300 with respect to the control simulation, and the sea surface temperature change. This fit is made with yearly data. Since for the "CESM1-BGC" model no value is available over 2100-2300, we use outputs from Randerson et al. (2015, data from figure 6b) instead. The CMIP5 fits are shown in figure S3.

### 2.3.2 Land carbon-cycle: intensive perturbation

Before considering the extensive perturbation of the terrestrial carbon-cycle, driven by land-use and land-cover changes, we
first focus on its intensive perturbation, i.e. the perturbation that changes the areal properties of the terrestrial ecosystems. This intensive perturbation is driven by changes in environmental conditions such as atmospheric $CO_2$, climate or nutrient deposition – albeit the latter not in this version of OSCAR. Since only the areal properties of the terrestrial biosphere are affected, this section only describes the evolution of OSCAR's state variables per unit area. The intensive variables, i.e. the variables per unit area, are thereafter written in lowercase; in opposition to the extensive variables, written in uppercase, that
we use in the next section.

The terrestrial carbon-cycle module is an upgrade of the previous versions (e.g. Gitz and Ciais, 2003; Gasser, 2014). The terrestrial biosphere is aggregated into several regions (superscript [i]) and further divided into various biomes (superscript [b]) in each region. Here, we note that the exact regional aggregation (both the number of regions and their definition), and to a lesser extent the way biomes are aggregated, can be chosen before every simulation, thus altering the numerical values of the related
parameters. For this description paper, we set the regional aggregation to the nine broad world regions used by (Houghton and Hackler, 2001, see also our figure S4), and the biome list to: bare soil, forests, mix of grasslands and shrublands, croplands, and pastures. Each doublet $(i, b)$ represents the "average" biome $b$ of the $i$-th region with assumed homogeneous biogeochemical characteristics. Each doublet $(i, b)$ is then represented as a three-box model where each box exchanges carbon with the others and/or the atmosphere.
Contrarily to complex models that simulate separately the gross primary productivity, our terrestrial carbon-cycle starts directly with the net primary productivity (NPP). The areal net productivity (npp) depends on a preindustrial intensity ($\eta$), and it responds to changes in atmospheric $CO_2$ *via* a fertilization function ($\mathcal{F}_{\mathrm{fert}}$), and changes in local surface temperature ($T_L$)





and local yearly precipitation ($P_L$) with assumed linear sensitivities ($\gamma_{\mathrm{npp},T}$ and $\gamma_{\mathrm{npp},P}$, respectively). This gives:

$$\Delta\mathrm{npp}^{i,b} =$$
$$\eta^{i,b}\left(\mathcal{F}_{\mathrm{fert}}^{i,b}[\Delta\mathrm{CO2}]\left(1+\gamma_{\mathrm{npp},T}^{i,b}\,\Delta T_L^i + \gamma_{\mathrm{npp},P}^{i,b}\,\Delta P_L^i\right)-1\right). \tag{8}$$

The functional form of $\mathcal{F}_{\mathrm{fert}}$ can be either logarithmic or hyperbolic (Friedlingstein et al., 1995). If logarithmic, it is a function

with one parameter that describes the intensity of the fertilization effect ($\beta_{\mathrm{npp}} > 0$):

$$\mathcal{F}_{\mathrm{fert}}^{i,b} = 1 + \beta_{\mathrm{npp}}^{i,b}\,\ln\left[1+\frac{\Delta\mathrm{CO2}}{\mathrm{CO2}_0}\right]; \tag{9}$$

and if hyperbolic, it is a function with two parameters: one also describing the intensity of the fertilization effect ($\tilde{\beta}_{\mathrm{npp}} > 1$) and the other being the compensation point ($\mathrm{CO2}_{\mathrm{cp}}$), i.e. the value of atmospheric $CO_2$ below which there is no NPP at all:

$$\mathcal{F}_{\mathrm{fert}}^{i,b} = \frac{1+\frac{\Delta\mathrm{CO2}}{\mathrm{CO2}_0-\mathrm{CO2}_{\mathrm{cp}}^{i,b}}}{\frac{\Delta\mathrm{CO2}}{\mathrm{CO2}_0}\left(\frac{1}{\tilde{\beta}_{\mathrm{npp}}^{i,b}}\frac{2\,\mathrm{CO2}_0-\mathrm{CO2}_{\mathrm{cp}}^{i,b}}{\mathrm{CO2}_0-\mathrm{CO2}_{\mathrm{cp}}^{i,b}}-1\right)+1}. \tag{10}$$

NPP fills a first carbon pool that corresponds to the vegetation's living biomass ($c_{\mathrm{veg}}$). This biomass is partly oxidized by wildfires, creating a flux to the atmosphere ($e_{\mathrm{fire}}$). This flux is proportional to the biomass available to be burnt and also depends on the preindustrial fire intensity ($\iota$) and a function representing the variation of this fire intensity ($\mathcal{F}_{\mathrm{igni}}$):

$$\Delta e_{\mathrm{fire}}^{i,b} =$$
$$\iota^{i,b}\,c_{\mathrm{veg},0}^{i,b}\left(\left(1+\frac{\Delta c_{\mathrm{veg}}^{i,b}}{c_{\mathrm{veg},0}^{i,b}}\right)\mathcal{F}_{\mathrm{igni}}^{i,b}\left[\Delta\mathrm{CO2},\Delta T_L^i,\Delta P_L^i\right]-1\right). \tag{11}$$

The change in fire intensity is a function of changes in atmospheric $CO_2$ – used as a proxy variable to encompass various effects such as change in leaf area index that would help wildfires to spread, or change in evapotranspiration and thus in soil moisture that would reduce their intensity – in local surface temperature, and in local yearly precipitation. We arbitrarily choose a linear sensitivity for each of the three environmental factors ($\gamma_{\mathrm{igni},C}$, $\gamma_{\mathrm{igni},T} > 0$ and $\gamma_{\mathrm{igni},P} < 0$, respectively). Here we note that our formulation implictly assumes that there is no direct human intervention to e.g. limit and control natural wildfires. So the

function $\mathcal{F}_{\mathrm{igni}}$ is formulated as:

$$\mathcal{F}_{\mathrm{igni}}^{i,b} = 1 + \gamma_{\mathrm{igni},C}^{i,b}\,\Delta\mathrm{CO2} + \gamma_{\mathrm{igni},T}^{i,b}\,\Delta T_L^i + \gamma_{\mathrm{igni},P}^{i,b}\,\Delta P_L^i. \tag{12}$$

The living biomass also partly dies at a fixed rate ($\mu$), which generates a flux we call "mortality" ($f_{\mathrm{mort}}$). The mortality rate does not depend on environmental conditions such as climate, but the lack of detailed outputs from complex models motivates this modelling choice. Thus:

$$\Delta f_{\mathrm{mort}}^{i,b} = \mu^{i,b}\,\Delta c_{\mathrm{veg}}^{i,b}. \tag{13}$$



The mortality flux goes into the litter carbon pool ($c_{\text{litt}}$), where heterotrophic respiration ($\text{rh}_{\text{litt}}$) occurs at a rate that depends on its own preindustrial value ($\rho_{\text{litt}}$) and a specific function of local climate conditions ($\mathcal{F}_{\text{resp}}$):

$$\Delta \text{rh}_{\text{litt}}^{i,b} =$$
$$\rho_{\text{litt}}^{i,b}\, c_{\text{litt},0}^{i,b} \left( \left(1 + \frac{\Delta c_{\text{litt}}^{i,b}}{c_{\text{litt},0}^{i,b}}\right) \mathcal{F}_{\text{resp}}^{i,b}\left[\Delta T_L^i, P_L^i\right] - 1 \right). \tag{14}$$

The functional form of $\mathcal{F}_{\text{resp}}$ can be either exponential or Gaussian (Tuomi et al., 2008) regarding its sensitivity to temperature. It is always linear regarding that to precipitations. If exponential, it is therefore a function with two parameters ($\gamma_{\text{resp},T} > 0$) and precipitations ($\gamma_{\text{resp},P}$):

$$\mathcal{F}_{\text{resp}}^{i,b} = \exp\left[\gamma_{\text{resp},T}^{i,b}\, \Delta T_L^i\right] \left(1 + \gamma_{\text{resp},P}^{i,b}\, \Delta P_L^i\right); \tag{15}$$

and if Gaussian, it is a function with three parameters, two of which being the sensitivity to temperature split between a first-order term ($\gamma_{\text{resp},T_1} > 0$) and a second-order term ($\gamma_{\text{resp},T_2} < 0$), and the third being the sensitivity to precipitations ($\tilde{\gamma}_{\text{resp},P}$):

$$\mathcal{F}_{\text{resp}}^{i,b} =$$
$$\exp\left[\gamma_{\text{resp},T_1}^{i,b}\, \Delta T_L^i + \gamma_{\text{resp},T_2}^{i,b}\, \Delta T_L^{i\,2}\right] \left(1 + \tilde{\gamma}_{\text{resp},P}^{i,b}\, \Delta P_L^i\right). \tag{16}$$

Part of the litter carbon is metabolized into soil organic carbon. This flux ($f_{\text{met}}$) is taken proportional to the heterotrophic respiration of the litter carbon pool (by a factor $\kappa_{\text{met}}$):

$$\Delta f_{\text{met}}^{i,b} = \kappa_{\text{met}}\, \Delta \text{rh}_{\text{litt}}^{i,b}. \tag{17}$$

Heterotrophic respiration ($\text{rh}_{\text{soil}}$) also occurs in the soil carbon pool ($c_{\text{soil}}$). It is a function of its preindustrial value ($\rho_{\text{soil}}$) and of the same function $\mathcal{F}_{\text{resp}}$ as for the litter:

$$\Delta \text{rh}_{\text{soil}}^{i,b} =$$
$$\rho_{\text{soil}}^{i,b}\, c_{\text{soil},0}^{i,b} \left( \left(1 + \frac{\Delta c_{\text{soil}}^{i,b}}{c_{\text{soil},0}^{i,b}}\right) \mathcal{F}_{\text{resp}}^{i,b}\left[\Delta T_L^i, P_L^i\right] - 1 \right). \tag{18}$$

And finally, the terrestrial carbon cycle of a given doublet $(i, b)$ follows:

$$\frac{\mathrm{d}}{\mathrm{d}t} \Delta c_{\text{veg}}^{i,b} = \Delta \text{npp}^{i,b} - \Delta e_{\text{fire}}^{i,b} - \Delta f_{\text{mort}}^{i,b}; \tag{19}$$

$$\frac{\mathrm{d}}{\mathrm{d}t} \Delta c_{\text{litt}}^{i,b} = \Delta f_{\text{mort}}^{i,b} - \Delta \text{rh}_{\text{litt}}^{i,b} - \Delta f_{\text{met}}^{i,b}; \tag{20}$$

$$\frac{\mathrm{d}}{\mathrm{d}t} \Delta c_{\text{soil}}^{i,b} = \Delta f_{\text{met}}^{i,b} - \Delta \text{rh}_{\text{soil}}^{i,b}. \tag{21}$$

The equation system described above by equations (19), (20) and (21) implies that our preindustrial equilibrium is:

$$\begin{cases} \text{npp}_0^{i,b} = f_{\text{mort},0}^{i,b} + e_{\text{fire},0}^{i,b} \\ f_{\text{mort},0}^{i,b} = \text{rh}_{\text{litt},0}^{i,b} + f_{\text{met},0}^{i,b} \; ; \\ f_{\text{met},0}^{i,b} = \text{rh}_{\text{soil},0}^{i,b} \end{cases} \tag{22}$$





which, in terms of flux parameters and preindustrial carbon stocks, is equivalent to:

$$
\begin{cases}
\eta^{i,b} = \left( \mu^{i,b} + \iota^{i,b} \right) c_{\text{veg},0}^{i,b} \\
\mu^{i,b} \, c_{\text{veg},0}^{i,b} = (1 + \kappa_{\text{met}}) \, \rho_{\text{litt}}^{i,b} \, c_{\text{litt},0}^{i,b} \\
\kappa_{\text{met}} \, \rho_{\text{litt}}^{i,b} \, c_{\text{litt},0}^{i,b} = \rho_{\text{soil}}^{i,b} \, c_{\text{soil},0}^{i,b}
\end{cases}
\tag{23}
$$

Note that to obtain the global preindustrial terrestrial carbon-cycle equilibrium one needs to multiply the above equilibrum by the preindustrial biome area extents ($A_0$), for instance: $\text{NPP}_0^{\text{global}} = \sum_{i,b} \text{NPP}_0^{i,b} = \sum_{i,b} \text{npp}_0^{i,b} A_0^{i,b}$. Note also that the extensive perturbation, described in the next section, alters the biome area extents so that we actually have $A^{i,b} = A_0^{i,b} + \Delta A^{i,b}$.

The parameters for the preindustrial fluxes (i.e. $\eta$, $\mu$, $\rho_{\text{litt}}$ and $\rho_{\text{soil}}$) can be calibrated on nine TRENDY v2 dynamic global vegetation models (Le Quéré et al., 2014; Sitch et al., 2015). To do so, we use the first thirty years of the so-called "S2" simulation, in which changing climate and $CO_2$ are prescribed to the models but no land-use change happens. We assume the average fluxes and pools over that period are at equilibrium, so that we can deduce the parameters from equation (23), taking $\kappa_{\text{met}} = 0.3/0.7$ (Foley, 1995). For the few models that do not report separately the litter pool, we assume the total reported soil carbon pool is made at 5% of litter carbon and 95% of soil carbon. Also, to account for the harvest of croplands, we alter the parameters of this biome following some arbitrary rules: NPP is reduced by 80%, thus we assume this fraction of the crops' productivity is harvested and oxidized within a year; and the mortality rate is set to $1 \, \text{yr}^{-1}$, which corresponds to a yearly harvest. Also, because the assumed preindustrial equilibrium based on TRENDY is 1901–1930 and not 1750, we scale down the NPP parameter $\eta$ by a factor equal to the ratio of our preindustrial atmospheric $CO_2$ over the one for the TRENDY preindustrial period i.e. by a factor of about 0.92.

The parameters for the transient response of NPP and hetetrotrophic respiration (i.e. $\beta_{\text{npp}}$, $\tilde{\beta}_{\text{npp}}$, $\text{CO2}_{\text{cp}}$, $\gamma_{\text{npp},T}$, $\gamma_{\text{npp},P}$, $\gamma_{\text{resp},T}$, $\gamma_{\text{resp},T_1}$, $\gamma_{\text{resp},T_2}$, $\tilde{\gamma}_{\text{resp},P}$) can be calibrated on seven CMIP5 Earth system models (see e.g. Arora et al., 2013). To do so, we use the outputs from three CMIP5 simulations: "1pctCO2", "esmFixClim", "esmFdbk1" which correspond to simulations with an increase of atmospheric $CO_2$ of +1% $\text{yr}^{-1}$ in the case of a fully coupled configuration, a fixed climate, or a fixed carbon-cycle, respectively. Depending on the functional form chosen, the fit for NPP is done on the basis of equations (8)+(9) or (8)+(10). That of the heterotrophic respiration rate is done on the basis of equation (15) or (16). The calibration is done in two steps. A first fit is made with decadal averages of the relevant variables and for which the parameter related to local precipitations is set to zero. A second fit is then made with annual values to find the remaining parameter. This approach is used to avoid over-fitting. The fit is made over the three simulations at the same time, using the 'piControl' values to define the preindustrial equilibrium. In the case of the respiration rate, we also add a new term to equation (15) or (16) to calibrate the parameters. We multiply $\mathcal{F}_{\text{resp}}$ by the term $(1 + \beta_{\text{prim}}^{i,b} \, \Delta F_{\text{input}}^{i,b} / F_{\text{input},0}^{i,b})$, where $\beta_{\text{prim}}$ is a new sensitivity and $F_{\text{input}}$ is the input carbon flux from the vegetation pool to the soil pool, so as to account for the "false priming" effect observed in CMIP5 models (Koven et al., 2015) – that is, the effect of an increased respiration not because of increased respiration rate *per se*, but because of new carbon falling into a pool with faster turnover time than the average turnover time of the soil. This additional term is only used for calibration purpose and thus it is not added to OSCAR's formulation because the two soil



boxes of OSCAR are expected to provide this "false priming" effect. The CMIP5 fits are shown in figures S5 to S11 for NPP, and figures S12 to S18 for heterotrophic respiration.

The fire-related parameters are similarly calibrated on TRENDY (for $\iota$) and CMIP5 (for $\gamma_{\mathrm{igni},C}$, $\gamma_{\mathrm{igni},T}$, $\gamma_{\mathrm{igni},P}$) but this is done independently from the other parameters. Six models with wildfire emissions are available to calibrate on TRENDY, and

four models are to calibrate on CMIP5. As previously, we alter the parameters obtained for croplands: we assume there is no wildfire at all within that biome. The CMIP5 fits are shown in figures S19 to S22 for fire intensity. Given how experimental it is to include fire processes in a model as simple as OSCAR, we also keep an option to turn off the preindustrial wildfire flux and/or its transient response.

Regarding the TRENDY and CMIP5 data processing, it has to be noted that none of the models provide biome-specific

outputs. So we choose to deduce biome-specific data by weighting the biome-aggregated outputs of a model by its biome area fraction map, taken to the power 3. This approach is used to give more importance – in a given region – to the gridcells in which biomes are purer, without taking the risk of having too few of those gridcells if we were to set a threshold of biome area fraction instead. Also, some of the complex models used to calibrate OSCAR are lacking some of the biomes implemented in our model. Thus, we need rules to establish parameters for the lacking biomes on the basis of the available ones. When

croplands are not in a model, we assume they have the same biogeochemical properties as grasslands, before any harvest or other human intervention. When pastures are not in a model, we assume their biogeochemical parameters are a mix of those of grasslands and bare soil, at 60% and 40% respectively. In a configuration of OSCAR in which shrublands are separated from grasslands – which is not the case in this paper – and shrublands are not in a model, we assume they are made at 85% of grasslands and 15% of forests.

The preindustrial area extents $A_0$ are obtained by combining the preindustrial land-use map consistent with the LULCC drivers (see section 2.2.2) for the anthropogenic biomes and one of thirteen vegetation maps to distinguish between the natural biomes. The first map is used to know the fractions of water/ice, croplands and pastures, in a given gridcell. The remaining fraction corresponds to natural vegetation, and this fraction is then subdivided into our different natural biomes following their proportions in each gridcell of the second map. Of the thirteen possible vegetation maps, two are recent observations of land-

cover, MODIS (Channan et al., 2014) and ESA-CCI (2015), two are potential natural vegetation maps (Ramankutty and Foley, 1999; Levavasseur et al., 2012), and the other ones are the land-cover map of the same TRENDY models used to calibrate the preindustrial carbon fluxes and pools. In the first four cases, given that the maps provide land-cover as 'land-cover classes' and not as 'plant functional types' – as used by TRENDY models – we use the cross-walking table developed by Poulter et al. (2011) to convert the former into the latter.

### 2.3.3 Land carbon-cycle: extensive perturbation

Now we consider the extensive perturbation of the terrestrial carbon-cycle, i.e. the one driven by changes in land-use and land-cover. This perturbation has a first-order effect that originates from the human-induced disturbance of a given biome which then transition from its disturbed state to a new equilibrium state. When both extensive – change in biome extent – and intensive – change in areal properties – perturbations occur at the same time, their interaction creates a second-order effect, which is





also included in the following equations. Here, we also note that in theory another extensive perturbation affects the terrestrial ecosystems: the migration of natural biomes caused by changes in environmental conditions (e.g. Jones et al., 2009). This is however not included in this version of OSCAR.

The book-keeping module used to estimate the carbon fluxes induced by the land-use drivers is very close to that of the
previous version of OSCAR (Gasser, 2014). It is built on the approach developed by Gitz and Ciais (2003), although it now includes algorithms to treat not only land-cover change but also wood harvest and shifting cultivation. See section 2.2.2 for a description of those drivers. Following the discussion and recommendation by Gasser and Ciais (2013), the book-keeping is written so as to follow exactly the carbon fluxes and pools of transitioning ecosystems with regard to their expected but yet-to-be-reached new equilibrium, so that the effect of the LUC perturbation tends toward zero in the case of infinitely old
land-use disturbances. This corresponds to "definition 3" of Gasser and Ciais (2013) and to "definition B" of Pongratz et al. (2014).

For the book-keeping itself, we need to define a new series of extensive state variables for the terrestrial biosphere affected by LULCC (subscript $_\text{luc}$). These variables are defined following three axes: the region $i$ axis, the biome $b$ axis, and a new age-class $a$ axis; so that the triplet $(i, b, a)$ represents the "average" biome $b$ of the $i$-th region that was originally disturbed at
$t = a$. This implies that at any given time $t$, all the variables with $a > t$ are nil.

The initialization of the book-keeping sequence, i.e. the initial disturbed state of a given triplet $(i, b, a)$, depends on the kind of land-use disturbance. When land-cover change occurs, i.e. when there is conversion of a given land area from one biome $b_1$ to another biome $b_2$ ($\delta A^{b_1 \rightarrow b_2}$), we assume that all the living biomass of $b_1$ is taken away, and the living biomass of $b_2$ has yet to grow. When harvest occurs, we assume that the total amount of harvested biomass ($\delta H^b$) is taken from the living
biomass pool of $b$, and this biomass will regrow in time. When shifting cultivation occurs, we assume it can be approximated by the harvest of all the living biomass over the shifting area ($\delta S^{b_1 \leftrightarrow b_2}$) of both biomes $b_1$ and $b_2$, except that the biomes are considered not to be fully grown. Their age is assumed to be equal to the shifting cultivation turnover rate ($\tau_\text{shift}$), and thus their living biomass pool is taken equal to that of their fully grown counterpart multiplied by a factor $\pi_\text{shift}^{i,b} = 1 - \exp[-\mu^{i,b}\,\tau_\text{shift}]$. So, the initialization of the LUC-disturbed living biomass ($C_\text{veg,luc}$) is:

$$
\Delta C_\text{veg,luc}^{i,b_2,a=t} =
$$
$$
-\left(c_\text{veg,0}^{i,b_2} + \Delta c_\text{veg}^{i,b_2}\right) \sum_{b_1} \delta A^{i,b_1 \rightarrow b_2}
$$
$$
-\delta H^{i,b_2}
$$
$$
-\left(c_\text{veg,0}^{i,b_2} + \Delta c_\text{veg}^{i,b_2}\right) \pi_\text{shift}^{i,b_1} \sum_{b_1} \delta S^{i,b_1 \leftrightarrow b_2}. \tag{24}
$$

In the case of land-cover change and shifting cultivation, the above-ground fraction ($\pi_\text{agb}$) of the biomass of the original biome
$b_1$ is partly harvested and allocated to three harvested wood product pools ($C_\text{hwp,luc}$), following allocation coefficients ($\pi_\text{hwp}$). Each wood product pool (supercript $^w$) has a specific decay time ($\tau_\text{hwp}$) that corresponds to a specific use ($w = 1$ is fuelwood, $w = 2$ is pulp-based products, $w = 3$ is hardwood-based products). In the case of harvest, all the harvested biomass follows the





same allocation coefficients. It gives the following initialization of the harvested wood products:

$$\Delta C_{\mathrm{hwp,luc}}^{w,i,b_1,a=t} =$$
$$+ \pi_{\mathrm{hwp}}^{w,i,b_1} \pi_{\mathrm{agb}}^{i,b_1} \left( c_{\mathrm{veg},0}^{i,b_1} + \Delta c_{\mathrm{veg}}^{i,b_1} \right) \sum_{b_2} \delta A^{i,b_1 \to b_2}$$
$$+ \pi_{\mathrm{hwp}}^{w,i,b_1} \, \delta H^{i,b_1}$$
$$+ \pi_{\mathrm{hwp}}^{w,i,b_1} \pi_{\mathrm{agb}}^{i,b_1} \left( c_{\mathrm{veg},0}^{i,b_1} + \Delta c_{\mathrm{veg}}^{i,b_1} \right) \pi_{\mathrm{shift}}^{i,b_1} \sum_{b_2} \delta S^{i,b_1 \leftrightarrow b_2}. \tag{25}$$

For all three kinds of land-use disturbance, the remaining fraction of the living biomass of the original biome $b_1$ is added to the litter carbon pool ($C_{\mathrm{litt,luc}}$) of the new biome $b_2$. This fraction is usually called "slash". Also, in the case of land-cover change, the soil carbon pool of $b_1$ has yet to transition to that of $b_2$. This transition will lead to additional carbon fluxes. The initialization of the LUC-disturbed litter carbon variable is thus:

$$\Delta C_{\mathrm{litt,luc}}^{i,b_2,a=t} =$$
$$+ \sum_{b_1} \left( c_{\mathrm{litt},0}^{i,b_1} + \Delta c_{\mathrm{litt}}^{i,b_1} - c_{\mathrm{litt},0}^{i,b_2} - \Delta c_{\mathrm{litt}}^{i,b_2} \right) \delta A^{i,b_1 \to b_2}$$
$$+ \sum_{b_1} \left( 1 - \pi_{\mathrm{agb}}^{i,b_1} \sum_w \pi_{\mathrm{hwp}}^{w,i,b_1} \right) \left( c_{\mathrm{veg},0}^{i,b_1} + \Delta c_{\mathrm{veg}}^{i,b_1} \right) \delta A^{i,b_1 \to b_2}$$
$$+ \left( 1 - \sum_w \pi_{\mathrm{hwp}}^{w,i,b_2} \right) \delta H^{i,b_2}$$
$$+ \sum_{b_1} \left( 1 - \pi_{\mathrm{agb}}^{i,b_1} \sum_w \pi_{\mathrm{hwp}}^{w,i,b_1} \right) \left( c_{\mathrm{veg},0}^{i,b_1} + \Delta c_{\mathrm{veg}}^{i,b_1} \right) \pi_{\mathrm{shift}}^{i,b_1} \delta S^{i,b_1 \leftrightarrow b_2}. \tag{26}$$

Only in the case of land-cover change is the LUC-disturbed soil carbon pool ($C_{\mathrm{soil,luc}}$) initialized by the difference in soil carbon density between the original and the new biomes. Therefore we assume harvest and shifting cultivation do not directly – i.e. at the initialization step – disturb the soil carbon pool. So, it simply gives:

$$\Delta C_{\mathrm{soil,luc}}^{i,b_2,a=t} =$$
$$+ \sum_{b_1} \left( c_{\mathrm{soil},0}^{i,b_1} + \Delta c_{\mathrm{soil}}^{i,b_1} - c_{\mathrm{soil},0}^{i,b_2} - \Delta c_{\mathrm{soil}}^{i,b_2} \right) \delta A^{i,b_1 \to b_2}. \tag{27}$$

Here, it should be outlined that the initialization round is carbon-neutral to the atmosphere: carbon is moved between the three biospheric pools and the wood product pools, but none of it is emitted yet. And finally, the change in biome area extents is also calculated. It is per definition:

$$\frac{\mathrm{d}}{\mathrm{d}t} \Delta A^{i,b} = \sum_{b_1} \delta A^{i,b_1 \to b} - \sum_{b_2} \delta A^{i,b \to b_2}. \tag{28}$$





Once the initialization round is done, the LUC-disturbed biospheric pools follow the same carbon-cycle as the one described in the previous section for undisturbed biomes:

$$\frac{\mathrm{d}}{\mathrm{d}t}\Delta C_{\mathrm{veg,luc}}^{i,b,a} =$$
$$- \iota^{i,b}\,\mathcal{F}_{\mathrm{igni}}^{i,b}\,\Delta C_{\mathrm{veg,luc}}^{i,b,a} - \mu^{i,b}\,\Delta C_{\mathrm{veg,luc}}^{i,b,a}; \tag{29}$$

$$\frac{\mathrm{d}}{\mathrm{d}t}\Delta C_{\mathrm{litt,luc}}^{i,b,a} =$$
$$+ \mu^{i,b}\,\Delta C_{\mathrm{veg,luc}}^{i,b,a} - (1 + \kappa_{\mathrm{met}})\,\rho_{\mathrm{litt}}^{i,b}\,\mathcal{F}_{\mathrm{resp}}^{i,b}\,\Delta C_{\mathrm{litt,luc}}^{i,b,a}; \tag{30}$$

$$\frac{\mathrm{d}}{\mathrm{d}t}\Delta C_{\mathrm{soil,luc}}^{i,b,a} =$$
$$+ \kappa_{\mathrm{met}}\,\rho_{\mathrm{litt}}^{i,b}\,\mathcal{F}_{\mathrm{resp}}^{i,b}\,\Delta C_{\mathrm{litt,luc}}^{i,b,a} - \rho_{\mathrm{soil}}^{i,b}\,\mathcal{F}_{\mathrm{resp}}^{i,b}\,\Delta C_{\mathrm{soil,luc}}^{i,b,a}. \tag{31}$$

Note that these equations are affected by environmental conditions through the $\mathcal{F}_{\mathrm{igni}}$ and $\mathcal{F}_{\mathrm{resp}}$ functions, but the arguments of these functions are not shown for legibility. There is no term for NPP in equation (29) because the cycle described here is the LUC-disturbed cycle (see Gasser and Ciais, 2013). And therefore, because in this version of OSCAR there is no difference of NPP between a disturbed biome and its undisturbed counterpart, the LUC-disturbed NPP is zero.

As for the harvested wood products, they are oxidized at a varying rate that depends on the characteristic time of the pool (i.e. on $\tau_{\mathrm{hwp}}$) and also on a function ($\mathcal{F}_{\mathrm{hwp}}$) of the time passed since they were harvested (i.e. a function of $t - a$):

$$\tau_{\mathrm{hwp}}^{w}\,\frac{\mathrm{d}}{\mathrm{d}t}\Delta C_{\mathrm{hwp,luc}}^{w,i,b,a} = -\mathcal{F}_{\mathrm{hwp}}^{w}\,[t - a]\,\Delta C_{\mathrm{hwp,luc}}^{w,i,b,a}. \tag{32}$$

The function $\mathcal{F}_{\mathrm{hwp}}$ is introduced to allow choice of the temporal profile of the wood product oxidation. For instance, if $\mathcal{F}_{\mathrm{hwp}} \equiv 1$ the products are oxidized following an exponential profile (e.g. Houghton and Hackler, 2001). Alternatively to the exponential option, the profile can be linear (McGuire et al., 2001) or it can follow a gamma-function (Earles et al., 2012). The oxidation profiles and the corresponding functions $\mathcal{F}_{\mathrm{hwp}}$ are shown in figure 3.

The $\tau_{\mathrm{shift}}$ parameter is set to 15 years (Hurtt et al., 2011). The above-ground biomass fractions $\pi_{\mathrm{agb}}$ can be calibrated on three TRENDY models, exactly as other preindustrial carbon-cycle parameters are (see section 2.3.2). The allocation coefficients of the harvested wood products $\pi_{\mathrm{hwp}}$ come from the work by Earles et al. (2012, table S1). Those being national, however, they are aggregated to obtain regional values by weighting them with the national estimates of above-ground biomass in forests assessed by FAO (2010, table 2). To introduce more variation in our modelling, we have two options for processing the data. In the 'low' biomass burning option, we assume all the "non-merchandable" biomass of Earles et al. (2012) becomes slash; while in the 'high' biomass burning option, we assume 50% of it is added to the fuelwood pool ($w = 1$). Finally, the time constants of oxidation of the wood products $\tau_{\mathrm{hwp}}$ can come either from Earles et al. (2012) – which is based on the IPCC guidelines – with values of 0.5, 2 and 30 years for $w = 1$, 2 and 3, respectively; or from Houghton and Hackler (2001) with values of 1, 10 and 100 years, respectively.

### 2.3.4 Atmospheric CO$_2$ and RF

The incremental change in atmospheric CO$_2$ can be written as the balance between two sources: fossil-fuel and cement emissions ($E_{\mathrm{FF}}$; see section 2.2.1) and land-use change emissions ($E_{\mathrm{LUC}}$); and two sinks: the ocean sink ($F_{\downarrow\mathrm{ocean}}$) and the land





sink ($F_{\downarrow\text{land}}$). Note that despite being usually called "source" and "sink", since it is their historical role, each term of the budget can theoretically be of the opposite sign, thus changing from a source to a sink or *vice versa*. Mathematically:

$$\alpha_{\text{atm}}^{\text{CO2}} \frac{\mathrm{d}}{\mathrm{d}t}\Delta\text{CO2} = E_{\text{FF}} + \Delta E_{\text{LUC}} + \Delta F_{\downarrow\text{ocean}} + \Delta F_{\downarrow\text{land}}; \tag{33}$$

where, on the basis of the three previous sections, we have:

$$\Delta F_{\downarrow\text{ocean}} = \Delta F_{\text{out}} - \Delta F_{\text{in}}; \tag{34}$$

$$\Delta F_{\downarrow\text{land}} =$$
$$\sum_{i,b} \left(\Delta\text{rh}_{\text{soil}}^{i,b} + \Delta\text{rh}_{\text{litt}}^{i,b} + \Delta e_{\text{fire}}^{i,b} - \Delta\text{npp}^{i,b}\right)\left(A_0^{i,b} + \Delta A^{i,b}\right); \tag{35}$$

$$\Delta E_{\text{LUC}} =$$
$$-\frac{\mathrm{d}}{\mathrm{d}t}\sum_{i,b,a}\Delta C_{\text{veg,luc}}^{i,b,a} + \Delta C_{\text{litt,luc}}^{i,b,a} + \Delta C_{\text{soil,luc}}^{i,b,a} + \sum_w \Delta C_{\text{hwp,luc}}^{w,i,b,a}. \tag{36}$$

The radiative forcing (RF) induced by the increase in atmospheric $CO_2$ follows the logarithmic formula by Myhre et al. (1998):

$$\Delta\text{RF}^{\text{CO2}} = \alpha_{\text{rf}}^{\text{CO2}}\ln\left[1 + \frac{\Delta\text{CO2}}{\text{CO2}_0}\right]; \tag{37}$$

where $\alpha_{\text{rf}}^{\text{CO2}}$ = 5.35 W m$^{-2}$ is given by Myhre et al. (2013b, table 8.SM.1). For the preindustrial atmospheric concentration, we take $\text{CO2}_0$ = 278 ppm (IPCC, 2013, table AII.1.1a).

## 2.4 Non-CO₂ species

This intermediary section is dedicated to two elements which will be needed hereafter for non-CO₂ species: first, the endogenous estimate of the emission of a given species from biomass burning; and second, the estimate of the lagged concentration of a given species, assumed to be a proxy of its mid-stratospheric concentration.

### 2.4.1 Biomass burning

The atmospheric CO₂ budget above does not isolate the fluxes caused by biomass burning from those caused by all other sources of oxidation. But the biomass burning emissions are needed for non-CO₂ species in the next sections. Biomass burning emissions are altered by two aspects of the carbon-cycle: one that relates to the land sink $F_{\downarrow\text{land}}$, and one that relates to the land-use change emissions $E_{\text{LUC}}$. The former can be isolated in equation (35) as being induced by changes in areal fire intensities and in land-cover. The latter can be isolated in equation (36) as being induced by change in living biomass stocks – itself induced by LULCC – and by the oxidation of the harvested wood product pool corresponding to fuelwood ($w = 1$). From these two CO₂ fluxes, we deduce the non-CO₂ ones by assuming that the emission of a species X from biomass burning ($E_{\text{bb}}^{\text{X}}$) is





proportional (by a factor $\alpha_{bb}^X$) to that of $CO_2$, which gives:

$$
\Delta E_{bb}^{X,i} = \\
+ \sum_b \alpha_{bb}^{X,i,b} \left( e_{fire,0}^{i,b} \, \Delta A^{i,b} + \Delta e_{fire}^{i,b} \, A_0^{i,b} + \Delta e_{fire}^{i,b} \, \Delta A^{i,b} \right) \\
+ \sum_{b,a} \alpha_{bb}^{X,i,b} \left( \iota_0^{i,b} \, f_{igni}^{i,b} \, \Delta C_{veg,luc}^{i,b,a} - \frac{d}{dt} \Delta C_{hwp,luc}^{w=1,i,b,a} \right).
\tag{38}
$$

The $\alpha_{bb}$ parameters come from the GFED v3.1 database (van der Werf et al., 2010). The biomass burning emissions of all species are averaged over the whole available time-period, and to each vegetation type – or sector – of GFED is associated a biome of OSCAR: 'def' and 'for' are forests, 'woo' is shrublands, 'sav' is grasslands, 'agr' is croplands; 'pea', i.e. peatlands, are left alone. As in section 2.3.2, pastures are assumed to be 60% grasslands and 40% bare soil. The parameters are then obtained by simply taking the ratio of the emissions of a given species over those of $CO_2$.

### 2.4.2   Lagged concentrations

In the next sections, we need an estimate of the stratospheric concentration change of some species. For relatively long-lived species, we assume the stratospheric concentration change of this species can be approximated by its change in atmospheric concentration (X), albeit with a time-lag ($\tau_{lag}$). This change in "lagged" concentration ($X_{lag}$) is formulated as:

$$
\tau_{lag} \frac{d}{dt} X_{lag} = \Delta X - \Delta X_{lag}.
\tag{39}
$$

This formula is a linearized form of the usual equation written with a delay (e.g. Newman et al., 2007): $\Delta X_{lag}[t] = \Delta X[t - \tau_{lag}]$. We opt for the linearized form because it is easier to implement in a numerical model, and because it allows the time-lag to vary with time – although, it is not the case in this version of OSCAR.

We set $\tau_{lag}$ to a value of 3 yr. That value corresponds broadly to the mean age of air in the mid-latitudes of the stratosphere (e.g. Newman et al., 2007). We also note that this approach to model stratospheric concentration, without an explicit represen-
tation of the stratosphere-troposphere exchange, does not hold for too short-lived species, i.e. for species with a lifetime lower than the time-lag parameter. This is one of the reasons why another approach is used for ozone in section 2.8.

### 2.5   Methane

### 2.5.1   Atmospheric sinks

The oxidation of atmospheric methane follows the same modelling approach as that of the previous version (Gasser, 2014)
or of other simple models (e.g. Meinshausen et al., 2011). It is represented by a one-box model with one specific lifetime associated to each oxidative process. Those lifetimes may vary with time so that the resulting model is not linear.

The flux of oxidized $CH_4$ ($F_\downarrow^{CH4}$) is caused by four processes (e.g. Prather et al., 2012): tropospheric oxidation by the hydroxyl radical (preindustrial lifetime $\tau_{OH}^{CH4}$), stratospheric oxidation ($\tau_{h\nu}^{CH4}$), oxidation in dry soils ($\tau_{soil}^{CH4}$), and oxidation in the oceanic boundary layer ($\tau_{ocean}^{CH4}$). Transient change in the availability of hydroxyl radicals in the troposphere is a function
($\mathcal{F}_{OH}$) of external factors: the atmospheric $CH_4$ concentration itself (CH4); the stratospheric ozone concentration (O3s) which





drives the actinic flux partially generating OH; global surface temperature ($T_G$) which is used to estimate changes in global atmospheric temperature and relative humidity; and emission of the three ozone precursors, represented in the form of another function ($\mathcal{F}_{\text{prec}}$) for now. For the stratospheric sink, the lagged $CH_4$ concentration is used instead of the atmospheric one, and its actual lifetime is also a function ($\mathcal{F}_{\text{h}\nu}$) which rationale and formulation are detailed in section 2.6.1. Using also the

5 atmospheric conversion factor $\alpha_{\text{atm}}^{\text{CH4}}$ defined in section 2.3.1, we can write:

$$
\begin{aligned}
\alpha_{\text{atm}}^{\text{CH4}-1}\,\Delta F_{\downarrow}^{\text{CH4}} =& \\
-\frac{\text{CH4}_0}{\tau_{\text{OH}}^{\text{CH4}}}& \left( \left(1+\frac{\Delta\text{CH4}}{\text{CH4}_0}\right) \mathcal{F}_{\text{OH}}\left[\Delta\text{CH4},\Delta\text{O3s},\Delta T_G,\mathcal{F}_{\text{prec}}\right]-1\right) \\
-\frac{\text{CH4}_0}{\tau_{\text{h}\nu}^{\text{CH4}}}& \left( \left(1+\frac{\Delta\text{CH4}_{\text{lag}}}{\text{CH4}_0}\right) \mathcal{F}_{\text{h}\nu}\left[\Delta\text{N2O}_{\text{lag}},\Delta\text{EESC},\Delta T_G\right]-1\right) \\
-\left(\frac{1}{\tau_{\text{soil}}^{\text{CH4}}}+\frac{1}{\tau_{\text{ocean}}^{\text{CH4}}}\right)&\Delta\text{CH4}.
\end{aligned}
\tag{40}
$$

The function $\mathcal{F}_{\text{OH}}$ mostly follows the formulation by Holmes et al. (2013). It is parameterized with chemical sensitivites of OH to: atmospheric $CH_4$ ($\xi_{\text{CH}_4}^{\text{OH}}$), stratospheric $O_3$ ($\xi_{\text{O3s}}^{\text{OH}}$), global atmospheric temperature ($\xi_{T_A}^{\text{OH}}$), and global atmospheric relative humidity ($\xi_{Q_A}^{\text{OH}}$). The absolute change in global atmospheric temperature ($T_A$) is assumed to be proportional (by a factor $\kappa_{T_A}$) to that in global surface temperature. The relative change in global atmospheric relative humidity is assumed to be propotional (by a factor $\kappa_{Q_A}$) to that in saturation vapor pressure. The latter follows an empirical function of global atmospheric temperature

change with two parameters ($\kappa_{\text{svp}}$ and $T_{\text{svp}}$). So far:

$$
\begin{aligned}
\ln\left[\mathcal{F}_{\text{OH}}\right] =& \\
+\xi_{\text{CH4}}^{\text{OH}}& \ln\left[1+\frac{\Delta\text{CH4}}{\text{CH4}_0}\right] \\
+\xi_{\text{O3s}}^{\text{OH}}& \ln\left[1+\frac{\Delta\text{O3s}}{\text{O3s}_0}\right] \\
+\xi_{T_A}^{\text{OH}}& \ln\left[1+\frac{\kappa_{T_A}\,\Delta T_G}{T_{A,0}}\right] \\
+\xi_{Q_A}^{\text{OH}}& \ln\left[1+\kappa_{Q_A}\left(\exp\left[\frac{\kappa_{\text{svp}}\,\kappa_{T_A}\,\Delta T_G}{T_{A,0}+T_{\text{svp}}}\right]-1\right)\right] \\
+\mathcal{F}_{\text{prec}}& \left[E_{\text{NOx}},\Delta E_{\text{bb}}^{\text{NOx}},E_{\text{CO}},\Delta E_{\text{bb}}^{\text{CO}},E_{\text{VOC}},\Delta E_{\text{bb}}^{\text{VOC}}\right].
\end{aligned}
\tag{41}
$$

The functional form of $\mathcal{F}_{\text{prec}}$ can be either linear (Ehhalt et al., 2001) or logarithmic (Holmes et al., 2013). In the linear case, it is a function parameterized with three absolute chemical sensitivities of OH to the ozone precursors: nitrogen oxides ($\xi_{\text{NOx}}^{\text{OH}}$), carbon monoxide ($\xi_{\text{CO}}^{\text{OH}}$) and non-methane volatile organic compounds ($\xi_{\text{VOC}}^{\text{OH}}$):

$$
\mathcal{F}_{\text{prec}} = \sum_{\substack{\text{X}\in\{\text{NOx},\\ \text{CO,VOC}\}}} \xi_{\text{X}}^{\text{OH}}\left(E_{\text{X}}+\sum_i \Delta E_{\text{bb}}^{\text{X},i}\right).
\tag{42}
$$



In the logarithmic case, it is parameterized by three relative chemical sensitivities ($\tilde{\xi}_{\mathrm{NOx}}^{\mathrm{OH}}$, $\tilde{\xi}_{\mathrm{CO}}^{\mathrm{OH}}$, $\tilde{\xi}_{\mathrm{VOC}}^{\mathrm{OH}}$), and the preindustrial natural emissions of the three ozone precursors ($E_{\mathrm{nat}}^{\mathrm{NOx}}$, $E_{\mathrm{nat}}^{\mathrm{CO}}$, $E_{\mathrm{nat}}^{\mathrm{VOC}}$). This gives:

$$\mathcal{F}_{\mathrm{prec}} = \sum_{\substack{\mathrm{X}\in\{\mathrm{NOx},\\ \mathrm{CO,VOC}\}}} \tilde{\xi}_{\mathrm{X}}^{\mathrm{OH}} \ln\left[1 + \frac{E_{\mathrm{X}} + \sum_i \Delta E_{\mathrm{bb}}^{\mathrm{X},i}}{E_{\mathrm{nat}}^{X}}\right]. \tag{43}$$

The four lifetimes of methane are taken as the present-day lifetimes given by Prather et al. (2012, tables A1 & A2): 11.2, 120, 150 and 200 years for $\tau_{\mathrm{OH}}^{\mathrm{CH4}}$, $\tau_{\mathrm{h}\nu}^{\mathrm{CH4}}$, $\tau_{\mathrm{soil}}^{\mathrm{CH4}}$ and $\tau_{\mathrm{ocean}}^{\mathrm{CH4}}$ respectively. The lifetime with regard to OH is then scaled down by an arbitrary factor of 0.80. We note that this does not follow the rescaling made by Prather et al. (2012) which was based on preliminary results from the ACCMIP models (Naik et al., 2013). The ACCMIP study is inconclusive about the change in methane lifetime between the preindustrial and present days: some models predict an increase while others predict a decrease. Because our function $\mathcal{F}_{\mathrm{OH}}$ is based on a subset of the ACCMIP models (see below) which all find the methane-OH lifetime increased, we scale down the preindustrial value of this lifetime, so that it roughly meets its present-day value during the simulation. Also, to introduce variation in this important parameter, we propose alternative values based on the ACCMIP chemistry-tranport models (Naik et al., 2013, table 1): optionally, the default lifetime can be rescaled by a factor equal to any of the sixteen model's estimate of the lifetime over the multi-model mean estimate. Finally, the stratospheric lifetime is also scaled up by a factor 1.06, following Prather et al. (2015) (see also section 2.6.1).

All the chemical sensitivities of the OH sink (i.e. $\xi_{\mathrm{CH4}}^{\mathrm{OH}}$, $\xi_{\mathrm{O3s}}^{\mathrm{OH}}$, $\xi_{T_A}^{\mathrm{OH}}$, $\xi_{Q_A}^{\mathrm{OH}}$, $\xi_{\mathrm{NOx}}^{\mathrm{OH}}$, $\xi_{\mathrm{CO}}^{\mathrm{OH}}$, $\xi_{\mathrm{VOC}}^{\mathrm{OH}}$, $\tilde{\xi}_{\mathrm{NOx}}^{\mathrm{OH}}$, $\tilde{\xi}_{\mathrm{CO}}^{\mathrm{OH}}$ and $\tilde{\xi}_{\mathrm{VOC}}^{\mathrm{OH}}$) are taken as one of the four sets of values from the study by Holmes et al. (2013, table 2). Alternatively, for backward compatibility, these parameters can also be taken as the mutli-model mean estimates from the Ox-Comp project (Ehhalt et al., 2001, table 4.11), in which case the sensitivities to temperature, humidity and ozone are nil. The preindustrial global atmospheric temperature $T_{A,0}$ is set to 251 K, and the proportionnality coefficients are $\kappa_{T_A} = 0.94$ and $\kappa_{Q_A} = 1.5$ (Holmes et al., 2013). The saturation vapor pressure parameters are obtained from Jacobson (2005, equation 2.62) for which a small temperature perturbation is assumed, giving: $\kappa_{\mathrm{svp}} = 17.67$ and $T_{svp} = -29.65$ K. The preindustrial stratospheric ozone burden $\mathrm{O3s}_0$ is set to 280 DU, roughly following Cionni et al. (2011). The values of $E_{\mathrm{nat}}^{\mathrm{NOx}}$, $E_{\mathrm{nat}}^{\mathrm{CO}}$ and $E_{\mathrm{nat}}^{\mathrm{VOC}}$ are from Skeie et al. (2011, table 1).

### 2.5.2 Wetlands emissions

Natural wetlands are the largest natural source of methane (Ciais et al., 2013b), and the future variation of this source could be significant for future climate change (e.g. O'Connor et al., 2010). We thus decided, since version 2.1 of OSCAR, to include a simple module describing the variation of this source of $CH_4$. The current version is very close to the previous one (Gasser, 2014), except that a larger variety of parameterizations is now available.

First, we estimate the regional change in $CH_4$ emission per unit area of wetlands ($e_{\mathrm{wet}}$) as being proportional to its preindustrial value and to the relative change in heterotrophic respiration of the litter carbon pool in the same region. To this end, wetlands are considered to be a mix of the other biomes, with partition coefficients ($\pi_{\mathrm{wet}}^b$; $\sum_b \pi_{\mathrm{wet}}^b = 1$) having a non-zero value only for natural biomes. We note that this is an *ad hoc* assumption that we make because we lack detailed outputs from complex wetlands models. The litter pool is chosen as a proxy of the changes in wetlands induced by more general changes



in the carbon-cycle. Therefore, here we implicitly assume that the sensitivity of areal wetlands emissions to environmental conditions – e.g. $CO_2$ or temperature – is the same as that of heterotrophic respiration. So we have:

$$\Delta e_{\text{wet}}^i = e_{\text{wet},0}^i \, \frac{\sum_b \pi_{\text{wet}}^{i,b} \, \Delta \text{rh}_F^{i,b}}{\sum_b \pi_{\text{wet}}^{i,b} \, \text{rh}_{F,0}^{i,b}}. \tag{44}$$

Second, we assume that the regional change in wetlands area extent ($A_{\text{wet}}$) depends on linear sensitivities to: atmospheric
$CO_2$ ($\gamma_{\text{wet},C}$), local surface temperatures ($\gamma_{\text{wet},T}$), local yearly precipitations ($\gamma_{\text{wet},P}$). This formulation is similar to that used for wildfire intensity in equation (12), and $CO_2$ is used as a proxy of changes in e.g. evapotranspiration or vegetation species distribution. Mathematically:

$$\Delta A_{\text{wet}}^i =$$
$$A_{\text{wet},0}^i \left( \gamma_{\text{wet},C}^i \, \Delta \text{CO2} + \gamma_{\text{wet},T}^i \, \Delta T_L^i + \gamma_{\text{wet},P}^i \, \Delta P_L^i \right). \tag{45}$$

Consequently, the change in regional emission of methane by wetlands ($E_{\text{wet}}$) is calculated as:

$$\Delta E_{\text{wet}}^i = e_{\text{wet},0}^i \, \Delta A_{\text{wet}}^i + \Delta e_{\text{wet}}^i \, A_{\text{wet},0}^i + \Delta e_{\text{wet}}^i \, \Delta A_{\text{wet}}^i. \tag{46}$$

We calibrate two sets of parameters for wetlands. First, the preindustrial equilibrium of the wetlands can be calibrated on seven WETCHIMP models (Melton et al., 2013). We deduce the $\pi_{\text{wet}}$ parameters by combining the wetlands map from the "exp 1" simulation, that is the equilibrium experiment of the WETCHIMP exercise, and the land-cover map used in section
2.3.2 for natural vegetation. The preindustrial areal emissions $e_{\text{wet},0}$ are also taken from this "exp 1" simulation, but they are scaled down by a factor equal to the ratio of our preindustrial atmospheric $CO_2$ over the one used in WETCHIMP i.e. by a factor of about 0.92, as we did with NPP in section 2.3.2. Second, the parameters for the transient response of wetlands extent (i.e. $\gamma_{\text{wet},C}$, $\gamma_{\text{wet},T}$, $\gamma_{\text{wet},C}$) can be calibrated on six WETCHIMP models (reminder: see appendix B for a list of those models). To do so, we use "exp4", "exp5" and "exp6": factorial simulations that separate the effect of temperature, precipitations and
atmospheric $CO_2$, respectively. For the same reasons as with wildfires, we also keep an option to turn off the preindustrial wetlands flux and/or its transient response.

### 2.5.3 Atmospheric $CH_4$ and RF

On the basis of the previous sections, the incremental change in atmospheric $CH_4$ follows the mass-balance equation:

$$\alpha_{\text{atm}}^{\text{CH4}} \frac{\text{d}}{\text{d}t} \Delta \text{CH4} = E_{\text{CH4}} + \Delta E_{\text{bb}}^{\text{CH4}} + \sum_i \Delta E_{\text{wet}}^i + \Delta F_{\downarrow}^{\text{CH4}}. \tag{47}$$

This implicitly assumes that all the natural sources of methane but natural wetlands remain unchanged since the preindustrial. Here, we also note that the anthropogenic emissions $E_{\text{CH4}}$ do include emissions from rice paddies – i.e. from anthropogenic wetlands.

The radiative forcing induced by the increase in atmospheric $CH_4$ follows a square-root formula to which an *ad hoc* function ($\mathcal{F}_{\text{over}}$) is added to account for the overlap between the absorption bands of methane and nitrous oxide (N2O), following Myhre





et al. (1998). It gives:

$$\Delta \text{RF}^{\text{CH4}} =$$
$$+ \alpha_{\text{rf}}^{\text{CH4}} \sqrt{\text{CH4}_0} \left( \sqrt{1 + \frac{\Delta\text{CH4}}{\text{CH4}_0}} - 1 \right)$$
$$- \left( \mathcal{F}_{\text{over}} \left[ \Delta\text{CH4}, \Delta\text{N2O} \right] - \mathcal{F}_{\text{over}} \left[ \Delta\text{CH4} = 0, \Delta\text{N2O} \right] \right); \qquad (48)$$

where $\alpha_{\text{rf}}^{\text{CH4}} = 0.036$ W m$^{-2}$ ppb$^{-0.5}$ and the analytical expression of $\mathcal{F}_{\text{over}}$ are given by Myhre et al. (2013a, table 8.SM.1). In addition to the RF induced by methane itself, we have to account for the RF induced by the increase in stratospheric water vapor caused by the oxidation of methane. To do so, as others (e.g. Meinshausen et al., 2011), we assume it is equal to 15% of the direct methane RF, but calculated with its lagged concentration:

$$\Delta \text{RF}^{\text{H2Os}} = \alpha_{\text{rf}}^{\text{H2Os}} \sqrt{\text{CH4}_0} \left( \sqrt{1 + \frac{\Delta\text{CH4}_{\text{lag}}}{\text{CH4}_0}} - 1 \right); \qquad (49)$$

where $\alpha_{\text{rf}}^{\text{H2Os}} = 0.15 \times \alpha_{\text{rf}}^{\text{CH4}} = 0.0054$ W m$^{-2}$ ppb$^{-0.5}$. For the preindustrial atmospheric concentration, we take CH4$_0$ = 722 ppb (IPCC, 2013, table AII.1.1a).

## 2.6 Nitrous oxide

### 2.6.1 Stratospheric sink

The oxidation of nitrous oxide follows the same modelling approach as that of methane, with only one sink in the stratosphere that has a varying lifetime. The law used to make the stratospheric lifetime vary, however, is recent and different from the previous version of the model.

The flux of oxidized N$_2$O ($F_{\downarrow}^{\text{N2O}}$) is driven by the preindustrial lifetime of nitrous oxide with regard to stratospheric oxidation ($\tau_{\text{h}\nu}^{\text{N2O}}$). The transient change in this stratospheric lifetime is a function ($\mathcal{F}_{\text{h}\nu}$) of: the lagged N$_2$O concentration (N2O$_{\text{lag}}$); the equivalent effective stratospheric chlorine (EESC; see section 2.8.2); and global surface temperature change ($T_G$). The dependency on N$_2$O and the EESC is meant to model the impact of a change in stratospheric ozone that changes the actinic flux, which in turn changes the stratospheric sink (e.g. Prather, 1998). We have:

$$\alpha_{\text{atm}}^{\text{N2O}-1} \Delta F_{\downarrow}^{\text{N2O}} =$$
$$- \frac{\text{N2O}_0}{\tau_{\text{h}\nu}^{\text{N2O}}} \left( \left( 1 + \frac{\Delta\text{N2O}_{\text{lag}}}{\text{N2O}_0} \right) \mathcal{F}_{\text{h}\nu} \left[ \Delta\text{N2O}_{\text{lag}}, \Delta\text{EESC}, \Delta T_G \right] - 1 \right). \qquad (50)$$

The formulation of $\mathcal{F}_{\text{h}\nu}$ is inspired by that used for methane and the study by Prather et al. (2015). It has three chemical sensitivities ($\xi_{\text{N2O}}^{\text{h}\nu}$, $\xi_{\text{EESC}}^{\text{h}\nu}$ and $\xi_{\text{age}}^{\text{h}\nu}$). This last parameter represent the sensitivity of the sink to a change in stratospheric age of air. This age-of-air change is itself driven by a changing Brewer-Dobson circulation which is induced by a changing climate (e.g. Butchart, 2014). In the following, we consider that the inverse of the relative change in age of air is a linear function of the absolute change in global surface temperature (parameterized by $\gamma_{\text{age}}$; see also figure S23). This leads to the following





formula:

$$\ln[\mathcal{F}_{\mathrm{h}\nu}] =$$
$$+ \xi_{\mathrm{N2O}}^{\mathrm{h}\nu} \ln\left[1 + \frac{\Delta\mathrm{N2O}_{\mathrm{lag}}}{\mathrm{N2O}_0}\right]$$
$$+ \xi_{\mathrm{EESC}}^{\mathrm{h}\nu} \ln\left[1 + \frac{\Delta\mathrm{EESC}}{\mathrm{EESC}_0}\right]$$
$$+ \xi_{\mathrm{age}}^{\mathrm{h}\nu} \ln\left[\frac{1}{1 + \gamma_{\mathrm{age}}\,\Delta T_G}\right]. \tag{51}$$

The preindustrial stratospheric lifetime $\tau_{\mathrm{h}\nu}^{\mathrm{N2O}}$ is taken as 123 years (Prather et al., 2015). As we do with methane, we introduce variation in the N$_2$O lifetime by having the option to rescale the default value by a factor equal to the lifetime simulated by any of the eight models of Prather et al. (2015, table 2) over the multi-model mean estimate. The first two chemical sensitivities of the stratospheric sink (i.e. $\xi_{\mathrm{N2O}}^{\mathrm{h}\nu}$ and $\xi_{\mathrm{EESC}}^{\mathrm{h}\nu}$) are taken as one of the four sets of values from the study by Prather et al. (2015). Three sets of value are given in their table 3, and the fourth is the recommandation in their text. Also, to translate their table 3 into our parameters, we assume that the preindustrial EESC in the models were 420 ppt – from IPCC (2013, table AII.1.1b) and Newman et al. (2007, table 1). Alternatively, for backward compatibility, these parameters can also follow (Prather et al., 2012, table A1), in which case the sensitivity to EESC is zero.

Regarding the chemical sensitivity to the age of air, we assume it is not zero only when the other sensitivities are deduced from the "G2d" model, therefore following the results by Prather et al. (2015, table 3) and their discussion pointing out the experimental aspect of such a parameterization. Nevertheless, in this specific case we need further information about the "G2d" model which we take from Fleming et al. (2011, figure 12) where one can see that the age of air at an altitude of 25 km changed from about 4.5 to 4.0 between the preindustrial and present-day periods. This is enough to deduce the $\xi_{\mathrm{age}}^{\mathrm{h}\nu}$ parameter. And then, the $\gamma_{\mathrm{age}}$ parameter can be calibrated on seven CCMVal2 chemistry-transport models (Morgenstern et al., 2010). To do so, we use outputs from the "REF-B2" experiment which is a fully transient simulation over 1961–2099: we use the "mean_age" output at a pressure-level of 25 hPa (~25 km) and the temperature at the surface level. We then fit the parameter following our inversed linear relationship, defining the preindustrial conditions as the averaged first ten years of the simulations. The CCMVal2 fits are shown in figure S23.

### 2.6.2 Atmospheric N$_2$O and RF

The incremental change in atmospheric N$_2$O follows:

$$\alpha_{\mathrm{atm}}^{\mathrm{N2O}} \frac{\mathrm{d}}{\mathrm{d}t}\Delta\mathrm{N2O} = E_{\mathrm{N2O}} + \Delta E_{\mathrm{bb}}^{\mathrm{N2O}} + \Delta F_{\downarrow}^{\mathrm{N2O}}; \tag{52}$$

noting again that this implicitly assumes natural emissions remain unchanged since the preindustrial.





Similarly to methane, the radiative forcing induced by the increase in atmospheric N$_2$O follows a square-root formula to which the *ad hoc* overlap function is added:

$$\Delta\mathrm{RF}^{\mathrm{N2O}} =$$

$$+ \alpha_{\mathrm{rf}}^{\mathrm{N2O}} \sqrt{\mathrm{N2O}_0} \left( \sqrt{1 + \frac{\Delta\mathrm{N2O}}{\mathrm{N2O}_0}} - 1 \right)$$

$$- \left( \mathcal{F}_{\mathrm{over}} \left[ \Delta\mathrm{CH4}, \Delta\mathrm{N2O} \right] - \mathcal{F}_{\mathrm{over}} \left[ \Delta\mathrm{CH4}, \Delta\mathrm{N2O} = 0 \right] \right); \tag{53}$$

where $\alpha_{\mathrm{rf}}^{\mathrm{N2O}} = 0.12$ W m$^{-2}$ ppb$^{-0.5}$ and $\mathcal{F}_{\mathrm{over}}$ are given by Myhre et al. (2013b, table 8.SM.1). For the preindustrial atmospheric concentration, we take N2O$_0$ = 270 ppb (IPCC, 2013, table AII.1.1a).

## 2.7 Halogenated compounds

OSCAR accounts for many halogenated species. These are grouped into three categories: eleven hydrofluorocarbons (HFC-23, HFC-32, HFC-125, HFC-134a, HFC-143a, HFC-152a, HFC-227ea, HFC-236fa, HFC-245fa, HFC-365mfc, HFC-43-10mee) noted together {HFC}; eight perfluorocarbons (CF$_4$, C$_2$F$_6$, C$_3$F$_8$, c-C$_4$F$_8$, C$_4$F$_{10}$, C$_5$F$_{12}$, C$_6$F$_{14}$, C$_7$F$_{16}$) to which we add SF$_6$ and NF$_3$, and noted together {PFC}; and sixteen ozone depleting substances (CFC-11, CFC-12, CFC-113, CFC-114, CFC-115, CCl$_4$, CH$_3$CCl$_3$, HCFC-22, HCFC-141b, HCFC-142b, Halon-1211, Halon-1202, Halon-1301, Halon-2402, CH$_3$Br, CH$_3$Cl) noted together {ODS}. These are the same as in previous version 2.1.

### 2.7.1 Atmospheric sinks

Conceptually, the modelling approach of the halogenated compounds' sinks is similar to that used for methane. Each of these species X is affected by three sinks, each sink with its specific preindustrial lifetime: a tropospheric oxidation by the hydroxyl radical ($\tau_{\mathrm{OH}}^{\mathrm{X}}$), a stratospheric oxidation ($\tau_{\mathrm{h}\nu}^{\mathrm{X}}$), and another sink which encloses all other processes such as oxidation in dry soils or in the oceanic boundary layer ($\tau_{\mathrm{othr}}^{\mathrm{X}}$). Note that a given oxidation process may not actually affect a given species; in this case the associated lifetime is set to a value of infinity ($\infty$). Mathematically, similarly to equation (40), we have for any species X being a HFC, PFC or ODS:

$$\alpha_{\mathrm{atm}}^{\mathrm{X}}{}^{-1} \Delta F_{\downarrow}^{\mathrm{X}} =$$

$$- \frac{1}{\tau_{\mathrm{OH}}^{\mathrm{X}}} \left( (\Delta\mathrm{X} + \mathrm{X}_0) \, \mathcal{F}_{\mathrm{OH}} \left[ \Delta\mathrm{CH4}, \Delta\mathrm{O3s}, \Delta T_G, \mathcal{F}_{\mathrm{prec}} \right] - \mathrm{X}_0 \right)$$

$$- \frac{1}{\tau_{\mathrm{h}\nu}^{\mathrm{X}}} \left( (\Delta\mathrm{X}_{\mathrm{lag}} + \mathrm{X}_0) \, \mathcal{F}_{\mathrm{h}\nu} \left[ \Delta\mathrm{N2O}_{\mathrm{lag}}, \Delta\mathrm{EESC}, \Delta T_G \right] - \mathrm{X}_0 \right)$$

$$- \frac{1}{\tau_{\mathrm{othr}}^{\mathrm{X}}} \Delta\mathrm{X}; \tag{54}$$

where the functions $\mathcal{F}_{\mathrm{OH}}$, $\mathcal{F}_{\mathrm{prec}}$ and $\mathcal{F}_{\mathrm{h}\nu}$ are the same as in sections 2.5.1 and 2.6.1.

The lifetimes $\tau_{\mathrm{OH}}^{\mathrm{X}}$, $\tau_{\mathrm{h}\nu}^{\mathrm{X}}$ and $\tau_{\mathrm{othr}}^{\mathrm{X}}$ are taken from the compilation by Montzka et al. (2011, table 1-3). However, the lifetimes with respect to the OH sink are rescaled using the same scaling factor as for methane (see section 2.5.1), for consistency. Similarly, the lifetimes with respect to the stratospheric sink are scaled up by a factor 1.06, as done by Prather et al. (2015).



### 2.7.2 Atmospheric concentrations and RFs

The incremental change in atmospheric concentration of any species X being a HFC, PFC or ODS is:

$$\alpha_{\mathrm{atm}}^{X} \frac{d}{dt} \Delta X = E_X + \Delta F_{\downarrow}^{X}. \tag{55}$$

With the exception of $CF_4$, $CH_3Br$ and $CH_3Cl$, all the halogenated compounds are anthropogenic in nature, thus no other

natural fluxes need to be considered. For the three former species, however, their natural emissions are assumed to remain constant through time.

The radiative forcing induced by the increase in atmospheric concentration of any of those species X is assumed to be propotionnal:

$$\Delta RF^{X} = \alpha_{\mathrm{rf}}^{X} \Delta X; \tag{56}$$

where the values of $\alpha_{\mathrm{rf}}^{X}$ are taken from Myhre et al. (2013b, table 8.A.1). In the following, all these RFs will be combined into one:

$$\Delta RF^{\mathrm{halo}} = \sum_{\substack{X \in \{HFC\} \cup \\ \{PFC\} \cup \{OFC\}}} \Delta RF^{X}. \tag{57}$$

Finally, only the three species cited hereabove have non-zero preindustrial atmospheric concentration: $CF4_0$ = 35 ppt (IPCC, 2013, table AII.1.1a), $CH3Br_0$ = 5.8 ppt and $CH3Cl_0$ = 480 ppt (Meinshausen et al., 2011).

## 2.8 Ozone

### 2.8.1 Tropospheric $O_3$ and RF

In OSCAR, as it is common in simple models (e.g. Meinshausen et al., 2011), short-lived species are not predicted using a dynamic model like long-lived species are. Rather, at each time-step, the short-lived species are supposed to be at chemical equilibrium with their drivers of change. For tropospheric ozone, we use a formulation close to that of the previous version of

OSCAR, which was the formulation by Ehhalt et al. (2001). In version 2.2, however, the chemical sensitivities are updated and regionalized, and a sensitivitiy to climate change is added.

The change in global tropospheric ozone burden (O3t) is a function of: atmospheric methane, with a logarithmic sensitivity ($\xi_{CH_4}^{O3t}$); global surface temperature, with a linear sensitivity ($\Gamma_{O3t}$); and the three ozone precursors, with linear global sensitivities ($\xi_{NOx}^{O3t}$, $\xi_{CO}^{O3t}$, $\xi_{VOC}^{O3t}$) that are regionalized thanks to region-specific weights ($\omega_{NOx}$, $\omega_{CO}$, $\omega_{VOC}$). Here, we introduce a

new regional axis (superscript $^r$) that is *de facto* different from the biospheric one (superscript $^i$). The regional axes are linked through parameters describing what fraction of a region $i$ is actually included in a region $r$ ($\pi_{\mathrm{reg}}^{r,i}$; $\sum_r \pi_{\mathrm{reg}}^{r,i} = 1$). So we finally

have:

$$\Delta O3t =$$

$$+ \xi_{CH4}^{O3t} \ln\left[1 + \frac{\Delta CH4}{CH4_0}\right]$$

$$+ \Gamma_{O3t}\,\Delta T_G$$

$$+ \sum_{\substack{X\in\{NOx,\\ CO,VOC\}}} \xi_X^{O3t} \sum_r \omega_X^r \sum_i \pi_{reg}^{r,i} \left(E_X^i + \Delta E_{bb}^{X,i}\right). \tag{58}$$

The global chemical sensitivities (i.e. $\xi_{CH4}^{O3t}$, $\xi_{NOx}^{O3t}$, $\xi_{CO}^{O3t}$, $\xi_{VOC}^{O3t}$) can be calibrated on four ACCMIP chemistry-transport models (Stevenson et al., 2013). To do so, we use their reference simulations for the year 2000, as well as the factorial simulations which were made so as to isolate each of the four drivers of the change in tropospheric ozone (namely "1850CH4", "1850NOx", "1850CO" and "1850NMVOC"). However, since we also have access to a simulation in which all of the four drivers vary at

the same time – i.e. the difference between the experiments for 1850s and the 2000s – we can estimate the non-linearity of this chemical system. We account for this non-linearity by rescaling the individual sensitivities by a factor equal to the ratio of the ozone change in the all-varying simulation over the sum of the ozone changes in each of the factorial simulations. The sensitivity to global climate change $\Gamma_{O3t}$ can be calibrated on eight models which participated to the same ACCMIP exercise and made simulations in which only climate varies. A simple linear fit is made over these simulations; and we also keep an

option to set this sensitivity to zero. The latter ACCMIP fits are shown in figure S24.

The regional weights $\omega_X$ can be deduced from the results of eleven HTAP chemistry-transport models (Fiore et al., 2009). To do so, we calculate regional ozone changes in the four HTAP regions thanks to table S5 from Fry et al. (2012) and regional precursors emissions thanks to table S1 from Fiore et al. (2009). Our weighting parameters are then deduced as the ratio of the regional ozone changes normalized by the precursors changes over the globally averaged normalized ozone change. A fifth

region is then added, to account for areas of the globe that are not within the four HTAP regions, and for which the weighting parameter is set to exactly 1. The $\pi_{reg}$ parameters are logically defined as the fraction of the area of a region $i$ that is inside a region $r$. Also, we keep an option to turn off that regionalization, i.e. setting all regional weights to 1.

Finally, the radiative forcing induced by the change in tropospheric ozone burden is assumed to be linear:

$$\Delta RF^{O3t} = \alpha_{rf}^{O3t}\,\Delta O3t; \tag{59}$$

where the value of $\alpha_{rf}^{O3t}$ is not unique – contrarily to what we do with greenhouse gases. This radiative efficiency can be: 0.042 W m$^{-2}$ DU$^{-1}$, as reported by (Myhre et al., 2013b); 0.032 W m$^{-2}$ DU$^{-1}$, as reported by (Forster et al., 2007); or one of the fifteen radiative efficiencies given by the ACCMIP chemistry-transport models (Stevenson et al., 2013, table 3).





### 2.8.2 Stratospheric O$_3$ and RF

In OSCAR, as in other simple models (e.g. Meinshausen et al., 2011), stratospheric ozone is estimated apart from tropospheric ozone. As with tropospheric ozone, we assume that each year stratospheric ozone is at equibrium with its drivers of change. Compared to the previous version, this module now has two additional drivers: nitrous oxide and climate change.

The first step to model stratospheric ozone is to estimate its first driver of change: the stratospheric chlorine and bromine available from the presence of the ODSs in the stratosphere. Those compounds release their chorine and/or bromine atoms at various rates and thus interact differently with ozone. A proxy variable is thus created to lump together these various effects, namely the equivalent effective stratospheric chlorine (EESC). The EESC is calculated following Newman et al. (2007), on the basis of: the fractional release of each ODS ($\pi_{\text{rel}}$); its numbers of chlorine atoms ($n_{\text{Cl}}$) and bromine atoms ($n_{\text{Br}}$); a parameter

measuring the efficiency in destroying ozone of bromine relative to that of chlorine ($\alpha_{\text{Cl}}^{\text{Br}}$); and the lagged concentration of the ODS. That is:

$$\Delta \text{EESC} = \sum_{\text{X} \in \{\text{ODS}\}} \pi_{\text{rel}}^{\text{X}} \left( n_{\text{Cl}}^{\text{X}} + \alpha_{\text{Cl}}^{\text{Br}} n_{\text{Br}}^{\text{X}} \right) \Delta \text{X}_{\text{lag}}. \tag{60}$$

Then, a change in stratospheric ozone burden (O3s) is assumed to happen with a change in EESC, with a linear sensitivity ($\xi_{\text{EESC}}^{\text{O3s}}$). To the effect of ODSs, we add the effect of nitrous oxide following the simple formulation by Daniel et al. (2010)

which needs two additional parameters: one to quantify the linear sensitivity of stratospheric ozone to nitrous oxide ($\xi_{\text{N2O}}^{\text{O3s}}$), and one to account for the non-linear interaction between chlorine and nitrogen chemistries (EESC$_\times$). As per tropospheric ozone, a linear sensitivity to global surface temperature change ($\Gamma_{\text{O3s}}$) is also added, which sums up to:

$$\Delta \text{O3s} =$$
$$+ \xi_{\text{EESC}}^{\text{O3s}} \Delta \text{EESC}$$
$$+ \xi_{\text{N2O}}^{\text{O3s}} \left( 1 - \frac{\Delta \text{EESC}}{\text{EESC}_\times} \right) \Delta \text{N2O}_{\text{lag}}$$
$$+ \Gamma_{\text{O3s}} \Delta T_G. \tag{61}$$

Regarding the EESC parameterization, Newman et al. (2006, tables A1 & A2) provide values of fractional release $\pi_{\text{rel}}$ for all our ODSs, assuming a mean age-of-air of 3 years taken equal to the time-lag of section 2.4.2. To introduce other possibilities of parameterization in the model, we can alternatively take fractional release values from Laube et al. (2013), either the values for

the mid-latitudes or those for the high latitudes. In this case, if a value is missing for a given ODS we take that from Newman et al. (2006). The chemical formula of each ODS gives $n_{\text{Cl}}$ and $n_{\text{Br}}$. And we take $\alpha_{\text{Cl}}^{\text{Br}} = 60$ (Newman et al., 2007).

The chemical sensitivity of stratospheric ozone to EESC and that to global climate change (i.e. $\xi_{\text{EESC}}^{\text{O3s}}$ and $\Gamma_{\text{O3s}}$) can be calibrated on eleven CCMVal2 chemistry-transport models studied by Douglass et al. (2014), using the results from their multi-linear regression. The sensitivity to nitrous oxide is calculated using the formula by Daniel et al. (2010): $\xi_{\text{N2O}}^{\text{O3s}} =$

$\xi_{\text{EESC}}^{\text{O3s}} \alpha_{\text{N2O}}^{\text{EESC}} \pi_{\text{rel}}^{\text{CFC11}}$; where $\alpha_{\text{N2O}}^{\text{EESC}}$ is a parameter measuring the relative strength importance of N$_2$O and chlorine. Values for the parameters are given by Daniel et al. (2010) and based on Ravishankara et al. (2009): $\alpha_{\text{N2O}}^{\text{EESC}} \simeq 10.4$ ppt ppb$^{-1}$ and EESC$_\times \simeq 2642$ ppt. Also, we keep an option to turn off this response of stratospheric ozone to nitrous oxide.



Finally, the radiative forcing induced by the change in stratospheric ozone burden is assumed to be linear:

$$\Delta \mathrm{RF}^{\mathrm{O3s}} = \alpha_{\mathrm{rf}}^{\mathrm{O3s}} \, \Delta \mathrm{O3s}; \tag{62}$$

where the radiative efficiency $\alpha_{\mathrm{rf}}^{\mathrm{O3s}}$ can be: 0.004 W m$^{-2}$ DU$^{-1}$, as reported by (Forster et al., 2007); or one of the four radiative efficiencies given by the ACCENT models (Gauss et al., 2006, tables 4 & 6).

## 2.9  Aerosols

### 2.9.1  Direct effect

The direct effect of aerosols refers to the direct radiative forcing caused by the aerosol-radiation interactions, i.e. without consideration of any short-term adjustment of the climate system (Boucher et al., 2013). This section describes how tropospheric burdens and the resulting RF of five anthropogenic aerosols, namely sulphate aerosols, primary organic aerosols, black carbon, nitrate aerosols and secondary organic aerosols, are calculated within our model. Because these aerosols are short-lived, it is assumed that their global atmospheric burden is in equilibrium with their respective drivers of change at each time-step of the model.

It must be noted that here we purposefully limit the number of these drivers of change: only two precursors are considered for each aerosol, to avoid overfitting on data that does not allow us to clearly separate the effect of each precursor; and we add the global surface temperature, used as a proxy of a changing climate. For the same reason – because of the calibration data – we keep the modelling simple with linear sensitivities. Note also that in this section every lifetime is said "apparent", because it corresponds to a globally averaged chemical sensitivity that has dimensions of time, and which results from several physical and/or chemical processes not explicitly modelled in OSCAR.

In the case of sulphate aerosols, their change in burden (SO4) is parameterized by the apparent lifetime of sulfur dioxide ($\tau_{\mathrm{SO2}}$) – with a regionalized weighting ($\omega_{\mathrm{SO2}}$) analogous to that used for tropospheric ozone in section 2.8.1 – the apparent lifetime of dimethyl sulfide ($\tau_{\mathrm{DMS}}$), and their sensitivity to global surface temperature ($\Gamma_{\mathrm{SO4}}$). So we have:

$$\Delta \mathrm{SO4} =$$
$$+ \tau_{\mathrm{SO2}} \sum_r \omega_{\mathrm{SO2}}^r \sum_i \pi_{\mathrm{reg}}^{r,i} \left( E_{\mathrm{SO2}}^i + \Delta E_{\mathrm{bb}}^{\mathrm{SO2},i} \right)$$
$$+ \tau_{\mathrm{DMS}} \, \Delta E_{\mathrm{DMS}}$$
$$+ \Gamma_{\mathrm{SO4}} \, \Delta T_G. \tag{63}$$

The change in burden of primary organic aerosols (POA) is parameterized by the apparent lifetime of fossil-based organic matter ($\tau_{\mathrm{OM,ff}}$) – also regionally weighted ($\omega_{\mathrm{OM}}$) – the apparent lifetime of pyrogenic organic matter ($\tau_{\mathrm{OM,bb}}$), their sensitivity





to global surface temperature ($\Gamma_{\text{POA}}$), as well as a factor used to convert organic carbon to organic matter ($\alpha_{\text{OC}}^{\text{OM}}$):

$$\Delta\text{POA} =$$
$$+ \tau_{\text{OM,ff}}\, \alpha_{\text{OC}}^{\text{OM}} \sum_r \omega_{\text{OM}}^r \sum_i \pi_{\text{reg}}^{r,i}\, E_{\text{OC}}^i$$
$$+ \tau_{\text{OM,bb}}\, \alpha_{\text{OM}}^{\text{OC}} \sum_i \Delta E_{\text{bb}}^{\text{OC},i}$$

$$+ \Gamma_{\text{POA}}\, \Delta\text{GST}. \tag{64}$$

The change in burden of black carbon (BC) is parameterized by the apparent lifetime of fossil-based black carbon ($\tau_{\text{BC,ff}}$) – also regionally weighted ($\omega_{\text{BC}}$) – the apparent lifetime of pyrogenic black carbon ($\tau_{\text{BC,bb}}$), and their sensitivity to global surface temperature ($\Gamma_{\text{BC}}$):

$$\Delta\text{BC} =$$
$$+ \tau_{\text{BC,ff}} \sum_r \omega_{\text{BC}}^r \sum_i \pi_{\text{reg}}^{r,i}\, E_{\text{BC}}^i$$
$$+ \tau_{\text{BC,bb}} \sum_i \Delta E_{\text{bb}}^{\text{BC},i}$$
$$+ \Gamma_{\text{BC}}\, \Delta T_G. \tag{65}$$

In the case of nitrate aerosols, inspired by Shindell et al. (2009), we assume their formation is driven by nitrogen oxides and ammonia emissions, and therefore we uncouple the nitrate and sulphate chemistries while they are coupled in reality (Boucher
et al., 2013). Hence, the change in burden of nitrate aerosols (NO3) is parameterized by the apparent lifetime of nitrogen oxides ($\tau_{\text{NOx}}$), the apparent lifetime of ammonia ($\tau_{\text{NH3}}$), and their sensitivity to global surface temperature ($\Gamma_{\text{NO3}}$). So we have:

$$\Delta\text{NO3} =$$
$$+ \tau_{\text{NOx}} \left( E_{\text{NOx}} + \sum_i \Delta E_{\text{bb}}^{\text{NOx},i} \right)$$
$$+ \tau_{\text{NH3}} \left( E_{\text{NH3}} + \sum_i \Delta E_{\text{bb}}^{\text{NH3},i} \right)$$
$$+ \Gamma_{\text{NO3}}\, \Delta T_G. \tag{66}$$

And finally, the change in burden of secondary organic aerosols (SOA) is parameterized by the apparent lifetime of anthropogenic NMVOCs ($\tau_{\text{VOC}}$), the apparent lifetime of biogenic NMVOCs ($\tau_{\text{BVOC}}$), and their sensitivity to global surface temperature ($\Gamma_{\text{SOA}}$). Here, the dependency of SOA on other factors such as atmospheric $\text{NO}_x$ or POA (Boucher et al., 2013) is neglected. So we have:

$$\Delta\text{SOA} =$$
$$+ \tau_{\text{VOC}} \left( E_{\text{VOC}} + \sum_i \Delta E_{\text{bb}}^{\text{VOC},i} \right)$$
$$+ \tau_{\text{BVOC}}\, \Delta E_{\text{BVOC}}$$
$$+ \Gamma_{\text{SOA}}\, \Delta T_G. \tag{67}$$



Finally, here it must be noted that, despite being used for the calibration (see below) and being shown in equations (63) and (67), DMS and BVOC emissions are constant in this version of OSCAR. In other words, in any simulation with OSCAR v2.2 we have $\Delta E_{\mathrm{DMS}} = 0$ and $\Delta E_{\mathrm{BVOC}} = 0$. Also in this version, we do not model any change in natural aerosols, i.e. in mineral dust and sea salt.

For SO$_4$, POA and BC, the apparent global lifetimes $\tau_X$ and the climate sensitivities $\Gamma_X$ can be calibrated on four CMIP5 or ACCMIP chemistry-climate models. To do so, we use the yearly outputs from the historical and RCPs simulations, assuming the average of the first ten years is our preindustrial equilibrium. We then fit the parameters on the basis of equation (63), (64) or (65), and over all the simulations at the same time. For SOA, it is done in the same way, except that only two models are available. Additionnally, because of our very low confidence in the SOA modelling, we also keep an option to turn it off.

For NO$_3$ we use other simulations and models: we do the exact same fit with the input and output data from either Bellouin et al. (2011) or Hauglustaine et al. (2014). In the latter case, $\Gamma_{\mathrm{NO3}}$ is set to zero because climate does not vary in the available simulations. The conversion factor $\alpha_{\mathrm{OC}}^{\mathrm{OM}}$ – which is the same here for fossil-based and biomass burning emissions – can take three values: a default and widely used value of 1.4; 1.3 (Koch et al., 2009); or 1.6 (Rotstayn et al., 2012). The CMIP5/ACCMIP fits are shown in figures S25 to S28.

The regional weights $\omega_X$ can be deduced from the results of seven HTAP chemistry-transport models (Yu et al., 2013). To do so, for the four HTAP regions, we take the normalized aerosol-induced RF data from the detail of their table 6. Our weighting parameters are then deduced as the ratio of the regional normalized RF over the globally averaged normalized RF. A fifth region is then added, to account for areas of the globe that are not within the four HTAP regions, and for which the weighting parameter is set to exactly 1. The $\pi_{\mathrm{reg}}$ parameters are the same as in section 2.8.1. Also, we keep an option to turn off that 20 regionalization, i.e. setting all regional weights to 1.

    For any of the five aerosols X described in this section, the direct radiative forcing induced by a change in atmospheric burden is assumed to be linear:

$$\Delta \mathrm{RF}^X = \alpha_{\mathrm{rf}}^X \, \Delta X; \tag{68}$$

where the radiative efficiencies $\alpha_{\mathrm{rf}}^X$ are taken from the AeroCom II intercomparison (Myhre et al., 2013a). This leads to fifteen 25 possible parameters for SO$_4$ (their table 4), fifteen for POA (their table 6), fifteen for BC (their table 5), eight for NO$_3$ (their table 8) and five for SOA (their table 7).

### 2.9.2 Cloud effects

Under this term, we group the so-called semi-direct and indirect effects – that is, the rapid adjustements in the atmospheric system induced by aerosol-radiation interactions and the adjusted aerosol-cloud interactions, according to the terminology by 30 Boucher et al. (2013) (see also Sherwood et al., 2015). The formulation we propose here is new to the model.

    For the semi-direct effect, the modelling approach is straightforward. According to Boucher et al. (2013) this effect can largely be attributed to absorbing aerosols, i.e. to BC in our model. We thus account for this effect simply by adding a RF term that is proportional (by a factor $\kappa_{\mathrm{adj}}^{\mathrm{BC}}$) to the direct RF of BC. For the aerosol-cloud interactions, the modelling is done in two



steps. First, we estimate the change in tropospheric burden of soluble aerosols ($\mathrm{AER_{sol}}$) thanks to soluble fractions specific to each type of anthropogenic and natural aerosol ($\pi_{\mathrm{sol}}^{\mathrm{X}}$). It gives:

$$\Delta \mathrm{AER_{sol}} = \sum_{\substack{\mathrm{X} \in \{\mathrm{SO4, POA,} \\ \mathrm{BC, NO3, SOA}\}}} \pi_{\mathrm{sol}}^{\mathrm{X}} \, \Delta \mathrm{X} \tag{69}$$

Second, inspired by several studies (Boucher and Pham, 2002; Hansen et al., 2005; Carslaw et al., 2013; Stevens, 2015), we

assume the aerosol-cloud interaction effective RF varies with the logarithm of the change in this soluble aerosols burden, parameterized by the intensity of the effect ($\Phi$) and the preindustrial soluble aerosols burden. This logarithmic functional form represents a saturating yet not bounded capacity of the emitted hydrophilic aerosols to alter the clouds' albedo (see e.g. Carslaw et al., 2013, figure 3). Finally, in OSCAR the RF of the two combined cloud effects is therefore formulated as:

$$\Delta \mathrm{RF^{cloud}} = \kappa_{\mathrm{adj}}^{\mathrm{BC}} \, \Delta \mathrm{RF^{BC}} + \Phi \ln\left[1 + \frac{\Delta \mathrm{AER_{sol}}}{\mathrm{AER_{sol,0}}}\right]. \tag{70}$$

One possible value for the coefficient used to account for the semi-direct effect is based on the fifth IPCC report (Boucher et al., 2013): $\kappa_{\mathrm{adj}}^{\mathrm{BC}}$ = –0.1/0.6. However, so as to introduce variation around this effect, we also add parameterizations based on the study by Lohmann et al. (2010). Using data from their figure 2, we multiply the IPCC-based value by one of the five models' estimate of the effect and divide it by the multi-model mean estimate, thus obtaining five alternative parameterizations.

The derivation of the parameters for the aerosol-cloud interaction is done in three steps. First, we need the soluble aerosol

fractions $\pi_{\mathrm{sol}}^{\mathrm{X}}$: they are taken either from the study by Hansen et al. (2005) or from that by Lamarque et al. (2011). When taken from Hansen et al. (2005, section 3.3.1), we assume that the soluble fraction of BC is a mix in equal shares of that of fossil BC and biomass burning BC, SOA has the same solubble fraction as POA, and mineral dust – not modelled by OSCAR but necessary here to deduce $\mathrm{AER_{sol,0}}$ – has a soluble fraction of zero. When taken from Lamarque et al. (2011), all soluble fractions are equal to one, except for POA and BC whose solubility is taken as the percentage of hydrophilic aerosol provided

by the study, and for mineral dust and sea salt whose solubility is taken as the percentage of aerosol with a diameter <1 µm.

Second, we calculate the intensity parameter $\Phi$ and a preliminary value of $\mathrm{AER_{sol,0}}$ using results from ACCMIP and CMIP5 models presented by Shindell et al. (2013). Using their table 7, we can base our parameters on one of seven ACCMIP/CMIP5 estimates of the indirect aerosol RF over 1850–2000, or on their multi-model mean. However, because these estimates are far from the IPCC best guess (Boucher et al., 2013), the chosen ACCMIP/CMIP5 value is rescaled by a factor equal to the IPCC

best guess divided by the multi-model mean. We then extract from the ACCMIP or CMIP5 outputs the atmospheric burden of each aerosol type simulated by the chosen model. These burdens are then combined using our own solubility fractions to calculate the soluble aerosols burden in the years 1850 and 2000. These two points in time, combined with the previously rescaled RF estimate, are enough to deduce $\Phi$ through the logarithmic formula. The soluble aerosols burden in 1850 is our preliminary value of $\mathrm{AER_{sol,0}}$.

Third, because this preliminary value of $\mathrm{AER_{sol,0}}$ is for the year 1850 and not the year 1750, we rescale it by a given factor from the study by Carslaw et al. (2013) and adapted to the logarithmic formula; its value is $\exp[(1.42 - 1.30)/\Phi]$ and it is named the "median" option. Again, in order to introduce variation in our modelling of the indirect effect, we also propose two





other arbitrary rescaling options: one with actually no rescale, named "high"; and one with the rescale factor applied twice, named "low". With these three steps, we expect to introduce enough variation for the model to cover a wide range of possible future evolution of the aerosol-cloud interactions, i.e. to span a large domain of the figure 3 of Carslaw et al. (2013) as it is illustrated in our figure 5.

## 2.10 Surface albedo

Anthropogenic perturbations of the Earth's energy budget through surface albedo change are difficult to model in a simple way, because they are local phenomena with significant seasonal variability. Moreover, they can involve non-radiative processes that are almost impossible to capture with simple models. The two OSCAR modules presented hereafter are first order models of two surface albedo perturbations: black carbon deposition on snow, and land-cover change. As such, they are not coupled with one another, nor are they with the climate module.

### 2.10.1 Black carbon on snow

The radiative forcing induced by BC deposition on snow is taken directly propotionnal to the regional BC emissions. It is parameterized by a global radiative efficiency with respect to emissions ($\alpha_{\text{rf}}^{\text{BCsnow}}$), and further regionalized by region-specific weights ($\omega_{\text{BCsnow}}$). Mathematically:

$$\Delta\text{RF}^{\text{BCsnow}} =$$
$$\alpha_{\text{rf}}^{\text{BCsnow}} \sum_{r'} \omega_{\text{BCsnow}}^{r'} \sum_i \pi_{\text{reg}}^{r',i} \left( E_{\text{BC}}^i + \Delta E_{\text{bb}}^{\text{BC},i} \right); \tag{71}$$

where the regionalization (superscript $r'$) is specific to this module, and therefore different from the regionalization based on HTAP seen in previous atmospheric chemistry modules (sections 2.8.1 and 2.9.1).

The global radiative efficiency with respect to emissions $\alpha_{\text{rf}}^{\text{BCsnow}}$ can be taken from eight ACCMIP models (Lee et al., 2013, table 3 & figure 15). The regional weights $\omega_{\text{BCsnow}}$ are obtained from the study by Reddy and Boucher (2007, table 1). As in section 2.9.1, the weighting parameters are deduced as the ratio of the regional radiative efficiencies over the globally averaged radiative efficiency. And a tenth region is added, to account for areas of the globe that are not within the nine regions of Reddy and Boucher (2007), and for which the weighting parameter is set to exactly 1. The $\pi_{\text{reg}}$ parameters are logically defined as the fraction of the area of a region $i$ that is inside a region $r'$.

### 2.10.2 Land-cover change

The radiative forcing induced by changes in land-cover is modelled following the first order equation of Bright and Kvalevåg (2013). It is parameterized by: the yearly averaged albedo at the biome and regional scale ($\alpha_{\text{alb}}$); the regional radiative short-wave and downward flux at the surface ($\varphi_{\text{rsds}}$); and the global short-wave and upward transmittance ($\pi_{\text{trans}}$). Here we note that both the drivers and the regional disaggregation are the same as those of the terrestrial carbon-cycle, which implies the $i$-axis





is the same as in sections 2.3.2 and 2.3.3. So we have:

$$\Delta RF^{LCC} = -\pi_{trans} \sum_i \varphi^i_{rsds} \sum_b \alpha^{i,b}_{alb} \frac{\Delta A^{i,b}}{A_{Earth}};$$ (72)

where $A_{Earth}$ designates the surface area of the Earth.

The upward transmittance is set to $\pi_{trans} = 0.854$ (Lenton and Vaughan, 2009). The radiation fluxes $\varphi_{rsds}$ are taken from
one of three climatologies: GEWEX (2010) over the 1984–2007 period, CERES (2015) over 2000–2014, or MERRA (2015)
over 1979–2014. The albedoes $\alpha_{alb}$ are based on one of two climatologies: either GlobAlbedo (Muller et al., 2012) over the
1998–2011 period, or MODIS (LPDAAC, 2011) over 2001–2010. We calculate the yearly averaged biome-specific albedoes by
weighting the albedo climatology by one of two land-cover climatologies – either MODIS (Channan et al., 2014) or ESA-CCI
(2015) – and by the radiation climatology used for $\varphi_{rsds}$, in a similar fashion as He et al. (2014) do. This approach ensures that
the yearly averaged albedo accounts for the local seasonality, and especially that of snow-cover. Also, regarding the deduction
of biome-specific albedoes, three more assumptions are made: we apply the same weighting method of the land-cover fraction
as in section 2.3.2; we remove the gridcells that see less than 1% of their area changing over the historical period and the RCPs
according to our LULCC dataset (Hurtt et al., 2011); and pastures are assumed to be made at 60% of grasslands and 40% of
bare soils.

## 2.11 Climate

### 2.11.1 Radiative forcings

The first step to calculate global warming is to calculate global radiative forcing. So as to ease the notations, following Myhre
et al. (2013b), we introduce two groups of anthropogenic forcings: the well-mixed greenhouse gases (WMGHGs) which radia-
tive forcing is defined as:

$$\Delta RF^{WMGHG} = \sum_{\substack{X \in \{CO2, \\ CH4, N2O\}}} \Delta RF^X + \Delta RF^{halo};$$ (73)

and the near-term climate forcers (NTCFs) which radiative forcing is defined as:

$$\Delta RF^{NTCF} = \sum_{\substack{X \in \{H2Os, O3t, \\ O3s, SO4, POA, \\ BC, NO3, SOA\}}} \Delta RF^X + \Delta RF^{cloud}.$$ (74)

Then, the global radiative forcing easily comes as:

$$\Delta RF =$$
$$+ \Delta RF^{WMGHG} + \Delta RF^{NTCF} + \Delta RF^{BCsnow} + \Delta RF^{LCC}$$
$$+ RF_{con} + RF_{volc} + RF_{solar};$$ (75)

where the last three terms are the three drivers directly prescribed to OSCAR as radiative forcing and detailed in section 2.2.3.



To estimate global warming, however, we have to account for the so-called "efficacy" of these forcings, i.e. we have to introduce new parameters ($\kappa_{\mathrm{warm}}^{\mathrm{X}}$) that measure the relative efficiency at warming the Earth of a given RF when compared to the RF of $CO_2$ (see e.g. Hansen et al., 2005; Forster et al., 2007). In OSCAR, we assume all efficacies are equal to 1 – although accounting for the semi-direct effect of BC could be defined as using an efficacy – except for the two surface albedo forcings

and for volcanic aerosols. Therefore, the RF used to calculate warming ($\mathrm{RF}_{\mathrm{warm}}$) is:

$$\Delta \mathrm{RF}_{\mathrm{warm}} =$$
$$+\, \Delta \mathrm{RF}^{\mathrm{WMGHG}} + \Delta \mathrm{RF}^{\mathrm{NTCF}} + \mathrm{RF}_{\mathrm{con}} + \mathrm{RF}_{\mathrm{solar}}$$
$$+\, \kappa_{\mathrm{warm}}^{\mathrm{BCsnow}} \Delta \mathrm{RF}^{\mathrm{BCsnow}} + \kappa_{\mathrm{warm}}^{\mathrm{LCC}} \Delta \mathrm{RF}^{\mathrm{LCC}} + \kappa_{\mathrm{warm}}^{\mathrm{volc}} \mathrm{RF}_{\mathrm{volc}}. \tag{76}$$

Here, $\kappa_{\mathrm{warm}}^{\mathrm{BCsnow}}$ can take three values: a median value of 3.0, a low value of 2.0, and a high value of 4.0, all from Boucher et al.

(2013, section 7.5.2.3); $\kappa_{\mathrm{warm}}^{\mathrm{LCC}}$ can take one of the four values given by Bright et al. (2015, table 7); and $\kappa_{\mathrm{warm}}^{\mathrm{volc}}$ is set to an arbitrary value of 0.6 based on Gregory et al. (2016). However, regarding volcanic aerosols, we note that since the forcing is normalized to zero over the historical period in section 2.2.3, its efficacy only influences the variability of our results and not the trend.

Now, to estimate global precipitations change, we also need to estimate how much of this top-of-the-atmosphere RF is

actually occuring within the atmosphere – thus creating a local energy imbalance – by opposition to the RF occuring at the Earth's surface. To do so, we introduce new parameters that quantify this atmospheric fraction for several groups of forcers: carbon dioxide alone ($\pi_{\mathrm{atm}}^{\mathrm{CO2}}$); all the other long-lived greenhouse gases, i.e. methane, nitrous oxide and the halogenated compounds ($\pi_{\mathrm{atm}}^{\mathrm{noCO2}}$); tropospheric ozone alone ($\pi_{\mathrm{atm}}^{\mathrm{O3t}}$); stratospheric greenhouse gases, i.e. stratospheric water vapor and ozone ($\pi_{\mathrm{atm}}^{\mathrm{strat}}$); scattering aerosols, i.e. sulphate, primary organic, nitrate, secondary organic and volcanic aerosols ($\pi_{\mathrm{atm}}^{\mathrm{scatter}}$); absorb-

ing aerosols, i.e. black carbon ($\pi_{\mathrm{atm}}^{\mathrm{absorb}}$); cloud-related forcings ($\pi_{\mathrm{atm}}^{\mathrm{cloud}}$); forcings from surface albedo change ($\pi_{\mathrm{atm}}^{\mathrm{alb}}$); and the solar forcing ($\pi_{\mathrm{atm}}^{\mathrm{solar}}$). The atmospheric radiative forcing ($\mathrm{RF}_{\mathrm{atm}}$) consequently is:

$$\Delta \mathrm{RF}_{\mathrm{atm}} =$$
$$+\pi_{\mathrm{atm}}^{\mathrm{CO2}} \Delta \mathrm{RF}^{\mathrm{CO2}} + \pi_{\mathrm{atm}}^{\mathrm{noCO2}} \left( \Delta \mathrm{RF}^{\mathrm{CH4}} + \Delta \mathrm{RF}^{\mathrm{N2O}} + \Delta \mathrm{RF}^{\mathrm{halo}} \right)$$
$$+\pi_{\mathrm{atm}}^{\mathrm{O3t}} \Delta \mathrm{RF}^{\mathrm{O3t}} + \pi_{\mathrm{atm}}^{\mathrm{strat}} \left( \Delta \mathrm{RF}^{\mathrm{H2Os}} + \Delta \mathrm{RF}^{\mathrm{O3s}} \right)$$
$$+\pi_{\mathrm{atm}}^{\mathrm{scatter}} \left( \Delta \mathrm{RF}^{\mathrm{SO4}} + \Delta \mathrm{RF}^{\mathrm{POA}} + \Delta \mathrm{RF}^{\mathrm{NO3}} + \Delta \mathrm{RF}^{\mathrm{SOA}} + \mathrm{RF}_{\mathrm{volc}} \right)$$
$$+\pi_{\mathrm{atm}}^{\mathrm{absorb}} \Delta \mathrm{RF}^{\mathrm{BC}} + \pi_{\mathrm{atm}}^{\mathrm{cloud}} \left( \Delta \mathrm{RF}^{\mathrm{cloud}} + \mathrm{RF}_{\mathrm{con}} \right)$$
$$+\pi_{\mathrm{atm}}^{\mathrm{alb}} \left( \Delta \mathrm{RF}^{\mathrm{BCsnow}} + \Delta \mathrm{RF}^{\mathrm{LCC}} \right) + \pi_{\mathrm{atm}}^{\mathrm{solar}} \mathrm{RF}_{\mathrm{solar}}. \tag{77}$$

We base our grouping of the forcers on Allan et al. (2013). This grouping assumes that the atmospheric fraction $\pi_{\mathrm{atm}}$ of $CH_4$ applies for all non-$CO_2$ long-lived greenhouse gases and that of $SO_4$ applies for all scattering aerosols. Additionally, we assume

that cloud, albedo-based and stratospheric forcers have a nil atmospheric fraction. Other than that, the atmospheric fractions are taken from Andrews et al. (2010, table 3) or Kvalevåg et al. (2013, table 2, case of highest perturbation), although in the latter case tropospheric ozone is also given a nil fraction.





### 2.11.2 Surface temperatures

Similarly to what is done in other simple models – e.g. Raupach et al. (2011), but not MAGICC (Meinshausen et al., 2011) – in OSCAR, the global surface temperature change is based on an impulse response function (IRF) calibrated on more complex global circulation models. The impulse response function, however, is hereby coded as a two-box model, but theoretically speaking it is strictly equivalent (see Geoffroy et al., 2013). And for regional temperatures, we use a simple linear approach – equivalent to pattern scaling.

The two-box model used to model the global surface temperature change has two state variables: the global surface temperature itself ($T_G$), and the temperature of the deep ocean ($T_D$). It is parameterized by: the climate sensitivity ($\lambda$); the time-inertia of the surface box ($\tau_{T_G}$); that of the deep box ($\tau_{T_D}$); and a coefficient describing the exchange of energy between the surface and deep boxes ($\theta$). Mathematically, it is formulated as:

$$\tau_{T_G} \frac{\mathrm{d}}{\mathrm{d}t} \Delta T_G = \lambda \, \Delta \mathrm{RF}_{\mathrm{warm}} - \Delta T_G - \theta \left( \Delta T_G - \Delta T_D \right); \tag{78}$$

$$\tau_{T_D} \frac{\mathrm{d}}{\mathrm{d}t} \Delta T_D = \theta \left( \Delta T_G - \Delta T_D \right). \tag{79}$$

So as to deduce the change in sea surface temperature ($T_S$) and in local surface temperatures ($T_L$) for each of our land regions (the $i$ axis from section 2.3.2), we use regional weighting coefficients ($\omega_{T_S}$ and $\omega_{T_L}$, respectively), so that:

$$\Delta T_S = \omega_{T_S} \, \Delta T_G; \tag{80}$$

$$\Delta T_L^i = \omega_{T_L}^i \, \Delta T_G. \tag{81}$$

The first set of parameters of this module, for global temperature, can be calibrated on twenty-five CMIP5 global circulation models. First, using outputs from the "abrupt4xCO2" and "piControl" experiments, we estimate the equilibrium temperature change at quadrupled $CO_2$ ($T_{4x}$) following the methodology by Gregory et al. (2004). Second, we fit the temporal response of global surface temperature to this quadrupled $CO_2$ experiment using the typical formula for a two-box model: $T_{4x} (1 - \pi \exp[-t/\tau_1] - (1 - \pi) \exp[-t/\tau_2])$, where $\pi$, $\tau_1$ and $\tau_2$ are temporary parameters used for the calibration only. Third, we deduce our three dynamical parameters (i.e. $\tau_{T_G}$, $\tau_{T_D}$ and $\theta$) by using the correspondence between the temporary parameters and ours, given by Geoffroy et al. (2013, table 1). Fourth, we deduce the climate sensitivity $\lambda$ of the model by normalizing $T_{4x}$ by the RF caused by a quadrupled $CO_2$ as quantified by the IPCC logarithmic formula given in equation (37).

The second set of parameters, for the pattern scaling, are calibrated on the same CMIP5 model chosen for the global temperature response. This pattern scaling can be based on the quadrupled $CO_2$ experiments, in which case the pattern is solely due to $CO_2$-induced warming – although, depending on the CMIP5 model, part of the regional response may come from the physiological effect of $CO_2$ (Sellers et al., 1996). Alternatively, it can be based on the transient historical and RCP experiments – when those RCPs are available – in which case the pattern is induced by all anthropogenic and natural perturbations, and it is thus expected to be more "realistic" but without a clear distinction of the role of each forcing. The parameter $\omega_{T_S}$ is calibrated thanks to a linear fit between yearly values of global and sea surface temperatures, whereas in the case of $\omega_{T_L}$ the linear fit is made with decadal averages of global and local surface temperatures. The CMIP5 fits are shown in figures S29 to S39.





### 2.11.3 Precipitations

Changes in global yearly precipitations ($P_G$) – actually used as another climate change indicator and to deduce changes in local yearly precipitations – are calculated following the simple model of Allan et al. (2013) (see also Shine et al., 2015). In this model, global precipitations vary with global temperature change and with the atmospheric fraction of RF. Two parameters are thus needed: one for the first term ($\alpha_{G_P} > 0$) that describes the long-term response of the hydrological cycle to global warming, and one for the second term ($\beta_{G_P} < 0$) that describes its short-term response to the local energy imbalance induced by radiatively active species. Hence:

$$\Delta P_G = \alpha_{P_G} \, \Delta T_G + \beta_{P_G} \, \Delta \text{RF}_{\text{atm}}. \tag{82}$$

As per surface temperature, we use a pattern scaling approach to deduce the local yearly precipitations ($P_L$) for each of our land regions, parameterized with regional weights ($\omega_{P_L}$):

$$\Delta P_L^i = \omega_{P_L}^i \, \Delta P_G. \tag{83}$$

The first set of parameters of this module, for global precipitations, can be calibrated on twenty-five CMIP5 global circulation models, chosen independently from the one used for the calibration of the temperature module. Using outputs from the "abrupt4xCO2" and "piControl" experiments, we calibrate the two parameters of equation (82) thanks to a linear fit with a constant term made between the global surface temperature and global precipitations. The constant term is assumed to correspond to the RF-term, since the radiative forcing is actually constant in the quadrupled $CO_2$ experiment. $\alpha_{G_P}$ is the slope of the fit, and $\beta_{G_P}$ is the $y$-intercept, albeit the latter needs to be divided by the RF of a quadrupled $CO_2$ as per the IPCC formula of equation (37), and by the value of $\pi_{\text{atm}}^{\text{CO}_2}$ from section 2.11.1.

The second set of parameters, for the pattern scaling, are also calibrated on the same CMIP5 model as the global precipitations response. The $\omega_{P_L}$ are fitted in the exact same way the $\omega_{T_L}$ are in the previous section, but logically using the precipitation CMIP5 variable this time. These CMIP5 fits are shown in figures S40 to S49.

### 2.11.4 Ocean heat content

The ocean heat content (OHC) – a third climate change indicator – is simply deduced from the two-box model used for the temperature. However, we need to introduce a coefficient ($\pi_{\text{ohc}}$) to account for the extra energy received by the planet but that is taken up to heat the continents, the atmosphere and to melt the ice. We have:

$$\frac{\text{d}}{\text{d}t} \Delta \text{OHC} = \pi_{\text{ohc}} \, A_{\text{Earth}} \left( \Delta \text{RF} - \frac{\Delta T_G}{\lambda} \right); \tag{84}$$

where we set $\pi_{\text{ohc}}$ = 0.94 (Otto et al., 2013, supplementary information section S1). Note also that by using RF instead of $\text{RF}_{\text{warm}}$, we implictly assume that the warming efficacies from section 2.11.1 originate from non-radiative processes only, which is not fully the case for the volcanic forcing (Gregory et al., 2016).





## 2.12 Numerical solving

When put together, all previous equations from (1) to (84) form a system of ordinary differential equations of first order, for a subset of the variables of the model. These variables are the state variables of the dynamical system described by the differential equations. They are compiled in table 1, along with the drivers of the model. Per definition, knowledge of both the drivers and the state variables, at any time-step, gives knowledge of all the other variables of the system, at that time-step. These other – secondary – variables are compiled in table 2. The differential system is solved with the forward Euler method (Euler, 1768) with a time-step ($\delta t$) that can be chosen before any simulation with OSCAR – although time-steps greater than a quarter of year systematically make the model diverge. This time-step is usually set to $\delta t = 1/6$ yr. We note that despite having a time-step for solving that is less than one year, the model's results cannot be interpreted at a time-scale shorter than the year, primarily because no seasonal process is implemented in the model.

## 3 First simulations

### 3.1 Experimental setup

We make two series of historical simulations, with the goal of evaluating the performance of each module of OSCAR v2.2 separately and of the fully coupled model itself. The simulations are realized within a probabilistic framework: a set a drivers and parameters is drawn randomly, with equiprobability, from the pool of potential driving datasets and parameterizations that is summarized in table 3. With the given drivers and parameters two simulations are made: one in which the atmospheric concentrations of well-mixed greenhouse gases, the total and per component radiative forcings, and the various climate variables are prescribed to the model; and another in which nothing more than the drivers is prescribed. The first simulation is called "offline", and the second "online". The offline simulation has the interest of uncoupling the different modules of OSCAR, thus separating them from each other and allowing an easier diagnosis of any potential issue or bias in each module. The online simulation is meant to diagnose the behavior of OSCAR when it is used as a proper Earth system model, i.e. when it is driven only by the anthropogenic perturbations of the system. The Monte Carlo ensemble size is 10,000 simulations which are drawn from a pool of more than $10^{43}$ potential combinations of parameters.

As described in section 2.3.2, the disaggregation of the terrestrial biosphere follows the nine regions of Houghton and Hackler (2001) and six biomes. The time-step of solving is one-sixth of a year. For the atmospheric concentrations of well-mixed greenhouse gases, the forcing data used for the offline simulation is from the IPCC (2013, tables AII.1.1a & AII.1.1b). For the component-based radiative forcings, the data is also from the IPCC (2013, table AII.1.2), altough we need a way to subdivide the two RFs that are kept aggregated by the IPCC: the one from non-$CO_2$ WMGHGs, and the one from aerosols (all effects). Regarding the former, we use the IPCC atmospheric concentrations which we combine with the data from Myhre et al. (2013b, tables 8.A.1 & 8.SM.1) to have component-based RFs. Regarding the latter, we take the timeseries from Meinshausen et al. (2011) for each individual aerosol direct effect and for the indirect effect. To ensure consistency, we rescale the component-based RFs so that: first, their value in 2010 meets the value provided by Myhre et al. (2013b); and then, their sum meets the



IPCC aggregated value every year. And finally, for the climate data used to force the offline simulation, we use the HadCRUT4 data for global surface temperature (Morice et al., 2012), the HadISST1 for sea surface temperature (Rayner et al., 2003), and the CRU TS3.23 dataset for local temperature and precipitations (Harris et al., 2014). For these three datasets, we assume the preindustrial equilibrium is their average over the 1901–1930 period.

## 3.2 Results

The following sections are dedicated to discussing the results of the historical simulations for the main variables of the model. Each section refers to one of figures 5 to 12. In the case of the offline simulation, we show and discuss the "reconstructed" timeseries of those variables that are prescribed to the model. In other words, in the following, the offline atmospheric growth rate and concentration of a given WMGHG are recontructed as the balance of the prescribed emissions and the simulated fluxes. The offline RFs are reconstructed on the basis of the reconstructed atmospheric concentrations. The climate variables, however, are reconstructed on the basis of the prescribed RFs, so that we can discuss the performance of the climate module alone, i.e. when it is not coupled to any other module.

### 3.2.1 Carbon dioxide (figure 5)

The median land-use change emissions simulated by the book-keeping module of OSCAR are of the same order of magnitude – though smaller than – the values reported by the global carbon project (Le Quéré et al., 2015) over the 1959–2010 period, be it for the online or offline simulations. The 90% range of our simulated emissions, however, is much larger than the uncertainty range reported by Le Quéré et al. (2015), and its distribution is far from a regular distribution. It can be shown (see Gasser and Ciais, 2013, appendix A) that these two results are a consequence of the biome-specific preindustrial carbon densities which are calibrated in section 2.3.2 on the TRENDY models. The large differences in carbon densities is a feature of the dynamic vegetation models themselves, although it is possible that our way of processing their output data exacerbates this discrepancy. More investigation in the matter is required, for instance using observed biomass densities as contraints, especially as the non-constrained setup leads to negative emissions under some parameterizations. We also note that the offline and online land-use change emissions are almost the same, as a direct consequence of our choice of definition that makes the land-use flux only slightly sensitive to environmental changes such as atmospheric $CO_2$ or climate (Gasser and Ciais, 2013).

The median land sink we simulate in the offline simulation is slightly smaller (in absolute value) than the estimate by Le Quéré et al. (2015), more importantly smaller in the online simulation. The slightly smaller median value in the offline case can be explained by the weight of the four (out of thirteen) preindustrial land-covers for which we use the cross-walking table of Poulter et al. (2011) to translate biomes into plant functional types (see section 2.3.2). Using this table indeed gives a more important fraction of land covered by bare soil than it is the case in most of the TRENDY models. As for the online simulation, the reduced land sink is also a consequence of the warmer tropical climate simulated by OSCAR than the one prescribed with the CRU dataset in the offline simulation (see below). The interannual variability of the land sink simulated by OSCAR in the offline case does not match that from Le Quéré et al. (2015), but we do not expect our crude and aggregated approach to model the terrestrial biosphere's response to climate to be able to reproduce this variability, especially as some factors such




as volcanoes do not directly influence the terrestrial carbon-cycle of our model while they seem to do in reality (e.g. Raupach et al., 2014). The large spread in our estimated land sink has the same origin as that in our estimated land-use change emissions, although this time the distribution appears more regular.

The median ocean sink OSCAR simulates matches relatively well the estimate by Le Quéré et al. (2015), albeit it is slighty stronger (in absolute value) in the online case. The discontinuous probability distribution of the ocean sink in the offline case reflects the fact that we only have twelve possible parameterizations for this module, when climate is fixed. In the online case, it also seems to be overestimated prior to the period over which we have data to compare it to.

In both the online and offline simulations, the simulated atmospheric growth rate is very close, on average, to the one reported by NOAA/ESRL (Tans and Keeling, 2015). In the online case, this happens in spite of the relatively small land sink discussed above, owing to the compensation of the reduced land sink by an enhanced ocean sink. This shows that there is a negative feedback loop occuring in the online setup. This loop occurs through the oceanic carbon-cycle: when the land sink is too low, atmospheric carbon dioxide increases faster, which in turn increases the ocean sink. This kind of anti-correlation between two of the global carbon budget's fluxes is also found between the land sink and land-use change emissions: a high productivity configuration of the model simulates high emissions of land-use change – because of high carbon density biomes – but also high terrestrial carbon sink.

Finally, regarding excess atmospheric $CO_2$, both median simulations follow fairly well the observations since 1959, with a slight positive offset for the online case and a slight negative one for the offline one, of ∼5 ppm in both cases. In the online case, however, the simulated atmospheric $CO_2$ prior to the direct observations is very close to the estimates derived from ice-cores (Etheridge et al., 1996; MacFarling Meure et al., 2006), at least until the simulation reaches the atmospheric plateau of the 1940s. Therefore, the offset we simulate over the recent period is a consequence of the model "missing" the plateau, as all complex models do (Bastos et al., 2016). The spread in the results from the two setups is high, but the spread in the offline simulation is much higher than in the online case, owing mainly to the spread in our simulated land sink. Some parameterizations in the offline setup even lead to negative atmospheric $CO_2$, resulting from combined negative land-use emissions and strong land sink. This unrealistic behavior of the model puts forward the need to use observational constraints to select only a subset of the parameterizations in future works.

### 3.2.2 Methane (figure 6)

The emissions from biomass burning are shown and discussed here, despite being mainly a product of the carbon-cycle in OSCAR, since they are part of the atmospheric balance of methane. One can see that our approach of calculating these emissions endogenously gives values of the same order of magnitude than that of Lamarque et al. (2010), albeit with a different temporal profile. This different profile of ours follow closely that of land-use change emissions in figure 5, which indicates that our emissions from biomass burning are mainly the product of land-use and land-cover change; or in other words that the second term of equation (38) dominates. In the offline simulation, however, there is a noticeable interannual variability, showing that the environmental conditions – and especially climate – also affect our biomass burning emissions; or in other words that the first term of equation (38) is not negligible.



When compared to the multi-model mean of WETCHIMP (Melton et al., 2013), our offline predicted change in the emission of methane by natural wetlands is of the right order of magnitude, albeit without a good reproduction of the interannual variability simulated by complex models. We see this relatively good performance for the offline simulation, i.e. for an experimental protocol with OSCAR that is very close to the one used in WETCHIMP. For the online simulation, however, one can

see that our simulated wetlands emissions are much lower – by a factor 2 – to that simulated in the offline case. This come from the inability of OSCAR to simulate a regional climate change – and especially precipitations (see below) – close to the forcing data we use in the offline simulation, therefore affecting the wetlands area extent predicted by the model.

The median lifetime of methane with regard to the OH sink which we simulate is very close to the best guess value of Prather et al. (2012) for present days. This is an *ex-post* justification to our arbitrary rescaling of the preindustrial lifetime $\tau_{\mathrm{OH}}^{\mathrm{CH4}}$

in section 2.5.1. Here, we also note that our 90% spread in methane's lifetime is greater than the correponding uncertainty range provided by Prather et al. (2012), particularly in the online simulation. This stems from the large spread in our simulated emissions of biomass burning – which itself is a consequence of the spread in land-use change emissions – as the biomass burning emissions of $NO_x$, CO and VOCs impact the OH sink capacity.

In the online simulation, the median atmospheric growth rate of methane we simulate is close to the observed one, over

the short period of observation we have at our disposal. OSCAR manages to reproduce the slowdown of atmospheric increase around the year 2000; this slowdown is mainly driven by anthropogenic emissions in our model. After 2005, however, the atmospheric growth resumption is too fast when compared to observations. In the offline simulation, the picture is completely different: the reconstructed atmospheric growth rate is systematically higher than in the online case, by 10 to 20 MtC yr$^{-1}$. If our wetlands emissions can explain 5 MtC, the rest must come from the anthropogenic emissions of methane we use for

reconstructing the growth rate. The remaining 5 to 15 MtC represent between 5% (around 2000) and 30% (in 1900) of the anthropogenic emissions. These relatively small percentages stress how sensitive to anthropogenic emissions predicted atmospheric methane is: the annual growth rate of $\sim$10 MtC yr$^{-1}$ results from the balance between source or sink fluxes of $\sim$250 MtC yr$^{-1}$, and any small error in one of the two fluxes can have marked impact on the growth rate. In the online configuration, there is an obvious negative feedback loop that reduces the importance of this: the sink is directly proportional to the

atmospheric concentration; but this feedback loop is cut off in the offline configuration.

Regarding atmospheric CH$_4$, in the online configuration we simulate a concentration that is close to recent observations albeit slightly lower. The distance between the median of our ensemble and AGAGE (Prinn et al., 2013) is $\sim$40 ppb over 1987–2005 and then decreases to be virtually zero in 2010. Before that, however, when compared to ice-cores data (Etheridge et al., 1998; MacFarling Meure et al., 2006) the simulated atmospheric CH$_4$ is systematically higher by $\sim$100 ppb. With the

offline configuration, as a direct consequence of the systematic overestimate of the reconstructed atmospheric growth rate, the reconstructed atmospheric concentration we simulate is completely offtrack. This could be solved by using our own estimates of compatible methane emissions (see e.g. Gasser et al., 2015) which would be 5% to 30% lower than those used here (and described in section 2.2.1), as explained above; but also by using constraints to exclude unrealistic realisations of the Monte Carlo ensemble.





### 3.2.3 Nitrous oxide (figure 7)

The nitrous oxide emissions from biomass burning as shown here mainly to point out that they are strictly similar to that of methane in figure 6. This is true for all non-$CO_2$ species in OSCAR: given our modelling approach, their biomass burning emissions roughly proportional by a factor equal to the ratio of two $\alpha_{\mathrm{bb}}^{\mathrm{X}}$ (see section 2.4.1).

The median lifetime of nitrous oxide with regard to the stratospheric sink which we simulate is very close to the best guess value of Prather et al. (2015) for present days. Its distribution, however, is asymmetrical and somewhat discontinuous. Both features are direct consequences of the distribution in the model's estimates of the lifetime which we base our parameter on; but the latter one also indicates that we do not have enough available parameterizations to produce a proper uncertainty range.

On average, the median atmospheric growth rate we simulate is close to the observed one over 1979–2010, although slightly

smaller for the offline simulation. The observed variability, however, is not reproduced by our model, be it in the online or offline setup. This suggests that a biological process related to nitrous oxide is missing in our model. Processes such as biological production in terrestrial or aquatic systems are viable candidates (Ciais et al., 2013b).

In the online simulation, the excess atmospheric concentration we simulate is lower than the one observed: the median is actually parallel to the observations with a distance of ∼4 ppb. This feature indicates that the growth rate simulated over the

recent period is good – as we explained above – and thus that the difference between simulation and observation originates from the earlier period. This is confirmed by the comparison with ice-cores data (MacFarling Meure et al., 2006). Assuming that our estimate of the nitrous oxide sink is right, the difference could be explained by any phenomenon that would imply higher emissions in the past than we use as input here, be they anthropogenic or of natural origin. As for the offline configuration, the simulated atmospheric $N_2O$ is even lower, owing to the lower growth rate mentioned above, and its spread is larger because of

the same reasons as for atmospheric $CH_4$.

### 3.2.4 Halogenated compounds (figure 8)

While other species are shown in figure S50, here we show only the first compound of each group of halogenated compounds (i.e. HFC-23 for HFCs, $CF_4$ for PFCs and CFC-11 for ODSs) to illustrate two points. First, OSCAR is able to reproduce relatively well the past evolution of the atmospheric concentration of these compounds, although not with very good performance

in all cases. Second, the fact that we only have one set of preindustrial lifetimes and one dataset of anthropogenic emissions hampers our ability to produce a proper distribution of results with OSCAR. Hence, if one or the other data is wrong, the simulation with OSCAR will also be wrong. Alternative parameters and/or input data should be used in future versions of the model, or – more importantly – in any future study that would focus on those compounds.

If we look at the variables that summarize the two effects of the halogenated compounds within the climate system, that is

effective equivalent stratospheric chlorine and radiative forcing, we can have an overview of the performance of this module. Regarding the EESC simulated by our model, it is lower than the one calculated on the basis of the IPCC (2013) atmospheric concentrations and the fractional release parameters from Newman et al. (2007) used by the WMO (Montzka et al., 2011). Note, however, that in OSCAR those fractional release factors can also take alternative values, as illustrated by the three lines





in the distribution of the offline EESC. Regarding the combined radiative forcing of all halogenated compounds, the offline simulation gives a slightly higher value than the IPCC's (Myhre et al., 2013b), whereas the online simulation gives a slightly lower one. In both cases, the values remain within the 90% uncertainty range assessed by the IPCC.

### 3.2.5 Ozone (figure 9)

Regarding tropospheric ozone, the median change in burden simulated by OSCAR is very close to the only point in time we have from the IPCC (2013, table AII.5.2) which is for the change in burden over 1850–2000. The corresponding RF, however, is higher in our simulation than the one provided by the IPCC (Myhre et al., 2013b) for the year 2010. Given that OSCAR seems to perform well over 1850–2000, the cause of the discrepancy between the IPCC RF estimate and ours can be a different estimate of change in burden before or after that period and/or a different radiative efficiency of tropospheric ozone. In any

case, our estimate remains within the IPCC uncertainty range, but it must be noted that our 90% range is almost systematically higher than the IPCC best guess.

Regarding stratospheric ozone, our slightly underestimated EESC induces a slightly underestimated change in column burden (in absolute value), again over the reference period 1850–2000. Nonetheless, the estimate by the IPCC (2013) is well within our 90% range – a range that is discontinuous in the offline configuration, as could be expected from the discontinuity of the EESC seen in figure 8. The corresponding median RF we estimate is close to the IPCC best guess and its spread is also

close to the uncertainty range provided by the IPCC, except that it does not go into the positive value domain.

### 3.2.6 Aerosols (figure 10)

Regarding the direct effect of aerosols, OSCAR's ability to match the IPCC best guess (Myhre et al., 2013b) in 2010 varies with the aerosol considered. In the case of sulphates, the median RF we simulate is slightly smaller (in absolute value) than the IPCC

reference, while the spread is larger than the reference and has a non-regular distribution. The cases of POA and BC are very comparable: our median RFs are significantly smaller (in absolute value) than the IPCC references, and the distributions are close to a log-normal one and with a relatively consistent spread. With both aerosols, however, if we remove the contribution of biomass burning aerosols to the IPCC best guesses, our median estimates are much closer. This odd feature does not affect much the overall performance of the model (see next section), as the IPCC best guess estimate for combined biomass burning

POA and BC is zero. It strongly suggests, however, that the way these biomass burning aerosols are treated is OSCAR can be improved. In the case of nitrate, our median RF is relatively close to the IPCC best guess, whereas our distribution does not go as far in the negative values as the IPCC uncertainty range. In the case of SOA, our median RF is very small – owing to the fact that one out of three simulations has the SOA turned off – and the distribution clearly show that we only have three possible paramererizations for this aerosol. Also, because all the radiative efficencies of SOA available to OSCAR are negative, the only

way it could go into the negative value domain would be to have varying biogenic emissions of NMVOCs, which is nit the case in this version.

Regarding the cloud effect of aerosols, which includes both the so-called semi-direct and indirect effects, OSCAR performs well and its median estimate meets the IPCC best guess in 2010. This is mostly due to the way this effect is calculated in our





model, as the main sensitivity parameter of the module (i.e. $\Phi$) is rescaled using the IPCC estimate. Nonetheless, this and the shape of the distribution, that is close to a log-normal one, show that our simple formulation of the cloud effect is consistent. Note also that the online and offline simulations are very close, both for the direct and cloud effects, because of the limited role of climate in our aerosol module.

### 3.2.7 Radiative forcing (figure 11)

When we combine together the RF induced by all well-mixed greenhouse gases, we see that the median of both our online and offline simulations are slightly higher in 2010 than the estimate by Myhre et al. (2013b), albeit with a larger spread than the reference in the online case, and a much larger spread in the offline one. The latter feature is a direct consequence of the large spread in the offline simulations of atmospheric $CO_2$ and $CH_4$ discussed above. When the RF induced by all near-term climate forcers is combined, we see similarly that the median of both our online and offline simulations is close to the IPCC estimate for 2010. This time, however, our simulated spread is relatively consistent with the IPCC uncertainty range.

Regarding the two RFs induced by surface albedo, one can see that our two simple modules simulate values that meet the IPCC estimate for the year 2010. For black carbon deposition on snow, this could be expected from our rescaling of the global sensitivity parameter $\alpha_{\mathrm{rf}}^{\mathrm{BCsnow}}$, although the spread in our results is smaller than the IPCC uncertainty. For land-cover change, however, no parameter was rescaled to meet the IPCC best guess, and the distribution of our simulated RF shows that this median result is actually the product of several parameterizations with very contrasted results. We also note that the offline and online simulations of this RF from land-cover change are strictly equal because the module is driven only by LULCC drivers, and it is therefore not coupled to any other module.

All in all, the total RF simulated by OSCAR – which is the sum of the above four RFs and the three drivers prescribed directly as radiative forcing – has a median value in the year 2010 close to the IPCC best guess, but slightly higher. In the online case it has a relatively consistent spread, whereas in the offline one the spread is much larger.

### 3.2.8 Climate (figure 12)

Global mean surface temperature, which is our prime proxy of climate change, is relatively well simulated by OSCAR over the 1900–2010 period. We note, however, that the 1940s warmer period is not reproduced, and during the last ten years of simulation the simulated temperature tends to be higher than the observations. Interestingly, OSCAR simulates a slowdown of the warming during these last ten years – the so-called hiatus period. The fact that the slowdown is simulated in both the offline and online setups suggests it is a feature our climate module alone. However, the lack of interannual variability in OSCAR makes any further investigation on the topic virtually impossible. Note also that the offline simulation gives a narrower range than the online one because only one set of radiative forcings is prescribed in the former case.

As for the global sea surface, one can see here the limits of our pattern scaling approach: the single proportionality parameter makes the timeseries of sea surface temperature homothetic to that of global surface temperature. If the simulated temperature follows relatively well the observations over 1900–2010, the simulated temporal variability does not match the observed one. Similarly, the simulated local surface temperatures, shown in figure S51, are proportional to the global one, which gives



temperature changes consistent with the CRU dataset (Harris et al., 2014) in most regions, with the notable exception of tropical regions. This suggests regional processes should be accounted for, especially as some anthropogenic activities such as emission of short-lived species and land-use change can have important regional impacts. This is discussed hereafter in conclusion.

5  Although we cannot compare our global yearly precipitations with a long enough timeseries of observation, we can note that OSCAR simulates a wide range of precipitation changes, with a non-negligible difference between the offline and online configurations. This is mostly caused by the difference between the simulated RF of aerosols in the online setup and the prescribed RF in the offline one. Regarding local yearly precipitations, shown in figure S52, OSCAR does not manage to capture the past variation of this variable, in any of our regions. This has limited impact on the model's results, since in section

10 2.3.2 we calibrate the sensitivity parameters of NPP and heterotrophic respiration in two steps, the first of which being driven by temperature alone. It does, however, impact our simulated methane emissions from wetlands (see above). More work is needed to improve that aspect of the model.

  Finally, the ocean heat content simulated with our model is of the right order of magnitude, owing to the good simulated RF and temperature. It follows relatively well the variations of the observations for both online and offline simulations, except

15 over the last 10 years of simulation. This could be explained by our choice of a single value for $\pi_{ohc}$, while this parameter should ideally be calibrated on each of the CMIP5 climate models we emulate. Alternatively, another explanation could be that our reference from NOAA/NODC (Levitus et al., 2012) actually estimates the ocean heat content down to a 2000-meter depth, potentially creating a slight bias in our comparison.

## 4 Conclusions

20 In this paper, we provided a complete description of the compact Earth system model OSCAR v2.2, and we presented the model's results in the case of an historical simulation. Overall, despite some caveats discussed in the previous section, we conclude that the model performance is good, especially given its level of complexity. OSCAR manages to satisfactorily reproduce most of the past changes in the global Earth system, with an even better performance over the recent period when better driving data is available. However, we note that a good performance of a simple model over the historical period does

25 not warrant a good performance in any other simulation. In the case of OSCAR, since its parameters are generally calibrated on simulations that go relatively far from the historical conditions (e.g. under quadrupled $CO_2$, or following the RCPs), we expect the model to provide reliable results over the plausible range of future climate change, in other words to cover all scenarios by Clarke et al. (2014). OSCAR's domain of validity is not as broad as that of complex models, however, and we would not recommend using the model e.g. to perform paleoclimate studies. Ultimately, OSCAR's domain of validity should

30 be investigated in future studies.

  The fact that OSCAR has been developed to be used in a probabilistic setup is an additional strength of the model, although the spread in the model's results for some components may greatly differ from the uncertainty range assessed by studies based on more complex models and/or observations. In addition to the reasons discussed in the sections above, there are two more



general causes to that feature, owing to the principles underpinning OSCAR's development (expounded in section 2.1). First, because all the modules of OSCAR interact with each other, the model's overall causal chain is fairly complex (as illustrated in figure 1) and it has many degrees of freedom – actually more than most CMIP5 complex Earth system models. These many degrees of freedom increase the odds of seeing a given simulation diverge, or at least depart unreasonably from the plausible

range of results. Second, OSCAR is not designed to emulate a given complex Earth system model as a whole: each of its modules is *per se* an emulator, and OSCAR is the combination of these emulators. Consequently, in a given parameterization, two modules could emulate the sensitivities of two complex models that are physically inconsistent with one another (e.g. the implicit ocean transport of the climate module could be inconsistent with that of the carbon-cycle module), therefore potentially leading to unreasonable results. These two elements explain why OSCAR's average or median simulation can

differ from the average of a model intercomparison exercise we used for calibration, and why the model's results can show very large spreads. A way to solve this and improve the probabilistic setup is to use observational constraints, either to rate a given parameterization and therefore give it a lower weight if its too far from the observations (e.g. Steinacher et al., 2013), or more abruptly to remove from the pool of the Monte Carlo experiment the parameterizations that lead to unrealistic results (e.g. Gasser, 2014). In any case, the observational constraints must be relevant to the study: for global climate change projections,

atmospheric concentrations of greenhouse gases and global surface temperature could suffice; while for a study focusing e.g. on land carbon-cycle, additional constraints on NPP and carbon densities might be required.

To conclude, we want to suggest a few tracks for future development of the model. Despite its overall good performance, the model can indeed be improved, especially in terms of consistency of modelling. We see three broad aspects of the model for doing so. First, the carbon-cycle can be improved by inclusion of nutrient limitations for the land carbon-cycle, and of the

biological pump for the ocean carbon-cycle. Inclusion of the nitrogen-cycle would couple the carbon-cycle and the atmospheric chemistry, as the carbon sinks would be affected by deposition of active nitrogen that would be induced by $NO_x$ and $NH_3$ emissions (e.g. Ciais et al., 2013b). This would also allow to compute endogenously some of the biogenic emissions of $N_2O$, $NO_x$ or $NH_3$, which would probably change our estimated past evolution of atmospheric nitrous oxide, e.g. by giving it more annual variability in the offline simulation. Second, the whole of OSCAR's atmospheric chemistry can be improved by making

it consistent. In OSCAR v2.2, the atmospheric chemistry is a patchwork of many sensitivity studies. When we choose the parameters for e.g. the stratospheric $N_2O$ sink, it should actually be coupled to the ozone stratospheric chemistry (e.g. Prather, 1998). Also, coupling of the tropospheric and stratospheric chemistries would be an improvement, especially for ozone, as would a finer regionalization be. We note however that a tremendous amount of factorial simulations by complex chemistry-transport models would be needed to make such an improvement. Third, the climate module can be improved, especially as it

performs poorly at the regional scale. This would need to be done, however, by accounting for regional processes that affect temperature or precipitations, such as the physiological effect of $CO_2$ (e.g. Sellers et al., 1996) and thus the biophysical effect of land-use and land-cover change (e.g. Feddema et al., 2005), or the local effects of atmospheric pollution (e.g. Ramanathan et al., 2001). Implementing this in OSCAR would also require an important amount of factorial simulations, so as to be able to apply forcing-dependent patterns of climate change, or alternatively a complete rewriting of the climate module to explicitly

model the local energy imbalance and water-cycle. In addition to these three huge undertakings, we acknowledge that many




smaller improvements could be made. But ultimately, the future development of OSCAR will depend on the data from complex models that will be made available.

## 5 Code availability

The source code of this version of OSCAR will be made available upon release of the final version of this description paper. A
brief user manual will be provided with the code.

## Appendix A: Changelog

### A1 OSCAR v2.1

Version 2.1 of OSCAR is completely described by Gasser (2014), although in French. Partial descriptions can be found in other studies (Cherubini et al., 2014; Li et al., 2016).

Main changes between v2.1 and v2.2 are the following: development of the ocean carbon-cycle module to include the stratification effect calibrated on CMIP5 models; extension of the terrestrial carbon-cycle module to be calibrated on many TRENDY and CMIP5 models; creation of a wildfire module; extension of the wetlands module to be calibrated on many WETCHIMP models; development of the stratospheric sink module to include the effect of ozone depleting substances and age-of-air change; development of the tropospheric ozone module to include a regionalization and the effect of climate change;
development of the stratospheric ozone module to include the effect of nitrous oxide and climate change; development of the aerosols module to have explicit and regionalized parameterizations; creation of the surface albedo modules; development of the climate module to include a global precipitation response.

Many other small and specific changes were also made during the development of the latest version.

### A2 OSCAR v2.0

Version 2.0 of OSCAR is exactly the same as version 2.1 with two significant exceptions. First, non-$CO_2$ species were not modelled at all, which means that v2.0 was a carbon-climate model. Second, only one climate response was available, that developed by Hooss et al., instead of the CMIP5 responses available now. Version 2.0 was used by Gasser and Ciais (2013) and very briefly described therein.

It can also be noted that the main change between the previous versions of OSCAR and v2.0 is the computing language
used to code the model. While previous versions were coded in Scilab, the next versions (i.e. from v2.0 onward) are coded in Python.

### A3 OSCAR v1.1

Version 1.1 of OSCAR is an update of version 1.0, described by Gitz (2004). The update is limited to the inclusion of a basic climate response and of a simple climate-carbon feedback, for the terrestrial carbon-cycle only.





## A4   OSCAR v1.0

Version 1.0 of OSCAR is described by Gitz and Ciais (2003). At that time, it was a simple carbon-cycle model designed to specifically focus on land-use change issues, as the book-keeping module was already included in the model (albeit not exactly coded in the way it is now).

**Appendix B: Complex models used for calibration**

These models are those whose outputs we use to calibrate some of OSCAR's parameters. In other words, we do not list here the models for which we simply read OSCAR's parameter value in e.g. a table of another study. Note that here we give the models' name as given by the study we base our calibration on. These names may vary across studies and from the official name itself.

**B1   CMIP5**

For the ocean carbon-cycle, stratification effect (section 2.3.1): CESM1-BGC, IPSL-CM5A-LR and MPI-ESM-LR.

For the land carbon-cycle, transient response of net primary productivity and heterotrophic respiration (section 2.3.2): BCC-CSM1.1, CESM1-BGC, CanESM2, HadGEM2-ES, IPSL-CM5A-LR, MPI-ESM-LR and NorESM1-ME.

For the land carbon-cycle, transient response of wildfires (section 2.3.2): CESM1-BGC, IPSL-CM5A-LR, MPI-ESM-LR
and NorESM1-ME.

For the atmospheric burden of sulphate, primary organic and black carbon aerosols (section 2.9.1): CSIRO-Mk3.6.0, GDFL-CM3 and MIROC-CHEM.

For the atmospheric burden of secondary organic aerosols (section 2.9.1): GFDL-CM3.

For the indirect effect of aerosols (section 2.9.2): CSIRO-Mk3.6.0, and IPSL-CM5A-LR.

For the climate module, both the temperatures and the precipitations (sections 2.11.2 and 2.11.3): ACCESS1.0, ACCESS1.3, BCC-CSM1.1, BCC-CSM1.1m, CanESM2, CCSM4, CNRM-CM5, CNRM-CM5.2, CSIRO-Mk3.6.0, GFDL-CM3, GFDL-ESM2G, GFDL-ESM2M, GISS-E2-H, GISS-E2-R, HadGEM2-ES, IPSL-CM5A-LR, IPSL-CM5A-MR, IPSL-CM5B-LR, MIROC5, MIROC-ESM, MPI-ESM-LR, MPI-ESM-MR, MPI-ESM-P, MRI-CGCM3 and NorESM1-M.

**B2   TRENDY v2**

For the terrestrial carbon-cycle, preindustrial net primary productivity and heterotrophic respiration (section 2.3.2): CLM4.5, JSBACH, JULES, LPJ, LPJ-GUESS, LPX-Bern, OCN, ORCHIDEE and VISIT.

For the terrestrial carbon-cycle, preindustrial wildfires (section 2.3.2): CLM4.5, JSBACH, LPJ, LPJ-GUESS, ORCHIDEE and VISIT.



### B3  WETCHIMP

For the natural wetlands, preindustrial state (section 2.5.2): CLM4-Me, DLEM, IAP-RAS, LPJ-Bern, LPJ-WSL, ORCHIDEE and SDGVM.

For the natural wetlands, transient reponse of the area extent (section 2.5.2): CLM4-Me, DLEM, LPJ-Bern, ORCHIDEE, SDGVM and UVic-ESCM.

### B4  CCMVal2

For the stratospheric sink, transient response of the age of air (section 2.6.1): AMTRAC3, CAM3.5, CMAM, Niwa-SOCOL, SOCOL, ULAQ and UMUKCA-UCAM.

For the stratospheric ozone, transient response to chlorine and climate change (section 2.8.2): AMTRAC3, CCSRNIES, CMAM, CNRM-ACM, LMDZrepro, MRI, Niwa-SOCOL, SOCOL, ULAQ, UMSLIMCAT and UMUKCA-UCAM.

### B5  ACCMIP

For the tropospheric ozone, transient response to precursors emissions (section 2.8.1): CICERO-OsloCTM2, NCAR-CAM3.5, STOC-HadAM3 and UM-CAM.

For the tropospheric ozone, transient response to climate change (section 2.8.1): CESM-CAM-superfast, GFDLAM3, GISS-E2-R, MIROC-CHEM, MOCAGE, NCARCAM3.5, STOC-HadAM3 and UM-CAM.

For the atmospheric burden of sulphate, primary organic and black carbon aerosols (section 2.9.1): GISS-E2-R.

For the atmospheric burden of secondary organic aerosols (section 2.9.1): GISS-E2-R.

For the indirect effect of aerosols (section 2.9.2): GFDL-AM3, GISS-E2-R, HadGEM2, MIROC-CHEM and NCARCAM5.1.

*Acknowledgements.* First of all, we thank all the modelling team that produced the data we processed to build OSCAR. We also thank those who shared data or info from their model or study: V. K. Arora, P. Cadule, A. R. Douglass, W. Fu, C. D. Jones, J.-F. Lamarque, Y. H. Lee, S. Levis, K. Lindsay, U. Lohmann, J. R. Melton, V. Naik, J. Pongratz, J. T. Randerson, B. Ringeval, L. Rotstayn, X. Shi, D. T. Shindell, S. Sitch, B. D. Stocker, J. Tjiputra, N. Viovy, S. Watanabe, H. Yu. Development of versions 2.0 and 2.1 was part of the ACACCYA project funded by the GIS Climat-Environnement-Société. Development of version 2.2 was supported by the European Research Council Synergy project IMBALANCE-P (grant ERC-2013-SyG-610028). T.G. also acknowledges support from the Research Council of Norway with a visiting researcher grant (#249972) during the writing of this paper. Download and processing of CMIP5 data was made on the IPSL Prodiguer-Ciclad facility which is supported by CNRS, UPMC and Labex L-IPSL, and funded by the ANR (grant #ANR-10-LABX-0018) and the European FP7 IS-ENES2 project (grant #312979). Download of ACCMIP and CCMVal2 data was made on the BADC facility which is part of the NERC-NCAS.





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





**Table 1.** List of drivers and state variables of the model.

| Notation | Name | Section |
|---|---|---|
| | Drivers | |
| $E_{\text{FF}}$ | Anthropogenic emissions of carbon dioxide from fossil-fuel burning and cement production. | 2.2.1 |
| $E_{\text{X}}$ | Anthropogenic emissions of a species X; X being any species but carbon dioxide. | 2.2.1 |
| $\delta A^{i,b_1 \rightarrow b_2}$ | Yearly land-cover change from biome $b_1$ to biome $b_2$; in region $i$. | 2.2.2 |
| $\delta H^{i,b}$ | Yearly harvest of biomass from biome $b$; in region $i$. | 2.2.2 |
| $\delta S^{i,b_1 \leftrightarrow b_2}$ | Yearly shifting cultivation between biomes $b_1$ and $b_2$; in region $i$. | 2.2.2 |
| $\text{RF}_{\text{con}}$ | Radiative forcing induced by aviation contrails and induced cirrus. | 2.2.3 |
| $\text{RF}_{\text{volc}}$ | Radiative forcing induced by volcanic aerosols. | 2.2.3 |
| $\text{RF}_{\text{solar}}$ | Radiative forcing induced by solar irradiance. | 2.2.3 |
| | State variables | |
| $\Delta C_{\text{surf}}^{o}$ | Carbon pool of the surface ocean; in subdivided box $o$. | 2.3.1 |
| $\Delta c_{\text{veg}}^{i,b}$ | Areal carbon pool of the vegetation; in region $i$ and biome $b$. | 2.3.2 |
| $\Delta c_{\text{litt}}^{i,b}$ | Areal carbon pool of the litter; in region $i$ and biome $b$. | 2.3.2 |
| $\Delta c_{\text{soil}}^{i,b}$ | Areal carbon pool of the soil; in region $i$ and biome $b$. | 2.3.2 |
| $\Delta C_{\text{veg,luc}}^{i,b,a}$ | Carbon pool of the LUC-disturbed vegetation; in region $i$, biome $b$ and age-class $a$. | 2.3.3 |
| $\Delta C_{\text{hwp,luc}}^{w,i,b,a}$ | Carbon pool of the harvested wood products; of type $w$, in region $i$, biome $b$ and age-class $a$. | 2.3.3 |
| $\Delta C_{\text{litt,luc}}^{i,b,a}$ | Carbon pool of the LUC-disturbed litter; in region $i$, biome $b$ and age-class $a$. | 2.3.3 |
| $\Delta C_{\text{soil,luc}}^{i,b,a}$ | Carbon pool of the LUC-disturbed soil; in region $i$, biome $b$ and age-class $a$. | 2.3.3 |
| $\Delta A^{i,b}$ | Area of a biome $b$; in region $i$. | 2.2.3 |
| $\Delta \text{CO2}$ | Atmospheric concentration of carbon dioxide. | 2.3.4 |
| $\Delta \text{X}_{\text{lag}}$ | Lagged concentration of a species X; X being methane, nitrous oxide or any HFC, PFC or ODS. | 2.4.2 |
| $\Delta \text{CH4}$ | Atmospheric concentration of methane. | 2.5.3 |
| $\Delta \text{N2O}$ | Atmospheric concentration of nitrous oxide. | 2.6.2 |
| $\Delta \text{X}$ | Atmospheric concentration of a species X; X being any HFC, PFC or ODS. | 2.7.2 |
| $\Delta T_G$ | Global mean surface temperature. | 2.11.2 |
| $\Delta T_D$ | Temperature of the deep ocean. | 2.11.2 |
| $\Delta \text{OHC}$ | Ocean heat content. | 2.11.4 |



**Table 2.** List of secondary variables of the model.

| Notation | Name | Section |
|---|---|---|
| | Secondary variables | |
| $\Delta F_{\mathrm{in}}$ | Flux of carbon going in the surface ocean. | 2.3.1 |
| $\Delta F_{\mathrm{out}}$ | Flux of carbon going out the surface ocean. | 2.3.1 |
| $\Delta\mathrm{dic}$ | Dissolved inorganic carbon in the surface ocean. | 2.3.1 |
| $\Delta h_{\mathrm{mld}}$ | Mixing layer depth of the surface ocean. | 2.3.1 |
| $\Delta F_{\mathrm{circ}}^{o}$ | Flux of carbon going from the surface ocean to the deep ocean. | 2.3.1 |
| $\Delta\mathrm{npp}^{i,b}$ | Areal net primary productivity; in region $i$ and biome $b$. | 2.3.2 |
| $\Delta e_{\mathrm{fire}}^{i,b}$ | Areal wildfire flux; in region $i$ and biome $b$. | 2.3.2 |
| $\Delta f_{\mathrm{mort}}^{i,b}$ | Areal mortality flux; in region $i$ and biome $b$. | 2.3.2 |
| $\Delta\mathrm{rh}_{\mathrm{litt}}^{i,b}$ | Areal heterotrophic respiration from the litter carbon pool; in region $i$ and biome $b$. | 2.3.2 |
| $\Delta f_{\mathrm{met}}^{i,b}$ | Areal flux of carbon going from the litter to the soil carbon pool; in region $i$ and biome $b$. | 2.3.2 |
| $\Delta\mathrm{rh}_{\mathrm{soil}}^{i,b}$ | Areal heterotrophic respiration from the soil carbon pool; in region $i$ and biome $b$. | 2.3.2 |
| $\Delta F_{\downarrow\mathrm{ocean}}$ | So-called "ocean sink" of carbon dioxide. | 2.3.4 |
| $\Delta F_{\downarrow\mathrm{land}}$ | So-called "land sink" of carbon dioxide. | 2.3.4 |
| $\Delta E_{\mathrm{LUC}}$ | So-called carbon dioxide "emissions from land-use and land-cover change". | 2.3.4 |
| $\Delta\mathrm{RF}^{\mathrm{CO2}}$ | Radiative forcing induced by atmospheric carbon dioxide. | 2.3.4 |
| $\Delta E_{\mathrm{bb}}^{\mathrm{X},i}$ | Emissions of a species X from biomass burning; X being any species but a HFC, PFC or ODS. | 2.4.1 |
| $\Delta F_{\downarrow}^{\mathrm{CH4}}$ | Total atmospheric sink of methane. | 2.5.1 |
| $\Delta e_{\mathrm{wet}}^{i}$ | Areal emissions of methane by wetlands; in region $i$. | 2.5.2 |
| $\Delta A_{\mathrm{wet}}^{i}$ | Wetlands area extent; in region $i$. | 2.5.2 |
| $\Delta E_{\mathrm{wet}}^{i}$ | Emissions of methane by wetlands; in region $i$. | 2.5.2 |
| $\Delta\mathrm{RF}^{\mathrm{CH4}}$ | Radiative forcing induced by atmospheric methane. | 2.5.3 |
| $\Delta\mathrm{RF}^{\mathrm{H2Os}}$ | Radiative forcing induced by stratospheric water vapor. | 2.5.3 |
| $\Delta F_{\downarrow}^{\mathrm{N2O}}$ | Total atmospheric sink of nitrous oxide. | 2.6.1 |
| $\Delta\mathrm{RF}^{\mathrm{N2O}}$ | Radiative forcing induced by atmospheric nitrous oxide. | 2.6.2 |
| $\Delta F_{\downarrow}^{\mathrm{X}}$ | Total atmospheric sink of a species X; X being any HFC, PFC or ODS. | 2.7.1 |
| $\Delta\mathrm{RF}^{\mathrm{X}}$ | Radiative forcing induced by a species X; X being any HFC, PFC or ODS. | 2.7.2 |
| $\Delta\mathrm{RF}^{\mathrm{halo}}$ | Radiative forcing induced by all the halogenated compounds combined. | 2.7.2 |
| $\Delta\mathrm{O3t}$ | Tropospheric ozone burden. | 2.8.1 |
| $\Delta\mathrm{RF}^{\mathrm{O3t}}$ | Radiative forcing induced by tropospheric ozone. | 2.8.1 |
| $\Delta\mathrm{EESC}$ | Equivalent effective stratospheric chlorine. | 2.8.2 |
| $\Delta\mathrm{O3s}$ | Stratospheric ozone burden. | 2.8.2 |
| $\Delta\mathrm{RF}^{\mathrm{O3s}}$ | Radiative forcing induced by stratospheric ozone. | 2.8.2 |



| Notation | Name | Section |
|---|---|---|
| $\Delta SO4$ | Atmospheric burden of sulphate aerosols. | 2.9.1 |
| $\Delta POA$ | Atmospheric burden of primary organic aerosols. | 2.9.1 |
| $\Delta BC$ | Atmospheric burden of black carbon aerosols. | 2.9.1 |
| $\Delta NO3$ | Atmospheric burden of nitrate aerosols. | 2.9.1 |
| $\Delta SOA$ | Atmospheric burden of secondary organic aerosols. | 2.9.1 |
| $\Delta RF^{Y}$ | Direct radiative forcing induced by an aerosol Y; Y being $SO_4$, POA, BC, $NO_3$ or SOA. | 2.9.1 |
| $\Delta AER_{sol}$ | Atmospheric burden of soluble aerosols. | 2.9.2 |
| $\Delta RF^{cloud}$ | Radiative forcing induced by the semi-direct and indirect effects of aerosols. | 2.9.2 |
| $\Delta RF^{BCsnow}$ | Radiative forcing induced by black carbon deposition on snow. | 2.10.1 |
| $\Delta RF^{LCC}$ | Radiative forcing induced by albedo change from land-cover change. | 2.10.2 |
| $\Delta RF^{WMGHG}$ | Radiative forcing induced by all well-mixed greenhouse gases combined. | 2.11.1 |
| $\Delta RF^{NTCF}$ | Radiative forcing induced by all near-term climate forcers combined. | 2.11.1 |
| $\Delta RF$ | Total radiative forcing. | 2.11.1 |
| $\Delta RF_{warm}$ | Total radiative forcing accounting for the forcings' efficacies. | 2.11.1 |
| $\Delta RF_{atm}$ | Total radiative forcing occuring within the atmosphere. | 2.11.1 |
| $\Delta T_S$ | Sea surface temperature. | 2.11.2 |
| $\Delta T_L^i$ | Local surface temperature; in region $i$. | 2.11.2 |
| $\Delta P_G$ | Global yearly precipitations. | 2.11.3 |
| $\Delta P_L^i$ | Local yearly precipitations; in region $i$. | 2.11.3 |



**Table 3.** List of driving datasets and parameterizations for the probabilistic setup of the model. The '#' column shows how many options are available for the given parameter or set of parameters. Superscripts are omitted for clarity.

| Drivers | Description | # | Section |
|---|---|---|---|
| $E_{\mathrm{FF}}$ | Emissions of carbon dioxide from fossil-fuel burning and industry. | 2 | 2.2.1 |
| $E_{\mathrm{CH4}}$ | Emissions of methane. | 3 | 2.2.1 |
| $E_{\mathrm{N2O}}$ | Emissions of nitrous oxide. | 2 | 2.2.1 |
| $\{E_{\mathrm{X}}\}_{\mathrm{X}\in\{\mathrm{HFC}\}\cup\{\mathrm{PFC}\}\cup\{\mathrm{ODS}\}}$ | Emissions of halogenated compounds. | 1 | 2.2.1 |
| $E_{\mathrm{NOx}}$ | Emissions of nitrogen oxides. | 2 | 2.2.1 |
| $E_{\mathrm{CO}}$ | Emissions of carbon monoxide. | 2 | 2.2.1 |
| $E_{\mathrm{VOC}}$ | Emissions of non-methane volatile organic compounds. | 2 | 2.2.1 |
| $E_{\mathrm{SO2}}$ | Emissions of sulfur dioxide. | 2 | 2.2.1 |
| $E_{\mathrm{NH3}}$ | Emissions of ammonia. | 2 | 2.2.1 |
| $E_{\mathrm{OC}}$ | Emissions of organic carbon. | 1 | 2.2.1 |
| $E_{\mathrm{BC}}$ | Emissions of black carbon. | 1 | 2.2.1 |
| $\delta A; \delta H; \delta S; A_0$ | Land-use and land-cover change drivers & preindustrial land-cover. | 1[§] | 2.2.2 |
| $\mathrm{RF}_{\mathrm{con}}$ | Additional anthropogenic radiative forcing. | 1 | 2.2.3 |
| $\mathrm{RF}_{\mathrm{volc}}; \mathrm{RF}_{\mathrm{solar}}$ | Additional natural radiative forcings. | 1 | 2.2.3 |

| Parameters | Description | # | Section |
|---|---|---|---|
| $\nu_{\mathrm{fg}}; A_{\mathrm{ocean}}; h_{\mathrm{mld},0}; \pi_{\mathrm{circ}}; \tau_{\mathrm{circ}}; T_{S,0}$ | Structural parameters of the oceanic carbon-cycle. | 4 | 2.3.1 |
| $\{\alpha_{\mathrm{atm}}^{\mathrm{X}}\}_{\mathrm{X}\in\{\mathrm{WMGHG}\}}$ | Atmospheric conversion factors for well-mixed greenhouse gases. | 1 | 2.3.1* |
| $\mathcal{F}_{\mathrm{pCO2}}$ | *Ad hoc* function to emulate the oceanic carbonate chemistry. | 2 | 2.3.1 |
| $\alpha_{\mathrm{sol}}$ | Conversion factor for dissolved inorganic carbon. | 1 | 2.3.1 |
| $\pi_{\mathrm{mld}}; \gamma_{\mathrm{mld}}$ | Transient response of the oceanic stratification. | 3 | 2.3.1 |
| $\eta; \mu; \rho_{\mathrm{litt}}; \rho_{\mathrm{soil}}$ | Preindustrial equilibrium of the land carbon-cycle excluding wildfires. | 9 | 2.3.2 |
| $\mathcal{F}_{\mathrm{fert}}$ | Functional form of the fertilisation function. | 2 | 2.3.2 |
| $\beta_{\mathrm{npp}}; \tilde{\beta}_{\mathrm{npp}}; \mathrm{CO2}_{\mathrm{cp}}; \gamma_{\mathrm{npp},T}; \gamma_{\mathrm{npp},P};$ $\gamma_{\mathrm{resp},T}; \gamma_{\mathrm{resp},P}; \gamma_{\mathrm{resp},T_1}; \gamma_{\mathrm{resp},T_2}; \tilde{\gamma}_{\mathrm{resp},P}$ | Transient response of the land carbon-cycle excluding wildfires. | 7 | 2.3.2 |
| $\iota$ | Preindustrial intensity of wildfires. | 7 | 2.3.2 |
| $\gamma_{\mathrm{igni},C}; \gamma_{\mathrm{igni},T}; \gamma_{\mathrm{igni},P}$ | Transient response of the wildfires. | 5 | 2.3.2 |
| $\mathcal{F}_{\mathrm{resp}}$ | Functional form of the respiration function. | 2 | 2.3.2 |
| $\kappa_{\mathrm{met}}$ | Factor for litter-to-soil carbon flux. | 1 | 2.3.2 |
| $A_0; \pi_{\mathrm{wet}}$ | Preindustrial natural land-cover. | 13[§,‖] | 2.3.2 |
| $\tau_{\mathrm{shift}}; \pi_{\mathrm{shift}}$ | Turnover time of shifting cultivation and associated biomass fraction. | 1 | 2.3.3 |
| $\pi_{\mathrm{agb}}$ | Above-ground biomass fraction. | 3 | 2.3.3 |
| $\pi_{\mathrm{hwp}}$ | Allocation coefficients for the harvested wood products. | 2 | 2.3.3 |
| $\tau_{\mathrm{hwp}}$ | Turnover times of the harvested wood products. | 2 | 2.3.3 |
| $\mathcal{F}_{\mathrm{hwp}}; \mathring{\mathcal{C}}$ | Functional form for the wood product oxidation and associated profile. | 3 | 2.3.3 |





| Parameters | Description | # | Section |
|---|---|---|---|
| $\{\alpha_{\mathrm{rf}}^{\mathrm{X}}\}_{\mathrm{X}\in\{\mathrm{WMGHG}\}}$ | Radiative efficiency of well-mixed greenhouse gases. | 1 | 2.3.4* |
| $\{X_0\}_{\mathrm{X}\in\{\mathrm{WMGHG}\}}$ | Preindustrial atmospheric concentration of well-mixed greenhouse gases. | 1 | 2.3.4* |
| $\alpha_{\mathrm{bb}}$ | Proportionnality factors for biomass burning. | 1 | 2.4.1 |
| $\tau_{\mathrm{lag}}$ | Time-lag used to estimate the lagged concentrations. | 1 | 2.4.2 |
| $\tau_{\mathrm{OH}}^{\mathrm{CH4}}$ | Preindustrial lifetime of methane for the OH sink. | 16 | 2.5.1 |
| $\tau_{\mathrm{h}\nu}^{\mathrm{CH4}};\tau_{\mathrm{soil}}^{\mathrm{CH4}};\tau_{\mathrm{ocean}}^{\mathrm{CH4}}$ | Preindustrial lifetime of methane for other sinks. | 1 | 2.5.1 |
| $\xi_{\mathrm{CH4}}^{\mathrm{OH}};\xi_{\mathrm{O3s}}^{\mathrm{OH}};\xi_{T_A}^{\mathrm{OH}};\xi_{Q_A}^{\mathrm{OH}};$ $\xi_{\mathrm{NOx}}^{\mathrm{OH}};\xi_{\mathrm{CO}}^{\mathrm{OH}};\xi_{\mathrm{VOC}}^{\mathrm{OH}};\tilde{\xi}_{\mathrm{NOx}}^{\mathrm{OH}};\tilde{\xi}_{\mathrm{CO}}^{\mathrm{OH}};\tilde{\xi}_{\mathrm{VOC}}^{\mathrm{OH}}$ | Transient response of the OH tropospheric chemistry. | 5 | 2.5.1 |
| $\mathcal{F}_{\mathrm{prec}}$ | Functional form of the OH sink response to ozone precursors function. | 2 | 2.5.1 |
| $\kappa_{T_A};\kappa_{Q_A};\kappa_{\mathrm{svp}};T_{\mathrm{svp}};T_{A;0};$ $\mathrm{O3s}_0;E_{\mathrm{nat}}^{\mathrm{NOx}};E_{\mathrm{nat}}^{\mathrm{CO}};E_{\mathrm{nat}}^{\mathrm{VOC}}$ | Other parameters for the response of the OH atmospheric chemistry. | 1 | 2.5.1 |
| $A_{\mathrm{wet};0};e_{\mathrm{wet};0};\pi_{\mathrm{wet}}$ | Preindustrial emissions and area extent of wetlands. | 8$^{\parallel}$ | 2.5.2 |
| $\gamma_{\mathrm{wet},C};\gamma_{\mathrm{wet},T};\gamma_{\mathrm{wet},P}$ | Transient response of the wetlands area extent. | 7 | 2.5.2 |
| $\mathcal{F}_{\mathrm{over}}$ | *Ad hoc* function for the overlap of the absorption bands. | 1 | 2.5.3* |
| $\tau_{\mathrm{h}\nu}^{\mathrm{N2O}}$ | Preindustrial lifetime of nitrous oxide for the stratospheric sink. | 9 | 2.6.1 |
| $\xi_{\mathrm{N2O}}^{\mathrm{h}\nu};\xi_{\mathrm{EESC}}^{\mathrm{h}\nu};\xi_{\mathrm{age}}^{\mathrm{h}\nu}$ | Transient response of the stratospheric chemistry. | 5 | 2.6.1 |
| $\gamma_{\mathrm{age}}$ | Transient response of the stratospheric age of air. | 7 | 2.6.1 |
| $\{\tau_{\mathrm{OH}}^{\mathrm{X}};\tau_{\mathrm{h}\nu}^{\mathrm{X}};\tau_{\mathrm{othr}}^{\mathrm{X}}\}_{\mathrm{X}\in\{\mathrm{HFC}\}\cup\{\mathrm{PFC}\}\cup\{\mathrm{ODS}\}}$ | Preindustrial lifetimes of halogenated compounds for various sinks. | 1 | 2.7.1 |
| $\xi_{\mathrm{CH4}}^{\mathrm{O3t}};\xi_{\mathrm{NOx}}^{\mathrm{O3t}};\xi_{\mathrm{CO}}^{\mathrm{O3t}};\xi_{\mathrm{VOC}}^{\mathrm{O3t}}$ | Transient response of tropospheric ozone to methane and its precursors. | 5 | 2.8.1 |
| $\omega_{\mathrm{NOx}};\omega_{\mathrm{CO}};\omega_{\mathrm{VOC}}$ | Regionalization of the tropospheric ozone chemistry. | 12 | 2.8.1 |
| $\pi_{\mathrm{reg}}$ | Matrix describing the overlap of the different regional aggregations. | – | 2.8.1* |
| $\Gamma_{\mathrm{O3t}}$ | Transient response of tropospheric ozone to climate change. | 9 | 2.8.1 |
| $\alpha_{\mathrm{rf}}^{\mathrm{O3t}}$ | Radiative efficiency of tropospheric ozone. | 17 | 2.8.1 |
| $\{\pi_{\mathrm{rel}}^{\mathrm{X}}\}_{\mathrm{X}\in\{\mathrm{ODS}\}};\xi_{\mathrm{N2O}}^{\mathrm{O3s}}$ | Fractional release factors of each ozone depleting substance. | 3$^{\P}$ | 2.8.2 |
| $\{n_{\mathrm{Cl}}^{\mathrm{X}};n_{\mathrm{Br}}^{\mathrm{X}}\}_{\mathrm{X}\in\{\mathrm{ODS}\}}$ | Number of chlorine and bromine atoms per ozone depleting substance. | – | 2.8.2 |
| $\alpha_{\mathrm{Cl}}^{\mathrm{Br}}$ | Relative efficiency in destroying ozone of bromine over chlorine. | 1 | 2.8.2 |
| $\xi_{\mathrm{EESC}}^{\mathrm{O3s}};\xi_{\mathrm{N2O}}^{\mathrm{O3s}}$ | Transient response of stratospheric ozone to stratospheric chlorine. | 11$^{\P}$ | 2.8.2 |
| $\alpha_{\mathrm{N2O}}^{\mathrm{EESC}};\mathrm{EESC}_{\times};\xi_{\mathrm{N2O}}^{\mathrm{O3s}}$ | Sensitivity of stratospheric ozone to nitrous oxide. | 1$^{\P}$ | 2.8.2 |
| $\Gamma_{\mathrm{O3s}}$ | Transient response of stratospheric ozone to climate change. | 5 | 2.8.2 |
| $\alpha_{\mathrm{rf}}^{\mathrm{O3s}}$ | Radiative efficiency of stratospheric ozone. | 5 | 2.8.2 |
| $\tau_{\mathrm{SO2}};\tau_{\mathrm{DMS}};\Gamma_{\mathrm{SO4}}$ | Transient response of the sulphate aerosols chemistry. | 4 | 2.9.1 |
| $\omega_{\mathrm{SO2}}$ | Regionalization of the sulphate aerosols chemistry. | 8 | 2.9.1 |
| $\tau_{\mathrm{OM,ff}};\tau_{\mathrm{OM,bb}};\Gamma_{\mathrm{POA}}$ | Transient response of the primary organic aerosols chemistry. | 4 | 2.9.1 |
| $\omega_{\mathrm{OM}}$ | Regionalization of the primary organic aerosols chemistry. | 8 | 2.9.1 |
| $\alpha_{\mathrm{OM}}^{\mathrm{OC}}$ | Conversion factor for organic matter. | 3 | 2.9.1 |
| $\tau_{\mathrm{BCff}};\tau_{\mathrm{BC,bb}};\Gamma_{\mathrm{BC}}$ | Transient response of the black carbon aerosols chemistry. | 4 | 2.9.1 |
| $\omega_{\mathrm{BC}}$ | Regionalization of the black carbon aerosols chemistry. | 8 | 2.9.1 |


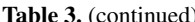



| Parameters | Description | # | Section |
|---|---|---|---|
| $\tau_{\mathrm{NOx}}; \tau_{\mathrm{NH3}}; \Gamma_{\mathrm{NO3}}$ | Transient response of the nitrate aerosols chemistry. | 2 | 2.9.1 |
| $\tau_{\mathrm{VOC}}; \tau_{\mathrm{BVOC}}; \Gamma_{\mathrm{SOA}}$ | Transient response of the secondary organic aerosols chemistry. | 3 | 2.9.1 |
| $\alpha_{\mathrm{rf}}^{\mathrm{SO4}}$ | Radiative efficiency of sulphate aerosols. | 15 | 2.9.1 |
| $\alpha_{\mathrm{rf}}^{\mathrm{POA}}$ | Radiative efficiency of primary organic aerosols. | 15 | 2.9.1 |
| $\alpha_{\mathrm{rf}}^{\mathrm{BC}}$ | Radiative efficiency of black carbon aerosols. | 15 | 2.9.1 |
| $\alpha_{\mathrm{rf}}^{\mathrm{NO3}}$ | Radiative efficiency of nitrate aerosols. | 8 | 2.9.1 |
| $\alpha_{\mathrm{rf}}^{\mathrm{SOA}}$ | Radiative efficiency of secondary organic aerosols. | 5 | 2.9.1 |
| $\kappa_{\mathrm{adj}}^{\mathrm{BC}}$ | Factor of the semi-direct effect of black carbon. | 6 | 2.9.2 |
| $\{\pi_{\mathrm{sol}}^{\mathrm{Y}}\}_{\mathrm{Y}\in\{\mathrm{SO4,POA,BC,NO3,SOA}\}}$ | Soluble fractions of each aerosol. | 2 | 2.9.2 |
| $\Phi; \mathrm{AER}_{\mathrm{sol},0}$ | Parameters to model the indirect effects of aerosols. | 7† | 2.9.2 |
| $\mathrm{AER}_{\mathrm{sol},0}$ | Preindustrial burden of soluble aerosols. | 3† | 2.9.2 |
| $\omega_{\mathrm{BCsnow}}$ | Regionalization of the deposition of black carbon on snow. | 1 | 2.10.1 |
| $\alpha_{\mathrm{rf}}^{\mathrm{BCsnow}}$ | Radiative efficiency of black carbon on snow with respect to emissions. | 8 | 2.10.1 |
| $\pi_{\mathrm{trans}}$ | Global short-wave and upward transmittance. | 1 | 2.10.2 |
| $\varphi_{\mathrm{rsds}}; \alpha_{\mathrm{alb}}$ | Climatology of radiative short-wave and downward flux at the surface. | 2‡ | 2.10.2 |
| $\alpha_{\mathrm{alb}}$ | Climatology of land surface albedo. | 2‡ | 2.10.2 |
| $\alpha_{\mathrm{alb}}$ | Climatology of land-cover. | 2‡ | 2.10.2 |
| $A_{\mathrm{Earth}}$ | Surface area of the Earth. | – | 2.10.2* |
| $\kappa_{\mathrm{warm}}^{\mathrm{BCsnow}}$ | Warming efficacy of black carbon on snow. | 3 | 2.11.1 |
| $\kappa_{\mathrm{warm}}^{\mathrm{LCC}}$ | Warming efficacy of the albedo effect of land-cover change. | 4 | 2.11.1 |
| $\kappa_{\mathrm{warm}}^{\mathrm{volc}}$ | Warming efficacy of volcanic aerosols. | 1 | 2.11.1 |
| $\pi_{\mathrm{atm}}^{\mathrm{CO2}}; \pi_{\mathrm{atm}}^{\mathrm{noCO2}}; \pi_{\mathrm{atm}}^{\mathrm{O3t}}; \pi_{\mathrm{atm}}^{\mathrm{strat}};$ $\pi_{\mathrm{atm}}^{\mathrm{scatter}}; \pi_{\mathrm{atm}}^{\mathrm{absorb}}; \pi_{\mathrm{atm}}^{\mathrm{cloud}}; \pi_{\mathrm{atm}}^{\mathrm{alb}}; \pi_{\mathrm{atm}}^{\mathrm{solar}}$ | Atmospheric fraction of radiative forcing for various forcers. | 2 | 2.11.1 |
| $\lambda; \tau_{T_G}; \tau_{T_D}; \theta$ | Climate sensitivity & global surface temperature dynamics. | 25 | 2.11.2 |
| $\omega_{T_S}; \omega_{T_L}$ | Pattern scaling of the temperature response. | 2 | 2.11.2 |
| $\alpha_{P_G}; \beta_{P_G}$ | Global precipitations response. | 25 | 2.11.3 |
| $\omega_{P_L}$ | Pattern scaling of the precipitations response. | 2 | 2.11.3 |
| $\pi_{\mathrm{ohc}}$ | Fraction of extra energy used to heat the ocean. | 1 | 2.11.4 |

| Total available parameterizations | | |
|---|---|---|
| – excluding driving datasets | $\gtrsim 10^{41}$ | |
| – including driving datasets | $\gtrsim 10^{43}$ | |

\* First mention of the parameter in this section.

§ The preindustrial land-cover ($A_0$) is determined by these two options.

‖ The wetlands partition coefficients ($\pi_{\mathrm{wet}}$) are determined by these two options.

¶ The sensitivity of stratospheric ozone to nitrous oxide ($\xi_{\mathrm{N2O}}^{\mathrm{O3s}}$) is determined by these three options.

† The preindustrial burden of hydrophilic aerosols ($\mathrm{AER}_{\mathrm{sol},0}$) is determined by these two options.

‡ The land albedoes ($\alpha_{\mathrm{alb}}$) are determined by these three options.



**Figure 1.** Simplified causal chain of OSCAR v2.2. Each node of the graph corresponds to a module described in the section whose number is shown below the node's name. Colored edges show the forcings of the model, black edges show the natural cause-effect chain, and dashed edges show the climate feedbacks. "Halo" groups all the halogenated compounds; "Ocean" is the ocean carbon-cycle, "Land" is the land carbon-cycle; "Albedo" groups the surface albedo effects; "hν" is the stratospheric chemistry; "OH" is the tropospheric chemistry; "Cloud" is the indirect aerosol effect; "Climate" groups the surface temperatures and precipitations.





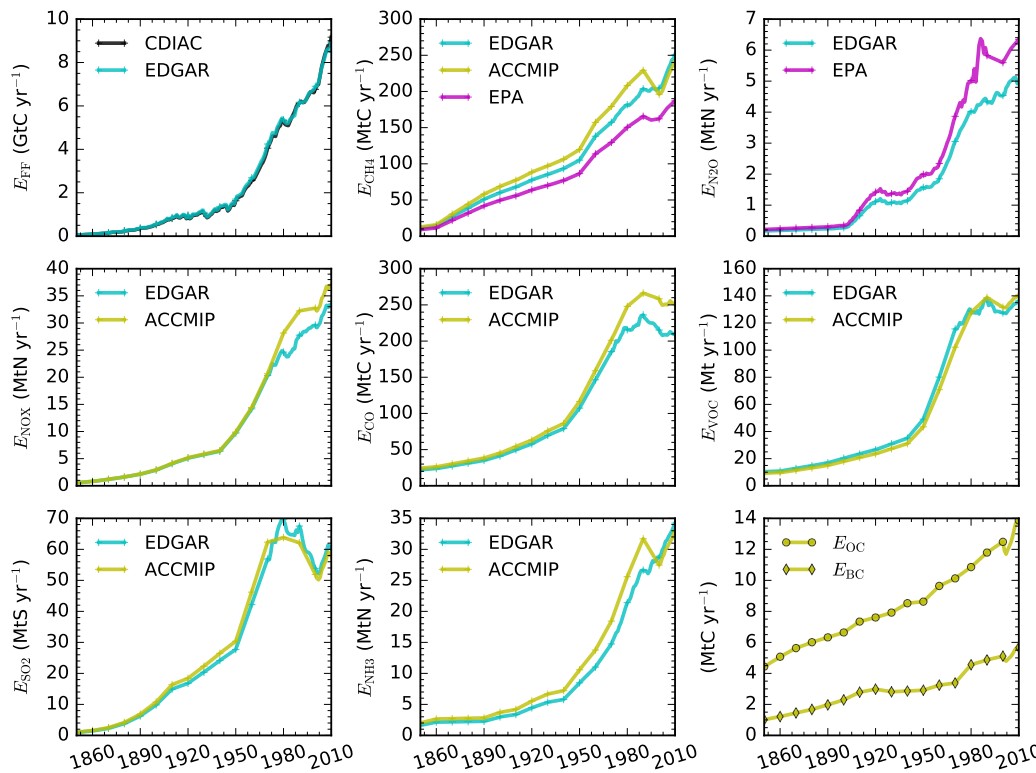

**Figure 2.** Time-series of the main anthropogenic emissions used as potential inputs of OSCAR (section 2.2.1). Other drivers of the model, i.e. emissions of halogenated compounds and LULCC, are shown in figure S1 and S2, respectively.





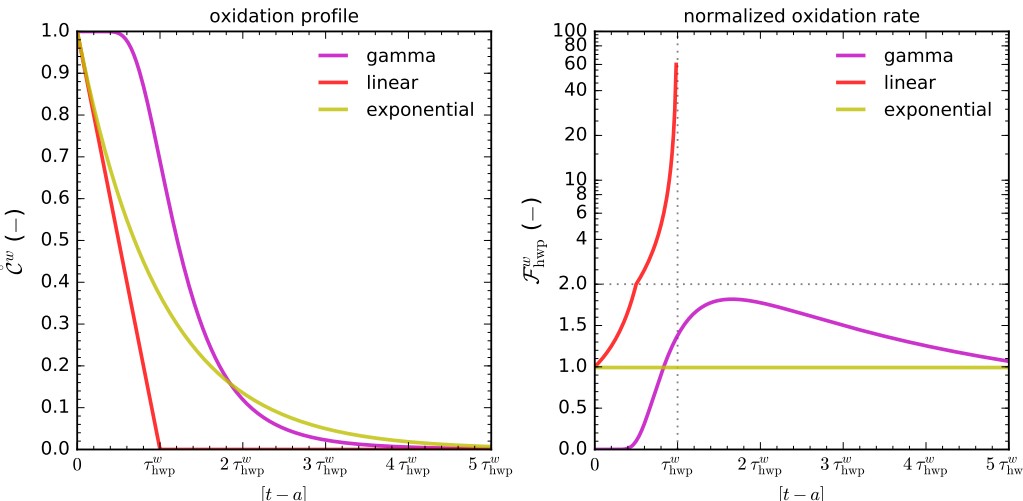

**Figure 3.** Functional forms possible for the harvested wood products oxidation (section 2.3.3). They are shown as the oxidation profile of a unit pool of wood product (left-hand panel) and as the corresponding normalized yearly oxidation rate (right-hand panel). The former is noted $\mathring{\mathcal{C}}$ and the latter is exactly the function $\mathcal{F}_{\mathrm{hwp}}$. They are linked by the following relationship: $\mathcal{F}_{\mathrm{hwp}}^{w} = -\tau_{\mathrm{hwp}}^{w} \frac{\mathrm{d}}{\mathrm{d}t} \ln[\mathring{\mathcal{C}}^{w}]$.





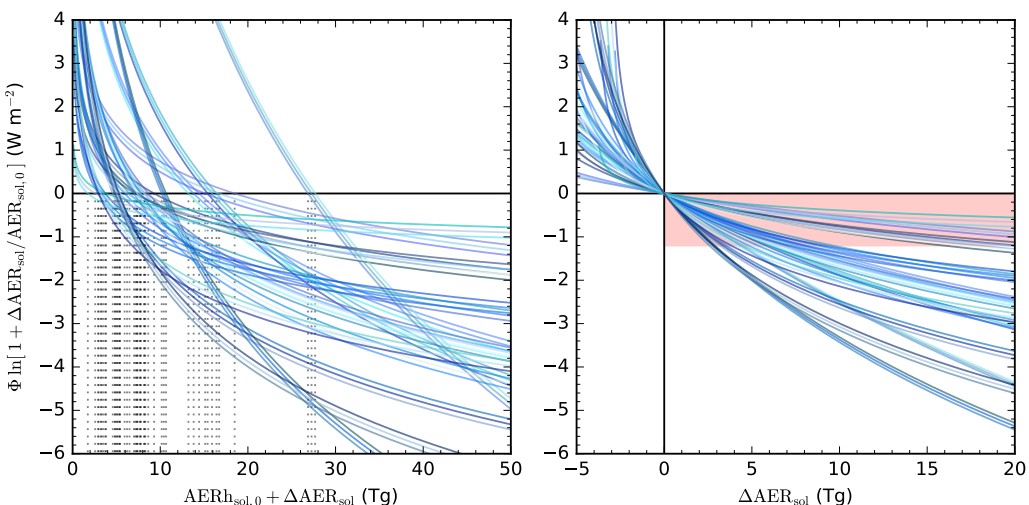

**Figure 4.** Ensemble of possible parameterizations of the aerosol-cloud interactions in OSCAR (section 2.9.2). Here we show the simulated radiative forcing as a function of the total burden of soluble aerosols (left-hand side) or of the change in that burden since preindustrial (right-hand side). In the former case, the grey dotted lines show the preindustrial burden we calculate; in the latter, the red area shows the 90% range of RF provided by Myhre et al. (2013b), and therefore the associated change in burden implied by our formula.





**Figure 5.** Results of our simulations with OSCAR, for carbon dioxide. The offline simulation is shown in blue, and the online simulation in black. Other colors are references we compare our results to. The left-hand panels show the timeseries from 1900 to 2010, the thick colored lines indicate the median of the ensemble of simulations, and the colored area its 5th to 95th percentiles. The right-hand panels show the probability distribution function (PDF) from the ensemble of simulations, for the averaged last 10 years of simulation. Reference for the first three fluxes is the GCP (Le Quéré et al., 2015), and the dashed red lines show the 90% uncertainty range (calculated as 1.645 times the $1\sigma$-uncertainty). Reference 1 for the atmospheric growth rate and concentration is NOAA/ESRL (Tans and Keeling, 2015). Reference 2 are Law Dome ice-cores (Etheridge et al., 1996; MacFarling Meure et al., 2006).





**Figure 6.** Results of our simulations with OSCAR, for methane; with the same format as per carbon dioxide. References are: ACCMIP (Lamarque et al., 2010) for biomass burning; WETCHIMP (Melton et al., 2013) for wetlands; Prather et al. (2012) for the lifetime and 90% uncertainty range (calculated as 1.645 times the $1\sigma$ uncertainty); and AGAGE (Prinn et al., 2013) for the atmospheric growth rate and concentration. Reference 2 are Law Dome ice-cores (Etheridge et al., 1998; MacFarling Meure et al., 2006, using the NOAA04 scale).





**Figure 7.** Results of our simulations with OSCAR, for nitrous oxide; with the same format as per carbon dioxide. References are: Prather et al. (2015) for the lifetime and 90% uncertainty range (calculated as 1.645 times the $1\sigma$ uncertainty); and AGAGE (Prinn et al., 2013) for the atmospheric growth rate and concentration. Reference 2 are Law Dome ice-cores (MacFarling Meure et al., 2006).



**Figure 8.** Results of our simulations with OSCAR, for halogenated compounds; with the same format as per carbon dioxide. Reference for the atmospheric concentrations is IPCC (2013); for the EESC it is the same concentrations combined with the fractional release values of Newman et al. (2007); and for the radiative forcing and its 90% uncertainty range it is IPCC (Myhre et al., 2013b).





**Figure 9.** Results of our simulations with OSCAR, for ozone; with the same format as per carbon dioxide. Reference for the global burden is IPCC (2013); and for the radiative forcing and its 90% uncertainty range it is IPCC (Myhre et al., 2013b).





**Figure 10.** Results of our simulations with OSCAR, for aerosols; with the same format as per carbon dioxide. Reference 1 for the radiative forcing and its 90% uncertainty range is IPCC (Myhre et al., 2013b). Reference 2 is the same except that contribution from biomass burning aerosols is removed.





**Figure 11.** Results of our simulations with OSCAR, for radiative forcing; with the same format as per carbon dioxide. Reference for the radiative forcing and its 90% uncertainty range is IPCC (Myhre et al., 2013b)







**Figure 12.** Results of our simulations with OSCAR, for climate; with the same format as per carbon dioxide. Reference 1 is Had-CRUT4 (Morice et al., 2012) for global surface temperature, and HadISST1 (Rayner et al., 2003) for sea surface temperature. Reference 2 is NOAA/NCDC (Smith et al., 2008) for global surface temperature, ERSST4 (Huang et al., 2015) for sea surface temperature, and NOAA/NODC (Levitus et al., 2012) for ocean heat content. Reference 3 is GISTEMP (Hansen et al., 2010) for global surface temperature.