# Peer review of "The compact Earth system model OSCAR v2.2: description and first results"

_Geoscientific Model Development, 2016_

## Referee Comment (RC1) · Anonymous Referee #1 · 22 Sep 2016

My apologies for taking so long to get my comments in. This is a very well written paper that carefully and meticulously documents the new parameterized ESM called OSCAR. I have some minor issues outlined below, but overall, I would be happy to use this model in any of my applications where it is needed. OSCAR rests clearly on the analyses of the full chemistry-carbon-climate models and these sources are carefully documented here. Intro is OK, but (p.2) is OSCAR just another box model? Please say so, describing it as? Parameterized, multicomponent ESM without a model grid? I like the idea of the regional boxes being used to account for the heterogeneity of the global forcing/response. Whatever, but get this upfront. I admit that I did not proof all of the equations here, but rather looked at structural problems and inclusiveness. A key question (answered for me at the end) was is there a source of interannual variability and climate chaos? Of ENSO? in OSCAR. Apparently not – but if it could be

implemented, how would it change the analyses and ensembles?

Specific Comments:

OSCAR = "Earth system change" model." Excellent.

p.3 "but all other sectors provided by the inventories are accounted for" – does this include aviation and shipping. OK, 2.3.3 explains part of this.

p.7 "nutrient deposition – albeit the latter not in this version of OSCAR." OK, I would have liked to see this included and am glad it is clearly stated. Perhaps a table of links and processes with a yes/no/partial would be useful to scan?

p.12 "the migration of natural biomes caused by changes in environmental conditions (e.g. Jones et al., 2009). This is however not included in this version of OSCAR." Another item for the table of processes.

p.13 "disturbe" >disturb – surprising, minor. Overall this paper is very well and clearly written.

p.18 "it is parameterized by three relative chemical sensitivities (-) and the preindustrial natural emissions of the three ozone precursors (ENOxnat , ECOnat , EVOCnat )" OK, but worrisome, would clearly like to have a latitudinally dependent impact of the NOx and VOC emissions at least (probably not CO). for example,

"All the chemical sensitivities of the OH sink (i.e. _OH CH4, _OH O3s, _OH TA , _OH QA , _OH NOx, _OH CO, _OH VOC, _OH NOx, _OH CO and _OH VOC) are taken as one of the four sets of values from the study by Holmes et al. (2013, table 2)" This is really fine, but again ignores the NOx vs Latitude reactivity (Wild et al, 2001, Indirect long-term global cooling from NOx emissions, Geophys. Res. Lett.)

p.19 "This implicitly assumes that all the natural sources of methane but natural wetlands remain unchanged since the preindustrial" – having trouble with the 'but', do you mean 'and'??

p.20 Nice job with the complex system that is N2O.

p.22 "in this case the associated lifetime is set to a value of infinity" This is a typical problem with using lifetimes instead of loss frequencies. I would have made all of the tau's into 1/tau = LF which can then be set to zero (inifinity is hard with a computer....)

p.23 "at chemical equilibrium with" – I think equilibrium is really abused by many of our colleagues, it is rather a steady-state that is reached. Equilibrium has deeper implications of detailed balance as in thermodynamics.

"with linear global sensitivities (...) that are regionalized thanks to region-specific weights..." This is very nicely done since that are large regional differences in both the chemical and RF response to these short-lived emissions!

p.25 The param for O3s is reasonable, but I am not sure that I would agree with the N2O impact being reduced at high EESC (eqn 61). The N2O loss occurs in a very different region from either Cly losses (lower strat or 40+km for ClO+O) and should simply be just linear?

p.26 I am worried about the formulation of eqns like 63 because if loss frequencies go to zero then Tau goes to infinity (and stated earlier) and this formula blows up. This should be written in the form like 1 / ( 1/tau + min.loss.freq )

p.35 "The differential system is solved with the forward Euler method (Euler, 1768) with a time-step (_t) that can be chosen before any simulation with OSCAR – although time-steps greater than a quarter of year systematically make the model diverge. This time-step is usually set to _t = 1/6 yr". Since these equations do not appear to be very stiff (you can get away with 1/6 yr) you might want to use a Bulirsch-Stoer high accuracy integrator (not much cost) with a 1-yr overall time step (it divides the large step into a nested sequence of explicit steps and then extrapolates to give you an almost perfect answer. Note that B-S does not help with very stiff equations like integration O(1D) at sunrise.

"more than 1043 potential combinations of parameters" – how about 2\*\*128 = 10\*\*43 Are there really 128 independent parameters in Table 2?

p.36 and elsewhere. "The interannual variability of the land sink simulated by OSCAR in the offline case does not match that from" – I am a bit confused as to how you implement interannual variability and climate modes in OSCAR. DO not volcanoes affect the C-cycle thru T? if not diffuse radiation.

p.37ff – Nice discussion and example with CH4.

p.38 – When comparing the 'online' to 'offline' CH4 simulations it might be useful to remind people that the difference between the two, because nominally these two designations describe computational differences rather than models not having feedbacks. Same for N2O in the following.

p.41 "the lack of interannual variability in OSCAR..." OK, now it is clear, but a discussion of this could be upfront – maybe I missed it.

p.43 "These many degrees of freedom increase the odds of seeing a given simulation diverge, or at least depart unreasonably from the plausible range of results." I am not sure this is true. Unless the equations are chaotic (e.g., Lorenz N-cycle model, Liapunov, ...) then divergence would not seem possible in a linearized model.

p.43 "Also, coupling of the tropospheric and stratospheric chemistries would be an improvement, especially for ozone, as would a finer regionalization be. We note however that a tremendous amount of factorial simulations by complex chemistry transport models would be needed to make such an improvement." Since the goal is to accomplish the cross-coupling and feedbacks, the N2O-CH4 link (Prather & Hsu, 2010 Science) would seem to be important (+10 molec of N2O => -3.6 molec of CH4). Is this feedback actually included in the strat-trop chemistry of OSCAR – if so great, and note it.

p. 43-44. Future or current improvements – I would vote for 2 as primary to simulating climate and to understanding the coupling across cycles. 1) Find a way to do ensembles with climate variability on interannual to decadal scales. 2) Do an eigenvalue analysis of your linearized matrix to identify time scales and the structure of the major coupled modes. 3) Add a table 4 that lists the processes and known couplings that are NOT included; this of course cannot be complete but at least some of the major areas that you chose not to include. Very useful as a reminder. 4) Describe how to go about updating the parameters when we have new results from the 'big' models.

p.63 Great figure, thanks. Minor fix: 'edges' in the figure caption should read 'lines'

---

## Referee Comment (RC2) · Anonymous Referee #2 · 10 Oct 2016

This paper is an extremely well written and thorough description of the OSCAR 2.2 model. The model carries a significant number of innovative approaches that will prove very useful in investigating the range of possible forcings, feedbacks and interactions within the full Earth system that will ultimately determine the global and regional response of this system to anthropogenic emissions of greenhouse gases, aerosol precursors and human-induced land use change.

From this perspective I recommend that the paper is published with only minor revisions for the sake of clarity and brevity. These are indicated below under requested revisions. In addition to this I have a number of general suggestions and comments/questions, which might further improve an already very good paper. The authors might like to consider some of these.

[Figure]

General points. 1. The paper is very long. I realize this is necessary to provide the level of detail required for a reader to properly understand the formulation of the model. The sections describing the model components are generally very good and very clear. That said, the section where model simulation results are presented for the historical period is actually quite thin and the weakest part of the paper. In particular, a number of areas where OSCAR deviates significantly from observations, more complex models or IPCC best estimates are not always fully explained or discussed. Also the model is designed, primarily, to allow a probabilistic investigation of future Earth system change. No examples of the application of the model to possible future conditions are included in the paper.

Hence my main suggestion is that the authors consider a high-level restructuring of the paper and instead submit 2 (linked, Part I, Part II) articles, with Part I essentially being the model description part of the submitted article and Part II being (i) an extended version of the present section on the historical period simulation (with some more discussion and explanation of deviations from observations/other models/IPCC estimates and (ii) include an initial example of how the model can/will be applied in the context of investigating future Earth system change. I realize that point (ii) is no doubt intended by the authors in subsequent papers, nevertheless, some brief examples of how the model is to be applied in a future projection sense would be illustrative in the context of the model description paper (my suggested Part I). Furthermore, in the Part I article I would recommend an initial section that gives a very brief overview of the model structure (aimed at non modeling scientists that may still be interested in the model application, e.g. results presented in Part II) along with a note to the effect that readers interested in the model formulation should read all of Part I, while those mainly interested in the model application and results can just read the short description and then jump to Part II.

This suggestion, which is just that (a suggestion), would in my opinion make the combined papers significantly more interesting to a wider audience than the present paper.

If the authors prefer to keep the present single article then I suggest they more carefully discuss/explain some of the key deficiencies in the historical simulation as this section was a little thin in places.

Requested revisions 2. In a general sense I found the approach of emulating/constraining the OSCAR components and parameterizations using more complex model results (e.g. CMIP5/HTAP) or IPCC results, assumed that the reader was already well informed of this type of approach. , i.e. it was not always clear if emulation/constraint was being applied and used across the ensemble of CMIP5 models or rather to a "best fit model vs observations" or an ensemble mean was being used to constrain OSCAR parameters for the historical period. Request revision: A brief and basic description of the emulation approach might help more general readers

3. Given the importance of marine carbon uptake and potential changes in the efficiency of this carbon sink/source in the future, I felt the level of detail describing the marine C cycle compared to the terrestrial C cycle was somewhat unbalanced. Equally, I was surprised that marine C cycle did not include any parameterization of the marine biological pump. While solubility processes may dominate historical and future marine C uptake, there is evidence that changes in the biological pump are likely to play a non-negligible role in future marine carbon uptake. On this note, does the relative accuracy of the ocean carbon uptake for the historical period (shown in figure 5) indicate that the biological pump was largely unimportant in this increased uptake? Or does it suggest the way the model has been constrained implicitly includes a biological component? Requested revision: Some comments on the importance or not of the marine biological pump seems warranted, particularly if the primary application of the model is to investigate future uncertainties in coupled climate-carbon cycle processes.

4. Is ocean acidification and its potential impact on marine carbon uptake included in the model? This was not obvious to me. Requested revision: Please make it clear if yes and if not, as with point 2 above, what are the possible consequences for application to future projections.

5. For certain model parameters: e.g. environmental controls on fire ignition, as an example, it is not clear how many more complex ESMs with interactive fire models were used to determine these parameters. The number of CMIP5 models with interactive fire models was pretty small. Requested revision: Particularly where only a small number of models were available for constraining parameters this should be made clear.

6. With respect to the terrestrial carbon cycle and atmospheric CH4, it seems that permafrost is not included in OSCAR? Is this correct. If so why was this decision taken and, like my comments in marine biology, there is evidence that permafrost melt may be an important future feedback in the Earth system. Requested revision: Omission of this feedback seems like it needs a motivation and acknowledgement of potential projection limitations due to this decision.

Questions (not particularly requiring modifications in the paper unless the authors feel it will help)

7. With respect to surface temperature and precipitation changes (pages 33-34) the global climate sensitivity ($\lambda$) plays an important role. This is derived from CMIP5 abrupt 4xCO2 and pre-industrial control simulations. $\lambda$ includes all "fast" climate feedbacks such as water vapour and cloud feedbacks. My question relates to the definition of cloud-aerosol effects in OSCAR, these seem to be potentially decoupled from (future) cloud changes, with the latter defined through $\lambda$. As future cloud aerosol effects will be mediated by any future changes in the distribution of fractional cloud and cloud microphysical properties, is there some risk that future cloud aerosol impacts may be inaccurate due to this decoupling?

8. A similar question arises with respect to the calculation of precipitation and in particular, the regional weights for precipitation. How are cloud-aerosol changes on regional precipitation included, if at all, in OSCAR?

9. It is stated that the model is primarily used for annual mean or longer analysis. This is understandable given the time step and basic aims of the paper. My question

is whether, in an approach analogous to statistical downscaling which brings an increased spatial dimension to coarse spatial resolution data can something similar be done in an effort to infer higher time frequency changes based on the annual mean timescale changes?

---

## Author Comment (AC3) · 7 Dec 2016

[revised manuscript text omitted]

& + \sum_b \alpha_{\mathrm{bb}}^{\mathrm{X},i,b} \left( e_{\mathrm{fire},0}^{i,b} \, \Delta A^{i,b} + \Delta e_{\mathrm{fire}}^{i,b} \, A_0^{i,b} + \Delta e_{\mathrm{fire}}^{i,b} \, \Delta A^{i,b} \right) \\
& + \sum_{b,a} \alpha_{\mathrm{bb}}^{\mathrm{X},i,b} \left( \iota_0^{i,b} \, \mathrm{f}_{\mathrm{igni}}^{i,b} \, \Delta C_{\mathrm{veg,luc}}^{i,b,a} - \frac{\mathrm{d}}{\mathrm{d}t} \Delta C_{\mathrm{hwp,luc}}^{w=1,i,b,a} \right).
\end{aligned}
\tag{38}
$$

The $\alpha_{\mathrm{bb}}$ parameters come from the GFED v3.1 database (van der Werf et al., 2010). The biomass burning emissions of all
20  species are averaged over the whole available time-period, and to each vegetation type – or sector – of GFED is associated a biome of OSCAR: 'def' and 'for' are forests, 'woo' is shrublands, 'sav' is grasslands, 'agr' is croplands; 'pea', i.e. peatlands, are left alone. As in section 2.3.2, pastures are assumed to be 60% grasslands and 40% bare soil. The parameters are then obtained by simply taking the ratio of the emissions of a given species over those of CO$_2$.

**2.4.2 Lagged concentrations**

25  In the next sections, we need an estimate of the stratospheric concentration change of some species. For relatively long-lived species, we assume the stratospheric concentration change of this species can be approximated by its change in atmospheric concentration (X), albeit with a time-lag ($\tau_{\mathrm{lag}}$). This change in "lagged" concentration ($\mathrm{X_{lag}}$) is formulated as:

$$
\tau_{\mathrm{lag}} \frac{\mathrm{d}}{\mathrm{d}t} \mathrm{X_{lag}} = \Delta \mathrm{X} - \Delta \mathrm{
[revised manuscript text omitted]
^{OH}_{CH4}$, $\chi^{OH}_{O3s}$, $\chi^{OH}_{T_A}$, $\chi^{OH}_{Q_A}$, $\chi^{OH}_{NOx}$, $\chi^{OH}_{CO}$, $\chi^{OH}_{VOC}$, $\tilde{\chi}^{OH}_{NOx}$, $\tilde{\chi}^{OH}_{CO}$ and $\tilde{\chi}^{OH}_{VOC}$) are taken as one of the four sets of values from the study by Holmes et al. (2013, table 2). Alternatively, for backward compatibility, these parameters can also be taken as the mutli-model mean estimates from the Ox-Comp project (Ehhalt et al., 2001, table 4.11), in which case the sensitivities to temperature, humidity and ozone are nil. The preindustrial global atmospheric temperature $T_{A,0}$ is set to 251 K, and the  proportionality coefficients are $\kappa_{T_A}$ = 0.94 and $\kappa_{Q_A}$ = 1.5 (Holmes et al., 2013). The saturation vapor pressure parameters are obtained from Jacobson (2005, equation 2.62) for which a small temperature perturbation is assumed, giving: $\kappa_{svp}$ = 17.67 and $T_{svp}$ = –29.65 K. The preindustrial stratospheric ozone burden $O3s_0$ is set to 280 DU, roughly following Cionni et al. (2011). The values of $E^{NOx}_{nat}$, $E^{CO}_{nat}$ and $E^{VOC}_{nat}$ are from Skeie et al. (2011, table 1).

**2.5.2 Wetlands emissions**

Natural wetlands are the largest natural source of methane (Ciais et al., 2013b), and the future variation of this source could be significant for future climate change (e.g. O'Connor et al., 2010). We thus decided, since version 2.1 of OSCAR, to include a simple module describing the variation of this source of $CH_4$. The current version is very close to the previous one (Gasser, 2014), except that a larger variety of parameterizations is now available.

First, we estimate the regional change in $CH_4$ emission per unit area of wetlands ($e_{wet}$) as being proportional to its preindustrial value and to the relative change in heterotrophic respiration of the litter carbon pool in the same region. To this end, wetlands are considered to be a mix of the other biomes, with partition coefficients ($\pi^b_{wet}$; $\sum_b \pi^b_{wet} = 1$) having a non-zero value only for natural biomes. We note that this is an *ad hoc* assumption that we make because we lack detailed outputs from complex wetlands models. The litter pool is chosen as a proxy of the changes in wetlands induced by more general changes in the carbon-cycle. Therefore, here we implicitly assume that the sensitivity of areal wetlands emissions to environmental conditions – e.g. $CO_2$ or temperature – is the same as that of heterotrophic respiration. So we have:

$$\Delta e^i_{wet} = e^i_{wet,0} \frac{\sum_b \pi^{i,b}_{wet} \Delta rh^{i,b}_F}{\sum_b \pi^{i,b}_{wet} rh^{i,b}_{F,0}}. \tag{44}$$

Second, we assume that the regional change in wetlands area extent ($A_{wet}$) depends on linear sensitivities to: atmospheric $CO_2$ ($\gamma_{wet,C}$), local surface temperatures ($\gamma_{wet,T}$), local yearly precipitations ($\gamma_{wet,P}$). This formulation is similar to that used for wildfire intensity in equation (12), and $CO_2$ is used as a proxy of changes in e.g. evapotranspiration or vegetation species distribution. Mathematically:

$$\Delta A^i_{wet} =$$
$$A^i_{wet,0} \left( \gamma^i_{wet,C} \Delta CO2 + \gamma^i_{wet,T} \Delta T^i_L + \gamma^i_{wet,P} \Delta P^i_L \right). \tag{45}$$

Consequently, the change in regional emission of methane by wetlands ($E_{\mathrm{wet}}$) is calculated as:

$$\Delta E_{\mathrm{wet}}^{i} = e_{\mathrm{wet},0}^{i}\,\Delta A_{\mathrm{wet}}^{i} + \Delta e_{\mathrm{wet}}^{i}\,A_{\mathrm{wet},0}^{i} + \Delta e_{\mathrm{wet}}^{i}\,\Delta A_{\mathrm{wet}}^{i}. \tag{46}$$

We calibrate two sets of parameters for wetlands. First, the preindustrial equilibrium of the wetlands can be calibrated on seven WETCHIMP models (Melton et al., 2013). We deduce the $\pi_{\mathrm{wet}}$ parameters by combining the wetlands map from the "exp 1" simulation, that is the  control experiment of the WETCHIMP exercise, and the land-cover map used in section 2.3.2 for natural vegetation. The preindustrial areal emissions $e_{\mathrm{wet},0}$ are also taken from this "exp 1" simulation, but they are scaled down by a factor equal to the ratio of our preindustrial atmospheric $CO_2$ over the one used in WETCHIMP i.e. by a factor of about 0.92, as we did with NPP in section 2.3.2. Second, the parameters for the transient response of wetlands extent (i.e. $\gamma_{\mathrm{wet},C}$, $\gamma_{\mathrm{wet},T}$, $\gamma_{\mathrm{wet},C}$) can be calibrated on six WETCHIMP models (reminder: see appendix B for a list of those models). To do so, we use "exp4", "exp5" and "exp6": factorial simulations that separate the effect of temperature, precipitations and atmospheric $CO_2$, respectively. For the same reasons as with wildfires, we also keep an option to turn off the preindustrial wetlands flux and/or its transient response.

**2.5.3 Atmospheric $CH_4$ and RF**

On the basis of the previous sections, the incremental change in atmospheric $CH_4$ follows the mass-balance equation:

$$\alpha_{\mathrm{atm}}^{\mathrm{CH4}}\,\frac{\mathrm{d}}{\mathrm{d}t}\Delta \mathrm{CH4} = E_{\mathrm{CH4}} + \Delta E_{\mathrm{bb}}^{\mathrm{CH4}} + \sum_{i}\Delta E_{\mathrm{wet}}^{i} + \Delta F_{\downarrow}^{\mathrm{CH4}}. \tag{47}$$

This equation implicitly assumes that all the natural sources of methane but natural wetlands remain unchanged since the preindustrial. Here, we also note that the anthropogenic emissions $E_{\mathrm{CH4}}$ do include emissions from rice paddies – i.e. from anthropogenic wetlands.

The radiative forcing induced by the increase in atmospheric $CH_4$ follows a square-root formula to which an *ad hoc* function ($\mathcal{F}_{\mathrm{over}}$) is added to account for the overlap between the absorption bands of methane and nitrous oxide (N2O), following Myhre et al. (1998). It gives:

$$\Delta \mathrm{RF}^{\mathrm{CH4}} =$$

$$+ \alpha_{\mathrm{rf}}^{\mathrm{CH4}}\,\sqrt{\mathrm{CH4}_0}\left(\sqrt{1 + \frac{\Delta \mathrm{CH4}}{\mathrm{CH4}_0}} - 1\right)$$

$$- \left(\mathcal{F}_{\mathrm{over}}\left[\Delta \mathrm{CH4}, \Delta \mathrm{N2O}\right] - \mathcal{F}_{\mathrm{over}}\left[\Delta \mathrm{CH4} = 0, \Delta \mathrm{N2O}\right]\right); \tag{48}$$

where $\alpha_{\mathrm{rf}}^{\mathrm{CH4}}$ = 0.036 W m$^{-2}$ ppb$^{-0.5}$ and the analytical expression of $\mathcal{F}_{\mathrm{over}}$ are given by Myhre et al. (2013a, table 8.SM.1). In addition to the RF induced by methane itself, we have to account for the RF induced by the increase in stratospheric water vapor caused by the oxidation of methane. To do so, as others (e.g. Meinshausen et al., 2011), we assume it is equal to 15% of the direct methane RF, but calculated with its lagged concentration:

$$\Delta \mathrm{RF}^{\mathrm{H2Os}} = \alpha_{\mathrm{rf}}^{\mathrm{H2Os}}\,\sqrt{\mathrm{CH4}_0}\left(\sqrt{1 + \frac{\Delta \mathrm{CH4}_{\mathrm{lag}}}{\mathrm{CH4}_0}} - 1\right); \tag{49}$$

where $\alpha_{\mathrm{rf}}^{\mathrm{H2Os}} = 0.15 \times \alpha_{\mathrm{rf}}^{\mathrm{CH4}} = 0.0054$ W m$^{-2}$ ppb$^{-0.5}$. For the preindustrial atmospheric concentration, we take CH4$_0 = 722$ ppb (IPCC, 2013, table AII.1.1a).

**2.6 Nitrous oxide**

In OSCAR, the stratospheric sink of nitrous oxide is included, and a particular attention is paid to how it varies with anthropogenic and natural external factors. However, no other natural processes are endogenous to the model, meaning that no change in natural sources or sinks of nitrous oxide (e.g. ocean, natural soils, biological fixation) is assumed.

**2.6.1 Stratospheric sink**

The oxidation of nitrous oxide follows the same modelling approach as that of methane, with only one sink in the stratosphere that has a varying lifetime. The law used to make the stratospheric lifetime vary, however, is recent and different from the previous version of the model.

The flux of oxidized N$_2$O ($F_{\downarrow}^{\mathrm{N2O}}$) is driven by the preindustrial lifetime of nitrous oxide with regard to stratospheric oxidation ($\tau_{\mathrm{h}\nu}^{\mathrm{N2O}}$). The transient change in this stratospheric lifetime is a function ($\mathcal{F}_{\mathrm{h}\nu}$) of: the lagged N$_2$O concentration (N2O$_{\mathrm{lag}}$); the equivalent effective stratospheric chlorine (EESC; see section 2.8.2); and global surface temperature change ($T_G$). The dependency on N$_2$O and the EESC is meant to model the impact of a change in stratospheric ozone that changes the actinic flux, which in turn changes the stratospheric sink (e.g. Prather, 1998). We have:

$$
\begin{aligned}
&\alpha_{\mathrm{atm}}^{\mathrm{N2O}\,-1} \Delta F_{\downarrow}^{\mathrm{N2O}} = \\
&-\frac{\mathrm{N2O}_0}{\tau_{\mathrm{h}\nu}^{\mathrm{N2O}}} \left( \left( 1 + \frac{\Delta \mathrm{N2O}_{\mathrm{lag}}}{\mathrm{N2O}_0} \right) \mathcal{F}_{\mathrm{h}\nu} [\Delta \mathrm{N2O}_{\mathrm{lag}}, \Delta \mathrm{EESC}, \Delta T_G] - 1 \right).
\end{aligned}
\tag{50}
$$

The formulation of $\mathcal{F}_{\mathrm{h}\nu}$ is inspired by that used for methane and the study by Prather et al. (2015). It has three chemical sensitivities ($\chi_{\mathrm{N2O}}^{\mathrm{h}\nu}$, $\chi_{\mathrm{EESC}}^{\mathrm{h}\nu}$ and $\chi_{\mathrm{age}}^{\mathrm{h}\nu}$). This last parameter represent the sensitivity of the sink to a change in stratospheric age of air. This age-of-air change is itself driven by a changing Brewer-Dobson circulation which is induced by a changing climate (e.g. Butchart, 2014). In the following, we consider that the inverse of the relative change in age of air is a linear function of the absolute change in global surface temperature (parameterized by $\gamma_{\mathrm{age}}$; see also figure S23). This leads to the following formula:

$$
\begin{aligned}
\ln[\mathcal{F}_{\mathrm{h}\nu}] = \\
+ \chi_{\mathrm{N2O}}^{\mathrm{h}\nu} \ln \left[ 1 + \frac{\Delta \mathrm{N2O}_{\mathrm{lag}}}{\mathrm{N2O}_0} \right] \\
+ \chi_{\mathrm{EESC}}^{\mathrm{h}\nu} \ln \left[ 1 + \frac{\Delta \mathrm{EESC}}{\mathrm{EESC}_0} \right] \\
+ \chi_{\mathrm{age}}^{\mathrm{h}\nu} \ln \left[ \frac{1}{1 + \gamma_{\mathrm{age}} \Delta T_G} \right].
\end{aligned}
\tag{51}
$$

The preindustrial stratospheric lifetime $\tau_{\mathrm{h}\nu}^{\mathrm{N2O}}$ is taken as 123 years (Prather et al., 2015). As we do with methane, we introduce variation in the N$_2$O lifetime by having the option to rescale the default value by a factor equal to the lifetime

simulated by any of the eight models of Prather et al. (2015, table 2) over the multi-model mean estimate. The first two chemical sensitivities of the stratospheric sink (i.e. $\chi^{\mathrm{h}\nu}_{\mathrm{N2O}}$ and $\chi^{\mathrm{h}\nu}_{\mathrm{EESC}}$) are taken as one of the four sets of values from the study by Prather et al. (2015). Three sets of value are given in their table 3, and the fourth is the  recommendation in their text. Also, to translate their table 3 into our parameters, we assume that the preindustrial EESC in the models were 420

5   ppt – from IPCC (2013, table AII.1.1b) and Newman et al. (2007, table 1). Alternatively, for backward compatibility, these parameters can also follow (Prather et al., 2012, table A1), in which case the sensitivity to EESC is zero.

Regarding the chemical sensitivity to the age of air, we assume it is not zero only when the other sensitivities are deduced from the "G2d" model, therefore following the results by Prather et al. (2015, table 3) and their discussion pointing out the experimental aspect of such a parameterization. Nevertheless, in this specific case we need further information about the "G2d"

10  model which we take from Fleming et al. (2011, figure 12) where one can see that the age of air at an altitude of 25 km changed from about 4.5 to 4.0 between the preindustrial and present-day periods. This is enough to deduce the $\chi^{\mathrm{h}\nu}_{\mathrm{age}}$ parameter. And then, the $\gamma_{\mathrm{age}}$ parameter can be calibrated on seven CCMVal2 chemistry-transport models (Morgenstern et al., 2010). To do so, we use outputs from the "REF-B2" experiment which is a fully transient simulation over 1961–2099: we use the "mean_age" output at a pressure-level of 25 hPa ($\sim$25 km) and the temperature at the surface level. We then fit the parameter following

15  our inversed linear relationship, defining the preindustrial conditions as the averaged first ten years of the simulations. The CCMVal2 fits are shown in figure S23.

**2.6.2   Atmospheric N$_2$O and RF**

The incremental change in atmospheric N$_2$O follows:

$$\alpha^{\mathrm{N2O}}_{\mathrm{atm}} \tfrac{\mathrm{d}}{\mathrm{d}t} \Delta \mathrm{N2O} = E_{\mathrm{N2O}} + \Delta E^{\mathrm{N2O}}_{\mathrm{bb}} + \Delta F^{\mathrm{N2O}}_{\downarrow}; \tag{52}$$

20  noting again that this implicitly assumes natural emissions remain unchanged since the preindustrial.

Similarly to methane, the radiative forcing induced by the increase in atmospheric N$_2$O follows a square-root formula to which the *ad hoc* overlap function is added:

$$\Delta \mathrm{RF}^{\mathrm{N2O}} =$$

$$+ \alpha^{\mathrm{N2O}}_{\mathrm{rf}} \sqrt{\mathrm{N2O}_0} \left( \sqrt{1 + \frac{\Delta \mathrm{N2O}}{\mathrm{N2O}_0}} - 1 \right)$$

25  $$- \left( \mathcal{F}_{\mathrm{over}} \left[ \Delta \mathrm{CH4}, \Delta \mathrm{N2O} \right] - \mathcal{F}_{\mathrm{over}} \left[ \Delta \mathrm{CH4}, \Delta \mathrm{N2O} = 0 \right] \right); \tag{53}$$

where $\alpha^{\mathrm{N2O}}_{\mathrm{rf}}$ = 0.12 W m$^{-2}$ ppb$^{-0.5}$ and $\mathcal{F}_{\mathrm{over}}$ are given by Myhre et al. (2013b, table 8.SM.1). For the preindustrial atmospheric concentration, we take N2O$_0$ = 270 ppb (IPCC, 2013, table AII.1.1a).

**2.7   Halogenated compounds**

OSCAR accounts for many halogenated species. These are grouped into three categories: eleven hydrofluorocarbons (HFC-23,

30  HFC-32, HFC-125, HFC-134a, HFC-143a, HFC-152a, HFC-227ea, HFC-236fa, HFC-245fa, HFC-365mfc, HFC-43-10mee)

noted together {HFC}; eight perfluorocarbons ($CF_4$, $C_2F_6$, $C_3F_8$, c-$C_4F_8$, $C_4F_{10}$, $C_5F_{12}$, $C_6F_{14}$, $C_7F_{16}$) to which we add $SF_6$ and $NF_3$, and noted together {PFC}; and sixteen ozone depleting substances (CFC-11, CFC-12, CFC-113, CFC-114, CFC-115, $CCl_4$, $CH_3CCl_3$, HCFC-22, HCFC-141b, HCFC-142b, Halon-1211, Halon-1202, Halon-1301, Halon-2402, $CH_3Br$, $CH_3Cl$) noted together {ODS}. These are the same as in previous version 2.1.

**2.7.1 Atmospheric sinks**

Conceptually, the modelling approach of the halogenated compounds' sinks is similar to that used for methane. Each of these species X is affected by three sinks, each sink with its specific preindustrial lifetime: a tropospheric oxidation by the hydroxyl radical ($\tau_{OH}^X$), a stratospheric oxidation ($\tau_{h\nu}^X$), and another sink which encloses all other processes such as oxidation in dry soils or in the oceanic boundary layer ($\tau_{othr}^X$). Note that a given oxidation process may not actually affect a given species; in this case the associated lifetime is set to a value of infinity ($\infty$) – or equivalently its loss frequency ($\nu^X = 1/\tau^X$) is set to zero. Mathematically, similarly to equation (40), we have for any species X being a HFC, PFC or ODS:

$$\alpha_{atm}^X{}^{-1} \Delta F_\downarrow^X =$$
$$-\frac{1}{\tau_{OH}^X} \left( (\Delta X + X_0) \, \mathcal{F}_{OH} \left[ \Delta CH4, \Delta O3s, \Delta T_G, \mathcal{F}_{prec} \right] - X_0 \right)$$
$$-\frac{1}{\tau_{h\nu}^X} \left( (\Delta X_{lag} + X_0) \, \mathcal{F}_{h\nu} \left[ \Delta N2O_{lag}, \Delta EESC, \Delta T_G \right] - X_0 \right)$$
$$-\frac{1}{\tau_{othr}^X} \Delta X; \tag{54}$$

where the functions $\mathcal{F}_{OH}$, $\mathcal{F}_{prec}$ and $\mathcal{F}_{h\nu}$ are the same as in sections 2.5.1 and 2.6.1.

[revised manuscript text omitted]

5 Then, a change in stratospheric ozone burden (O3s) is assumed to happen with a change in EESC, with a linear sensitivity ($\chi_{\text{EESC}}^{\text{O3s}}$). To the effect of ODSs, we add the effect of nitrous oxide following the simple formulation by Daniel et al. (2010) which needs two additional parameters: one to quantify the linear sensitivity of stratospheric ozone to nitrous oxide ($\chi_{\text{N2O}}^{\text{O3s}}$), and one to account for the non-linear interaction between chlorine and nitrogen chemistries ($\text{EESC}_{\times}$). As per tropospheric ozone, a linear sensitivity to global surface temperature change ($\Gamma_{\text{O3s}}$) is also added, which sums up to:

10 $\Delta \text{O3s} =$

$+ \chi_{\text{EESC}}^{\text{O3s}} \Delta \text{EESC}$

$+ \chi_{\text{N2O}}^{\text{O3s}} \left( 1 - \dfrac{\Delta \text{EESC}}{\text{EESC}_{\times}} \right) \Delta \text{N2O}_{\text{lag}}$

$+ \Gamma_{\text{O3s}} \Delta T_G. \tag{61}$

Regarding the EESC parameterization, Newman et al. (2006, tables A1 & A2) provide values of fractional release $\pi_{\text{rel}}$ for all
15 our ODSs, assuming a mean age-of-air of 3 years taken equal to the time-lag of section 2.4.2. To introduce other possibilities of parameterization in the model, we can alternatively take fractional release values from Laube et al. (2013), either the values for the mid-latitudes or those for the high latitudes. In this case, if a value is missing for a given ODS we take that from Newman et al. (2006). The chemical formula of each ODS gives $n_{\text{Cl}}$ and $n_{\text{Br}}$. And we take $\alpha_{\text{Cl}}^{\text{Br}} = 60$ (Newman et al., 2007).

The chemical sensitivity of stratospheric ozone to EESC and that to global climate change (i.e. $\chi_{\text{EESC}}^{\text{O3s}}$ and $\Gamma_{\text{O3s}}$) can be
20 calibrated on eleven CCMVal2 chemistry-transport models studied by Douglass et al. (2014), using the results from their multi-linear regression. The sensitivity to nitrous oxide is calculated using the formula by Daniel et al. (2010): $\chi_{\text{N2O}}^{\text{O3s}} = \chi_{\text{EESC}}^{\text{O3s}} \alpha_{\text{N2O}}^{\text{EESC}} \pi_{\text{rel}}^{\text{CFC11}}$; where $\alpha_{\text{N2O}}^{\text{EESC}}$ is a parameter measuring the relative strength importance of N$_2$O and chlorine. Values for the parameters are given by Daniel et al. (2010) and based on Ravishankara et al. (2009): $\alpha_{\text{N2O}}^{\text{EESC}} \simeq 10.4$ ppt ppb$^{-1}$ and $\text{EESC}_{\times} \simeq 2642$ ppt. Also, we  add two extra options to this module: one for which this response
25 of stratospheric ozone to nitrous oxide is simply turned off; and one for which it is assumed linear – instead of saturating – by setting the $\text{EESC}_{\times}$ parameter to infinity.

Finally, the radiative forcing induced by the change in stratospheric ozone burden is assumed to be linear:

$$\Delta \text{RF}^{\text{O3s}} = \alpha_{\text{rf}}^{\text{O3s}} \Delta \text{O3s}; \tag{62}$$

[revised manuscript text omitted]

To estimate global warming, however, we have to account for the so-called "efficacy" of these forcings, i.e. we have to introduce new parameters ($\kappa_{warm}^X$) that measure the relative efficiency at warming the Earth of a given RF when compared to the RF of $CO_2$ (see e.g. Hansen et al., 2005; Forster et al., 2007). In OSCAR, we assume all efficacies are equal to 1 – although accounting for the semi-direct effect of BC could be defined as using an efficacy – except for the two surface albedo forcings and for volcanic aerosols. Therefore, the RF used to calculate warming ($RF_{warm}$) is:

$$\Delta RF_{warm} =$$
$$+ \Delta RF^{WMGHG} + \Delta RF^{NTCF} + RF_{con} + RF_{solar}$$
$$+ \kappa_{warm}^{BCsnow} \Delta RF^{BCsnow} + \kappa_{warm}^{LCC} \Delta RF^{LCC} + \kappa_{warm}^{volc} RF_{volc}. \tag{76}$$

[revised manuscript text omitted]

---

## Author Response (AR1)

**Comment 1.0.** My apologies for taking so long to get my comments in. This is a very well written paper that carefully and meticulously documents the new parameterized ESM called OSCAR. I have some minor issues outlined below, but overall, I would be happy to use this model in any of my applications where it is needed. OSCAR rests clearly on the analyses of the full chemistry-carbon-climate models and these sources are carefully documented here.

**Response 1.0.** We thank the referee for his/her review and support.

**C1.1.** Intro is OK, but (p.2) is OSCAR just another box model? Please say so, describing it as? Parameterized, multicomponent ESM without a model grid?

**R1.1.** Following this comment, some below, and comments from the other referee, we've decided to expand the introduction by with one paragraph, to take more time to present the model and the general idea of how its parameters are derived (what we call meta-modeling).

*"Here, we present an important update of a simple Earth system model that has already been used for some time. The model is named OSCAR, and this paper provides a comprehensive description of version 2.2. OSCAR can be described as a non-linear box model whose number of boxes, however, is fairly large. It is not spatially resolved (i.e. it is not gridded) but key processes such as land-use change or aerosol physico-chemistry are regionalized to account for the disparity in such processes that is observed in the real world. OSCAR does not endogenously simulate intra- or inter-annual variability. Consequently, although the time-step of its inputs and outputs is one year, the main purpose of the model is to simulate trends in the Earth system change, and not year-to-year variations. OSCAR is also a parametric model whose relatively large number of parameters are almost all calibrated on complex models. We call this approach meta-modelling: each module of OSCAR is designed to emulate the behavior of other more specialized models (e.g. global climate models, dynamical vegetation models, or chemistry-transport models). For most modules, we have access to several sets of parameters (one per complex model used to calibrate) and, rather than taking the average or arbitrarily choosing one, we adopt a probabilistic approach in which a given simulation with OSCAR is repeated many times with different sets of parameters picked at random."*

**C1.2.** I like the idea of the regional boxes being used to account for the heterogeneity of the global forcing/response. Whatever, but get this upfront.

**R1.2.** The general approach for regionalizing is now also explained in this added paragraph in introduction.

**C1.3.** I admit that I did not proof all of the equations here, but rather looked at structural problems and inclusiveness. A key question (answered for me at the end) was is there a source of interannual variability and climate chaos? Of ENSO? in OSCAR. Apparently not – but if it could be implemented, how would it change the analyses and ensembles?

**R1.3.** The fact that there is no inter-annual (nor intra-annual) variability is now mentioned upfront in this additional paragraph about what OSCAR is.

Now, the issue of whether the inter-annual variability can be implemented in the model and how it would affect its results is not a small one! First, since OSCAR is not process-based it seems impossible that the variability be endogenous – except by adding specific (stochastic) equations to

generate it. Second, since OSCAR has been thought and developed to focus on the trend of (anthropogenic) climate change, all equations would need to be modified to include the variability as an explicit driver. This would involve, for instance, to implement a stochastic model of the Southern Oscillation Index (or at least forced with observed values), with the value of the SOI then influencing e.g. the NPP or emissions from biomass burning, following laws calibrated on e.g. CMIP5 models.

One can easily see that implementing all this is no small work, especially if natural modes of variability other than ENSO are to be considered (e.g. the Atlantic Multidecadal Oscillation or the QBO for the stratospheric chemistry). We agree with the referee, however, that this would be an extremely interesting direction in which the model could be developed. That said, there is little if any evidence that this would result in any important change in the trends simulated by the model, given how relatively 'smooth' the differential system of this version is. We believe that the work on natural variability would be more interesting if 'strong' non-linearities of the system were also added to the model (e.g. permafrost, ice-sheets, …).

**C1.4.** OSCAR = "Earth system change" model." Excellent.

**R1.4.** Thank you. We do believe this is an important distinction to make.

**C1.5.** p.3 "*but all other sectors provided by the inventories are accounted for*" – does this include aviation and shipping. OK, 2.2.3 explains part of this.

**R1.5.** Emissions from aviation and shipping are included in what we call the "anthropogenic emissions" drivers. What is detailed in 2.2.3 is that the effect of contrails is included in the model only as a direct RF.

We extended the sentence quoted by the referee to add shipping and aviation as an example of what is included in these drivers. This gives: "*Note that the emissions from biomass burning of natural vegetation are removed from these datasets, since those emissions are endogenous to the OSCAR model (see section 2.4.1), but all other sectors provided by the inventories are included, notably (but not only) agricultural waste burning, and shipping and aviation.*"

**C1.6.** p.7 "*nutrient deposition – albeit the latter not in this version of OSCAR.*" OK, I would have liked to see this included and am glad it is clearly stated. Perhaps a table of links and processes with a yes/no/partial would be useful to scan?

**R1.6.** We understand the referee's motivation in suggesting this table of included/excluded processes. And we gave it a try but weren't satisfied with the result. We find really difficult to decide where to stop in the details provided in this table. Let's take the ocean carbon-cycle as an example. If we want to speak generally, so as to keep the table short and easy to read, one can say that our model includes the physical pump but not the biological pump. But even for the physical pump, one can argue that it is only partially included, as e.g. no change in sea-ice cover or over-turning circulation is implemented in the model. And, because our model is formulated in perturbation, one can also argue that there is a biological pump that (implicitly) remains constant whatever the environmental/anthropogenic perturbations. So to be thorough we should provide a table of the links between the state variables of the model, e.g. the DIC of the surface ocean is a function of the carbon in the surface ocean, of the mixed layer depth, and of the sea surface temperature. But this

would make the table rather long and complicated. And summarizing these causal links is the reason why we made figure 1 initially.

However, we can agree with the referee that the mentions of what is not included in the model were difficult to find in the main text. So we've added additional introductory paragraphs to the subsections dedicated to CO2, CH4, N2O, O3 and the aerosols, in which we very briefly list the main processes that are implemented in OSCAR and those that are lacking. We believe that these introductory paragraphs, combined to figure 1, are enough for the reader to understand what is or not implemented in OSCAR.

**C1.7.** p.12 "the migration of natural biomes caused by changes in environmental conditions (e.g. Jones et al., 2009). This is however not included in this version of OSCAR." Another item for the table of processes.

**R1.7.** Yes. See above.

**C1.8.** p.13 "*disturbe*" >disturb – surprising, minor. Overall this paper is very well and clearly written.

**R1.8.** Corrected.

**C1.9.** p.18 "*it is parameterized by three relative chemical sensitivities (-) and the preindustrial natural emissions of the three ozone precursors (ENOxnat , ECOnat , EVOCnat )*" OK, but worrisome, would clearly like to have a latitudinally dependent impact of the NOx and VOC emissions at least (probably not CO). for example, "*All the chemical sensitivities of the OH sink (i.e. _OH CH4, _OH O3s, _OH TA , _OH QA , _OH NOx, _OH CO, _OH VOC, _OH NOx, _OH CO and _OH VOC) are taken as one of the four sets of values from the study by Holmes et al. (2013, table 2)*" This is really fine, but again ignores the NOx vs Latitude reactivity (Wild et al, 2001, Indirect long-term global cooling from NOx emissions, Geophys. Res. Lett.)

**R1.9.** Yes, the referee is entirely right: the OH chemistry (affecting the methane lifetime) should be regionalized. However, we did not find relatively recent studies that were readily exploitable to do this regionalization (e.g. the HTAP simulations used for O3 did not report OH-relative variables). Also, we feel this regionalization is somewhat less crucial than that of e.g. O3 because in the model it affects only CH4, and CH4 is treated as a well-mixed gas. However, we agree this should be tested and requires additional model development. We decided to leave this feature and discussion for a future version of the model for which a better (and more consistent) atmospheric chemistry would need to be developed, based on more detailed global atmospheric models.

Nevertheless, it appears our paper failed to acknowledge this overlooked effect. We have therefore added a sentence right after these equations: "*None of these two formulations shows regionalized chemical sensitivities of the OH sink, however, whereas in reality the sink is sensitive to where the ozone precursors are emitted -- especially the NOx (e.g. Wild et al., 2001).*" We've also added a small specific reference to that aspect in the discussion/conclusion section.

**C1.10.** p.19 "*This implicitly assumes that all the natural sources of methane but natural wetlands remain unchanged since the preindustrial*" – having trouble with the 'but', do you mean 'and'??

**R1.10.** The emissions of methane by natural wetlands are estimated by the model, and therefore are not implicitly held constant. This appears clearly in equation (47) which is right above the sentence quoted by the referee. And this explains the use the word "but". We've changed the beginning of the sentence to: "*This equation implicitly assumes […]*", pointing the reader to the equation above to make the sentence clearer.

**C1.11.** p.20 Nice job with the complex system that is N2O.

**R1.11.** Thank you.

**C1.12.** p.22 "*in this case the associated lifetime is set to a value of infinity*" This is a typical problem with using lifetimes instead of loss frequencies. I would have made all of the tau's into 1/tau = LF which can then be set to zero (inifinity is hard with a computer: : :.)

**R1.12.** It is true that loss frequencies are a scientifically sounder way of writing these equations. But (unfortunately) using lifetimes is much more typical within the community; therefore we chose to stick to these. We've just slightly extended that sentence, adding: "*– or equivalently its loss frequency ($v^X = 1/\tau^X$) is set to zero.*"

Note however that the 'numpy' library of 'Python' language handles values of 'infinity'. And OSCAR is actually coded with such infinity values.

**C1.13.** p.23 "*at chemical equilibrium with*" – I think equilibrium is really abused by many of our colleagues, it is rather a steady-state that is reached. Equilibrium has deeper implications of detailed balance as in thermodynamics.

**R1.13.** Agreed! This is of course a steady-state. We've checked our usage of "equilibrium" throughout the whole paper, and changed to steady-state when relevant. We kept the wording "preindustrial equilibrium" however. Although it is debatable whether it was really an equilibrium, a steady-state, or none of the two, it is a trivial equilibrium of our model.

**C1.14.** "*with linear global sensitivities (: : :) that are regionalized thanks to region-specific weights: : :*" This is very nicely done since that are large regional differences in both the chemical and RF response to these short-lived emissions!

**R1.14.** Thank you. We are not entirely satisfied by the degree of detail of this regionalization: four regions is not much. But, just as we couldn't regionalize at all the OH sink, we were here limited by the availability of (exploitable) data.

**C1.15.** p.25 The param for O3s is reasonable, but I am not sure that I would agree with the N2O impact being reduced at high EESC (eqn 61). The N2O loss occurs in a very different region from either Cly losses (lower strat or 40+km for ClO+O) and should simply be just linear?

**R1.15.** Thanks to the referee, we have added a new option for in the model to have this change linear. So we end up with three options: no N2O-O3 effect at all, the saturating effect following

Daniel et al. (2010), and the new option of linear variation (with EESCx set to 'infinity'). Main text has been changed accordingly.

**C1.16.** p.26 I am worried about the formulation of eqns like 63 because if loss frequencies go to zero then Tau goes to infinity (and stated earlier) and this formula blows up. This should be written in the form like 1 / ( 1/tau + min.loss.freq )

**R1.16.** We understand the referee's concerns. However, we do not share them for two reasons. First, in this version of the model these lifetimes are constant. Second, if the loss frequencies go to zero, the lifetimes increase and the species are no longer short-lived. So our assumed steady-state would become wrong much sooner than the equation would 'blow up', and we would have to use a (time) differential equation instead.

**C1.17.** p.35 "*The differential system is solved with the forward Euler method (Euler, 1768) with a time-step (_t) that can be chosen before any simulation with OSCAR – although time-steps greater than a quarter of year systematically make the model diverge. This time-step is usually set to _t = 1/6 yr*". Since these equations do not appear to be very stiff (you can get away with 1/6 yr) you might want to use a Bulirsch-Stoer high accuracy integrator (not much cost) with a 1-yr overall time step (it divides the large step into a nested sequence of explicit steps and then extrapolates to give you an almost perfect answer. Note that B-S does not help with very stiff equations like integration O(1D) at sunrise.

**R1.17.** We thank the referee for this suggestion that we will keep in mind if in the future the model appears to need too many time-steps with the explicit method. But we are reluctant to change anything as to the numerical solving method we use: it is unclear what the benefits would be in the short term. On the precision side, it is very likely that the difference of result between two solving methods is much smaller than the difference between two random parameterizations; and so the uncertainty from the method is likely completely covered by that from the parameters. On the computation time side, although the Burlisch-Stoer method may be faster, we believe that in the current state of the model there are many ways to improve the speed of computation even more, starting with coding the core function of OSCAR in e.g. C or Fortran instead of Python. This work of optimizing the model is not a priority for us, however.

**C1.17.** "*more than 1043 potential combinations of parameters*" – how about 2**128 = 10**43 Are there really 128 independent parameters in Table 2?

**R1.17.** Yes, the 10^43 combinations are obtained by a simple product of the number of 'individual' parameters or parameter sets presented in table 3. And yes, there are that many independent choices possible for any simulation with OSCAR. Now, the question of whether these parameters are actually independent in the real world is a completely different one! We know for sure that it is not the case for some. In the discussion we give the example of the parameters related to the ocean dynamic which are different for the C-cycle and for the climate response (whereas they should be the same). However, there is no solution to that problem as it stems from the way OSCAR is built as an aggregate of small emulators!

**C1.18.** p.36 and elsewhere. "*The interannual variability of the land sink simulated by OSCAR in the offline case does not match that from*" – I am a bit confused as to how you implement interannual

variability and climate modes in OSCAR. DO not volcanoes affect the C-cycle thru T? if not diffuse radiation.

**R1.18.** There is no variability in OSCAR (as now explained upfront in introduction). The variability in the offline simulations come only from the variability in the climatology (CRU data) used to force the model. This is explained in the introductory subsection of the results section, but we've now also added a sentence in the 'experimental setup' section to remind the reader of this: "*Note also that, because the climate data is based on observation, the offline simulation will show natural variability, albeit not as a feature of OSCAR but as one of the driving data.*".

**C1.19.** p.37ff – Nice discussion and example with CH4. p.38 – When comparing the 'online' to 'offline' CH4 simulations it might be useful to remind people that the difference between the two, because nominally these two designations describe computational differences rather than models not having feedbacks. Same for N2O in the following.

**R1.19.** Yes. We've added a reminder of how the atmospheric growth rate in the offline setup is reconstructed: "*[the offline atmospheric growth rate] reconstructed as the balance between the concentration-driven sinks and the anthropogenic emissions normally driving OSCAR in online mode*".

**C1.20.** p.41 "*the lack of interannual variability in OSCAR: : :*" OK, now it is clear, but a discussion of this could be upfront – maybe I missed it.

**R1.20.** Done. See **R1.3** and **R1.18**.

**C1.21.** p.43 "*These many degrees of freedom increase the odds of seeing a given simulation diverge, or at least depart unreasonably from the plausible range of results.*" I am not sure this is true. Unless the equations are chaotic (e.g., Lorenz N-cycle model, Liapunov, : : :) then divergence would not seem possible in a linearized model.

**R1.21.** Yes, this is an abuse of the word 'diverge'. Removed.

**C1.22.** p.43 "*Also, coupling of the tropospheric and stratospheric chemistries would be an improvement, especially for ozone, as would a finer regionalization be. We note however that a tremendous amount of factorial simulations by complex chemistry transport models would be needed to make such an improvement.*" Since the goal is to accomplish the cross-coupling and feedbacks, the N2O-CH4 link (Prather & Hsu, 2010 Science) would seem to be important (+10 molec of N2O => -3.6 molec of CH4). Is this feedback actually included in the strat-trop chemistry of OSCAR – if so great, and note it.

**R1.22.** This coupling is indeed simulated by OSCAR, albeit not as a direct coupling. The change in N2O does impact stratospheric O3 which in turn impacts the OH sink and then CH4. The coupling therefore appears as a succession of several arrows in figure 1.

**C1.23.** p. 43-44. Future or current improvements – I would vote for 2 as primary to simulating climate and to understanding the coupling across cycles.
1) Find a way to do ensembles with climate variability on interannual to decadal scales.

**R1.23.** As discussed in **R1.3**, we do believe it is a very interesting way to develop the model further. But it is likely a huge undertaking.

**C1.24.** 2) Do an eigenvalue analysis of your linearized matrix to identify time scales and the structure of the major coupled modes.

**R1.24.** This is a good suggestion. We are currently working on some theoretical analysis of the system/model, and an eigenvalue analysis is definitely have its place in such a study.

**C1.25.** 3) Add a table 4 that lists the processes and known couplings that are NOT included; this of course cannot be complete but at least some of the major areas that you chose not to include. Very useful as a reminder.

**R1.25.** Answered. See **R1.6**.

**C1.26.** 4) Describe how to go about updating the parameters when we have new results from the 'big' models.

**R1.26.** The creation of OSCAR took a lot of time in this respect, as we had to process a large amount of data from the complex models: more than 1 TB. And it was too often that the 'official' data format was not followed by a given modeling group, which makes the processing even longer. So we doubt there is an easy way to update OSCAR with the future new results from the complex models. Of course, we will follow the CMIP6 exercise, especially as more diverse data is supposed to be made available. For instance, if PFT-specific data is indeed provided for the land C-cycle, we will be able to get rid of our fractional weighting and we will have a better estimate of land carbon densities and therefore of land-use change emissions.

**C1.27.** p.63 Great figure, thanks. Minor fix: 'edges' in the figure caption should read 'lines'

**R1.27.** Thanks. The word 'edges' was from the graph theory vocabulary, but that's probably not necessary.

**Comment 2.0.** This paper is an extremely well written and thorough description of the OSCAR 2.2 model. The model carries a significant number of innovative approaches that will prove very useful in investigating the range of possible forcings, feedbacks and interactions within the full Earth system that will ultimately determine the global and regional response of this system to anthropogenic emissions of greenhouse gases, aerosol precursors and human-induced land use change.

From this perspective I recommend that the paper is published with only minor revisions for the sake of clarity and brevity. These are indicated below under requested revisions. In addition to this I have a number of general suggestions and comments/questions, which might further improve an already very good paper. The authors might like to consider some of these.

**Response 2.0.** We thank the referee for his/her review.

**C2.1.** General points. 1. The paper is very long. I realize this is necessary to provide the level of detail required for a reader to properly understand the formulation of the model. The sections describing the model components are generally very good and very clear. That said, the section where model simulation results are presented for the historical period is actually quite thin and the weakest part of the paper. In particular, a number of areas where OSCAR deviates significantly from observations, more complex models or IPCC best estimates are not always fully explained or discussed. Also the model is designed, primarily, to allow a probabilistic investigation of future Earth system change. No examples of the application of the model to possible future conditions are included in the paper.

Hence my main suggestion is that the authors consider a high-level restructuring of the paper and instead submit 2 (linked, Part I, Part II) articles, with Part I essentially being the model description part of the submitted article and Part II being (i) an extended version of the present section on the historical period simulation (with some more discussion and explanation of deviations from observations/other models/IPCC estimates and (ii) include an initial example of how the model can/will be applied in the context of investigating future Earth system change. I realize that point (ii) is no doubt intended by the authors in subsequent papers, nevertheless, some brief examples of how the model is to be applied in a future projection sense would be illustrative in the context of the model description paper (my suggested Part I).

Furthermore, in the Part I article I would recommend an initial section that gives a very brief overview of the model structure (aimed at non modeling scientists that may still be interested in the model application, e.g. results presented in Part II) along with a note to the effect that readers interested in the model formulation should read all of Part I, while those mainly interested in the model application and results can just read the short description and then jump to Part II.

This suggestion, which is just that (a suggestion), would in my opinion make the combined papers significantly more interesting to a wider audience than the present paper. If the authors prefer to keep the present single article then I suggest they more carefully discuss/explain some of the key deficiencies in the historical simulation as this section was a little thin in places.

**R2.1.** We perfectly understand the reasons motivating the referee's suggestion. And the referee is perfectly right to assume that we plan a paper equivalent to his/her suggested 'Part II' for the future.

However, the main reason why we did not submit two companion papers is the time needed to write that second paper. During this time the first paper would not be available to the scientific community. OSCAR is a model that has been used already in various studies, and that is currently being used for others. So we felt (as many reviewers of said studies did) that a clear and precise description of the model was long overdue. And for the sake of time, we decided to submit a first paper, with a second one clearly in mind (simulations are ongoing as these lines are being written).

Note also that the specific policy of GMD consisting in linking papers about the same model will partly compensate for the absence of 'Part I' and 'Part II' mentions on the papers.

As for the issue of what simulations should appear in this description paper, it was a difficult choice to make. The historical simulations with comparison to observations seemed to be a pre-requisite. But there are several reasons why we opted for no projections are all.

- First, we could not find one scenario to follow for the 'example of use' of the model: why make simulations for the RCP8.5 and not the RCP2.6? But if you choose 2 of the 4 RCPs, why not the 4? Coupled to the fact that the RCPs can be run in an emission-driven or concentration-driven fashion, this leads to already a lot of additional simulations.
- Second, this description paper is already quite long. To discuss even one simulated scenario would make it even longer – especially as it would likely require to increase the number of figures shown – and this is something we wanted to avoid.
- Third, there is an important conceptual difference between historical simulations and projections. In the former case, we can compare our results with observations. In the latter case, we only have other models results to compare with. This complicates (and extends!) the discussion: in the former case, a departure from the observation is a bad performance of the model; in the latter case, it may be explained by a difference in the models' structure.

For all these reasons, we decided to show only the historical simulations, as it is ultimately the only way to (in)validate a model such as OSCAR. And we decided to keep the (lengthy) discussion about how to use observations to constrain the parameterizations of OSCAR, and how the (constrained or unconstrained) model performs for projections compared to e.g. CMIP5 for 'Part II'. We only hint at those things in the first two paragraphs of our conclusion.

**C2.2.** Requested revisions 2. In a general sense I found the approach of emulating/constraining the OSCAR components and parameterizations using more complex model results (e.g. CMIP5/HTAP) or IPCC results, assumed that the reader was already well informed of this type of approach. , i.e. it was not always clear if emulation/ constraint was being applied and used across the ensemble of CMIP5 models or rather to a "best fit model vs observations" or an ensemble mean was being used to constrain OSCAR parameters for the historical period.
Request revision: A brief and basic description of the emulation approach might help more general readers

**R2.2.** We have added a paragraph in introduction that has 2 purposes: summarizing what OSCAR is, and explaining the main idea behind how it is made (i.e. the emulation/calibration and probabilistic approach). The text referring to the emulation approach in the paragraph is: "*OSCAR is also a parametric model which relatively large number of parameters are almost all calibrated on complex models. We call this approach meta-modelling: each module of OSCAR is designed to emulate the behavior of other more specialized models (e.g. global climate models, dynamical vegetation models, or chemistry-transport models). For most modules, we have access to several sets of parameters (one per complex model used to calibrate) and, rather than taking the average or arbitrarily choosing one, we adopt a probabilistic approach in which a given simulation with OSCAR is repeated many times with different sets of parameters picked at random.*".

**C2.3.** 3. Given the importance of marine carbon uptake and potential changes in the efficiency of this carbon sink/source in the future, I felt the level of detail describing the marine C cycle compared to the terrestrial C cycle was somewhat unbalanced. Equally, I was surprised that marine C cycle did not include any parameterization of the marine biological pump. While solubility processes may dominate historical and future marine C uptake, there is evidence that changes in the biological

pump are likely to play a non-negligible role in future marine carbon uptake. On this note, does the relative accuracy
of the ocean carbon uptake for the historical period (shown in figure 5) indicate that the biological pump was largely unimportant in this increased uptake? Or does it suggest the way the model has been constrained implicitly includes a biological component?
Requested revision: Some comments on the importance or not of the marine biological pump seems warranted, particularly if the primary application of the model is to investigate future uncertainties in coupled climate-carbon cycle processes.

**R2.3.** Unfortunately, the level of detail used to describe the marine C-cycle compared to the terrestrial one simply reflects the complexity of the modeling approach for these two components of the Earth system. Therefore it is true that the module for ocean carbon is 'simpler' than the one for terrestrial carbon, especially because it lacks a representation of key processes such as those related to the biological pump and also does not take into account regional specificities. This is both because of the history of the model development (see Changelog in appendix A in paper) and because the land-use module makes the terrestrial C-cycle rather complex. That said, we do not know of any other 'compact' model that would have a more complex ocean C-cycle representation, since it very quickly requires to explicitly model the multi-dimensional (fluid) dynamic in the ocean.

Because OSCAR is formulated as a difference to a preindustrial equilibrium (in which zero anthropogenic forcing is assumed), we can say that there is an implicit biological pump in the model, with the important limitation that this pump is implicitly assumed to remain unchanged throughout any simulation. Is this bad? Well, for the historical period, as the referee noticed, no change in the biological pump seems to be needed to model the ocean C sink. Back in the Third Assessment Report, IPCC stated that so far (i.e. 2001) the physical pump explains virtually 100% of the ocean sink, that is: "Despite the importance of biological processes for the ocean's natural cycle, current thinking maintains that the oceanic uptake of anthropogenic CO2 is primarily a physically and chemically controlled process surimposed on a biogically driven carbon cycle that is close to steady state". In other words, the anthropogenic perturbation of the ocean C-cycle can only be seen in the physical pump and not the biological one. Additionally, we did try to implement a simple biological pump model on top of the Joos et al. (1996) physical pump model, but those happen to be incompatible, and more work in this direction is clearly needed.

To sum this up: we chose to keep the biological pump constant, and we will see with the RCP projections (albeit in another paper) how the model performs in the future. We've made this clearer in the sentence that was initially dedicated to this in the 'ocean C-cycle' section: "*Note that this model of the ocean carbon-cycle implicitly assumes no change in the biological pump -- change that could be induced e.g. by changes in temperature, ocean circulation, nutrient availability or surface acidity (e.g. Ciais et al., 2013). This is one of the several processes not implemented in this version of OSCAR.*"; and added a specific comment in the results section: "*This relative good performance of the ocean carbon-cycle module, given that no change in the biological pump is simulated by OSCAR, suggests that the physical pump is enough to satisfactorily simulate the (recent) past carbon uptake by the ocean. Whether this would be enough to simulate future changes remain to be tested.*".

**C2.4.** 4. Is ocean acidification and its potential impact on marine carbon uptake included in the model? This was not obvious to me.
Requested revision: Please make it clear if yes and if not, as with point 2 above, what are the possible consequences for application to future projections.

**R2.4.** It is not entirely clear what the referee refers to, here. If the referee is mentioning the non-linearity of the carbonate chemistry (i.e. the 'effect' of acidification on carbonate chemistry and on the saturation of the carbon sink) then yes it is accounted for in the *FpCO2* non-linear function. If instead the referee refers to the potential effect on the biological pump then it is not accounted for. Note that the acidification effect on biological production is not included as well in most comprehensive models of the ocean carbon cycle. The modifications made in **R2.3** make the latter point clearer.

**C2.5.** 5. For certain model parameters: e.g. environmental controls on fire ignition, as an example, it is not clear how many more complex ESMs with interactive fire models were used to determine these parameters. The number of CMIP5 models with interactive fire models was pretty small.
Requested revision: Particularly where only a small number of models were available for constraining parameters this should be made clear.

**R2.5.** In the first version of the paper, we do provide how many models were used to calibrate the parameters in each case. For instance, in the previous manuscript, on page 11 line 4: "*Six models with wildfire emissions are available to calibrate on TRENDY, and four models are to calibrate on CMIP5.*" All these values are also summarized in the last table of the paper.

**C2.6.** 6. With respect to the terrestrial carbon cycle and atmospheric CH4, it seems that permafrost is not included in OSCAR? Is this correct. If so why was this decision taken and, like my comments in marine biology, there is evidence that permafrost melt may be an important future feedback in the Earth system.
Requested revision: Omission of this feedback seems like it needs a motivation and acknowledgement of potential projection limitations due to this decision.

**R2.6.** It is correct that permafrost is not included. And it is true that we failed to acknowledge that permafrost was missing. This is now corrected in the section dedicated to atmospheric CO2: "*In equation (33) this version of OSCAR notably ignores the permafrost carbon that may be emitted under a warming climate (e.g. Ciais et al., 2013).*".

We don't think, however, that this requires much explanation in the paper: many processes are missing, and some of these processes may have a greater impact than permafrost (e.g. N-limitation of the terrestrial C sink, or biophysical effect of land-use). The reasons why these processes are missing are various: unavailability of data exploitable to calibrate, unsuccessful attempts, decision to freeze the model at some arbitrary point. We do not believe these reasons should appear in the paper.
s

**C2.7.** Questions (not particularly requiring modifications in the paper unless the authors feel it will help)
7. With respect to surface temperature and precipitation changes (pages 33-34) the global climate sensitivity () plays an important role. This is derived from CMIP5 abrupt 4xCO2 and pre-industrial control simulations.  includes all "fast" climate feedbacks such as water vapour and cloud feedbacks. My question relates to the definition of cloud-aerosol effects in OSCAR, these seem to be potentially decoupled from (future) cloud changes, with the latter defined through . As future cloud aerosol effects will be mediated by any future changes in the distribution of fractional cloud and cloud microphysical properties, is there some risk that future cloud aerosol impacts may be inaccurate due to this decoupling?

**R2.7.** The referee is right: the aerosol-cloud effect is estimated independently from any actual change in cloud cover induced by the overall climate change. It is however impossible to couple both effects without developing an explicit energy/water model (with atmospheric transport!) in place of the response functions we use. We have added a sentence in the "cloud effect" section to precise this non-coupling: "*Note that the cloud effects are estimated independently from any change in cloud cover that is happening implicitly in the climate system module.*".

**C2.8.** 8. A similar question arises with respect to the calculation of precipitation and in particular, the regional weights for precipitation. How are cloud-aerosol changes on regional precipitation included, if at all, in OSCAR?

**R2.8.** Again the referee is right. But it is actually true for all forcings: they are not distinguished when it comes to their impact on regional precipitation (though they are for global precipitation). A sentence has also been added: "*As per surface temperature, the pattern scaling approach ignores the difference in effect the various climate forcers may have on regional precipitations.*". Note also that this was already acknowledged in the discussion/conclusion.

**C2.9.** 9. It is stated that the model is primarily used for annual mean or longer analysis. This is understandable given the time step and basic aims of the paper. My question is whether, in an approach analogous to statistical downscaling which brings an increased spatial dimension to coarse spatial resolution data can something similar be done in an effort to infer higher time frequency changes based on the annual mean timescale changes?

**R2.9.** This is a very interesting point. In some aspects, it relates to a point regarding inter-annual variability made by the other referee. The answer to the question as to whether it is possible to implement intra-annual variability in OSCAR is: yes, in theory. This would likely be done ex-post, using outputs from a simulation to 'downscale' the timeseries. One would need to calibrate the intra-annual cycle of the preindustrial period, e.g. on 'piControl' experiments from CMIP5, but also to calibrate how that cycle is affected by other changes in annual/decadal variables (e.g. how regional mean temperature affects the monthly or daily temperature profile). There is little doubt that the latter point would require a lot of work!

[revised manuscript text omitted]

$$\mathcal{F}_{\mathrm{resp}}^{i,b} = \exp \left[ \gamma_{\mathrm{resp},T}^{i,b}\, \Delta T_L^i \right] \left( 1 + \gamma_{\mathrm{resp},P}^{i,b}\, \Delta P_L^i \right); \tag{15}$$

and if Gaussian, it is a function with three parameters, two of which being the sensitivity to temperature split between a first-order term ($\gamma_{\mathrm{resp},T_1} > 0$) and a second-order term ($\gamma_{\mathrm{resp},T_2} < 0$), and the third being the sensitivity to precipitations ($\tilde{\gamma}_{\mathrm{resp},P}$):

$$\mathcal{F}_{\mathrm{resp}}^{i,b} = \exp\left[\gamma_{\mathrm{resp},T_1}^{i,b}\,\Delta T_L^i + \gamma_{\mathrm{resp},T_2}^{i,b}\,\Delta T_L^{i\,2}\right]\left(1 + \tilde{\gamma}_{\mathrm{resp},P}^{i,b}\,\Delta P_L^i\right). \tag{16}$$

5  Part of the litter carbon is metabolized into soil organic carbon. This flux ($f_{\mathrm{met}}$) is taken proportional to the heterotrophic respiration of the litter carbon pool (by a factor $\kappa_{\mathrm{met}}$):

$$\Delta f_{\mathrm{met}}^{i,b} = \kappa_{\mathrm{met}}\,\Delta \mathrm{rh}_{\mathrm{litt}}^{i,b}. \tag{17}$$

Heterotrophic respiration ($\mathrm{rh}_{\mathrm{soil}}$) also occurs in the soil carbon pool ($c_{\mathrm{soil}}$). It is a function of its preindustrial value ($\rho_{\mathrm{soil}}$) and of the same function $\mathcal{F}_{\mathrm{resp}}$ as for the litter:

10  $$\Delta \mathrm{rh}_{\mathrm{soil}}^{i,b} = \rho_{\mathrm{soil}}^{i,b}\,c_{\mathrm{soil},0}^{i,b}\left(\left(1 + \frac{\Delta c_{\mathrm{soil}}^{i,b}}{c_{\mathrm{soil},0}^{i,b}}\right)\mathcal{F}_{\mathrm{resp}}^{i,b}\left[\Delta T_L^i, P_L^i\right] - 1\right). \tag{18}$$

And finally, the terrestrial carbon cycle of a given doublet $(i,b)$ follows:

$$\frac{\mathrm{d}}{\mathrm{d}t}\Delta c_{\mathrm{veg}}^{i,b} = \Delta \mathrm{npp}^{i,b} - \Delta e_{\mathrm{fire}}^{i,b} - \Delta f_{\mathrm{mort}}^{i,b}; \tag{19}$$

$$\frac{\mathrm{d}}{\mathrm{d}t}\Delta c_{\mathrm{litt}}^{i,b} = \Delta f_{\mathrm{mort}}^{i,b} - \Delta \mathrm{rh}_{\mathrm{litt}}^{i,b} - \Delta f_{\mathrm{met}}^{i,b}; \tag{20}$$

15  $$\frac{\mathrm{d}}{\mathrm{d}t}\Delta c_{\mathrm{soil}}^{i,b} = \Delta f_{\mathrm{met}}^{i,b} - \Delta \mathrm{rh}_{\mathrm{soil}}^{i,b}. \tag{21}$$

The equation system described above by equations (19), (20) and (21) implies that our preindustrial equilibrium is:

$$\begin{cases} \mathrm{npp}_0^{i,b} = f_{\mathrm{mort},0}^{i,b} + e_{\mathrm{fire},0}^{i,b} \\ f_{\mathrm{mort},0}^{i,b} = \mathrm{rh}_{\mathrm{litt},0}^{i,b} + f_{\mathrm{met},0}^{i,b} \\ f_{\mathrm{met},0}^{i,b} = \mathrm{rh}_{\mathrm{soil},0}^{i,b} \end{cases}; \tag{22}$$

which, in terms of flux parameters and preindustrial carbon stocks, is equivalent to:

$$\begin{cases} \eta^{i,b} = \left(\mu^{i,b} + \iota^{i,b}\right)c_{\mathrm{veg},0}^{i,b} \\ \mu^{i,b}\,c_{\mathrm{veg},0}^{i,b} = (1 + \kappa_{\mathrm{met}})\,\rho_{\mathrm{litt}}^{i,b}\,c_{\mathrm{litt},0}^{i,b} \\ \kappa_{\mathrm{met}}\,\rho_{\mathrm{litt}}^{i,b}\,c_{\mathrm{litt},0}^{i,b} = \rho_{\mathrm{soil}}^{i,b}\,c_{\mathrm{soil},0}^{i,b} \end{cases}. \tag{23}$$

20  Note that to obtain the global preindustrial terrestrial carbon-cycle equilibrium one needs to multiply the above equilibrium by the preindustrial biome area extents ($A_0$), for instance: $\mathrm{NPP}_0^{\mathrm{global}} = \sum_{i,b}\mathrm{NPP}_0^{i,b} = \sum_{i,b}\mathrm{npp}_0^{i,b}\,A_0^{i,b}$. Note also that the extensive perturbation, described in the next section, alters the biome area extents so that we actually have $A^{i,b} = A_0^{i,b} + \Delta A^{i,b}$.

The parameters for the preindustrial fluxes (i.e. $\eta$, $\mu$, $\rho_{\mathrm{litt}}$ and $\rho_{\mathrm{soil}}$) can be calibrated on nine TRENDY v2 dynamic global vegetation models (Le Quéré et al., 2014; Sitch et al., 2015). To do so, we use the first thirty years of the so-called "S2"

simulation, in which changing climate and $CO_2$ are prescribed to the models but no land-use change happens. We assume the average fluxes and pools over that period are at steady-state, so that we can deduce the parameters from equation (23), taking $\kappa_{\mathrm{met}} = 0.3/0.7$ (Foley, 1995). For the few models that do not report separately the litter pool, we assume the total reported soil carbon pool is made at 5% of litter carbon and 95% of soil carbon. Also, to account for the harvest of croplands,

we alter the parameters of this biome following some arbitrary rules: NPP is reduced by 80%, thus we assume this fraction of the crops' productivity is harvested and oxidized within a year; and the mortality rate is set to $1 \mathrm{\ yr^{-1}}$, which corresponds to a yearly harvest. Also, because the assumed preindustrial equilibrium based on TRENDY is 1901–1930 and not 1750, we scale down the NPP parameter $\eta$ by a factor equal to the ratio of our preindustrial atmospheric $CO_2$ over the one for the TRENDY preindustrial period i.e. by a factor of about 0.92.

The parameters for the transient response of NPP and hetetrotrophic respiration (i.e. $\beta_{\mathrm{npp}}$, $\tilde{\beta}_{\mathrm{npp}}$, $CO2_{\mathrm{cp}}$, $\gamma_{\mathrm{npp},T}$, $\gamma_{\mathrm{npp},P}$, $\gamma_{\mathrm{resp},T}$, $\gamma_{\mathrm{resp},T_1}$, $\gamma_{\mathrm{resp},T_2}$, $\tilde{\gamma}_{\mathrm{resp},P}$) can be calibrated on seven CMIP5 Earth system models (see e.g. Arora et al., 2013). To do so, we use the outputs from three CMIP5 simulations: "1pctCO2", "esmFixClim", "esmFdbk1" which correspond to simulations with an increase of atmospheric $CO_2$ of +1% $\mathrm{yr^{-1}}$ in the case of a fully coupled configuration, a fixed climate, or a fixed carbon-cycle, respectively. Depending on the functional form chosen, the fit for NPP is done on the basis of equations (8)+(9) or (8)+(10). That of the heterotrophic respiration rate is done on the basis of equation (15) or (16). The calibration is done in two steps. A first fit is made with decadal averages of the relevant variables and for which the parameter related to local precipitations is set to zero. A second fit is then made with annual values to find the remaining parameter. This approach is used to avoid over-fitting. The fit is made over the three simulations at the same time, using the 'piControl' values to define the preindustrial equilibrium. In the case of the respiration rate, we also add a new term to equation (15) or (16) to calibrate the parameters. We multiply $\mathcal{F}_{\mathrm{resp}}$ by the term $(1 + \beta_{\mathrm{prim}}^{i,b} \Delta F_{\mathrm{input}}^{i,b}/F_{\mathrm{input},0}^{i,b})$, where $\beta_{\mathrm{prim}}$ is a new sensitivity and $F_{\mathrm{
[revised manuscript text omitted]

$$+ \sum_b \alpha_{\mathrm{bb}}^{\mathrm{X},i,b} \left( e_{\mathrm{fire},0}^{i,b} \, \Delta A^{i,b} + \Delta e_{\mathrm{fire}}^{i,b} \, A_0^{i,b} + \Delta e_{\mathrm{fire}}^{i,b} \, \Delta A^{i,b} \right)$$
$$+ \sum_{b,a} \alpha_{\mathrm{bb}}^{\mathrm{X},i,b} \left( \iota_0^{i,b} \, \mathrm{f}_{\mathrm{igni}}^{i,b} \, \Delta C_{\mathrm{veg},\mathrm{luc}}^{i,b,a} - \frac{\mathrm{d}}{\mathrm{d}t} \Delta C_{\mathrm{hwp},\mathrm{luc}}^{w=1,i,b,a} \right). \tag{38}$$

The $\alpha_{\mathrm{bb}}$ parameters come from the GFED v3.1 database (van der Werf et al., 2010). The biomass burning emissions of all
20   species are averaged over the whole available time-period, and to each vegetation type – or sector – of GFED is associated a biome of OSCAR: 'def' and 'for' are forests, 'woo' is shrublands, 'sav' is grasslands, 'agr' is croplands; 'pea', i.e. peatlands, are left alone. As in section 2.3.2, pastures are assumed to be 60% grasslands and 40% bare soil. The parameters are then obtained by simply taking the ratio of the emissions of a given species over those of CO$_2$.

**2.4.2  Lagged concentrations**

25   In the next sections, we need an estimate of the stratospheric concentration change of some species. For relatively long-lived species, we assume the stratospheric concentration change of this species can be approximated by its change in atmospheric concentration (X), albeit with a time-lag ($\tau_{\mathrm{lag}}$). This change in "lagged" concentration (X$_{\mathrm{lag}}$) is formulated as:

$$\tau_{\mathrm{lag}} \frac{\mathrm{d}}{\mathrm{d}t} \mathrm{X}_{\mathrm{lag}} = \Delta \mathrm{X} - \Delta \mathrm{X}_{\mathrm{
[revised manuscript text omitted]
^i_{\text{wet}} = e^i_{\text{wet},0}\,\Delta A^i_{\text{wet}} + \Delta e^i_{\text{wet}}\,A^i_{\text{wet},0} + \Delta e^i_{\text{wet}}\,\Delta A^i_{\text{wet}}. \tag{46}$$

We calibrate two sets of parameters for wetlands. First, the preindustrial equilibrium of the wetlands can be calibrated on seven WETCHIMP models (Melton et al., 2013). We deduce the $\pi_{\text{wet}}$ parameters by combining the wetlands map from the "exp 1" simulation, that is the  control experiment of the WETCHIMP exercise, and the land-cover map used in section 2.3.2 for natural vegetation. The preindustrial areal emissions $e_{\text{wet},0}$ are also taken from this "exp 1" simulation, but they are scaled down by a factor equal to the ratio of our preindustrial atmospheric $CO_2$ over the one used in WETCHIMP i.e. by a factor of about 0.92, as we did with NPP in section 2.3.2. Second, the parameters for the transient response of wetlands extent (i.e. $\gamma_{\text{wet},C}$, $\gamma_{\text{wet},T}$, $\gamma_{\text{wet},C}$) can be calibrated on six WETCHIMP models (reminder: see appendix B for a list of those models). To do so, we use "exp4", "exp5" and "exp6": factorial simulations that separate the effect of temperature, precipitations and atmospheric $CO_2$, respectively. For the same reasons as with wildfires, we also keep an option to turn off the preindustrial wetlands flux and/or its transient response.

**2.5.3 Atmospheric $CH_4$ and RF**

On the basis of the previous sections, the incremental change in atmospheric $CH_4$ follows the mass-balance equation:

$$\alpha^{\text{CH4}}_{\text{atm}}\,\frac{d}{dt}\Delta\text{CH4} = E_{\text{CH4}} + \Delta E^{\text{CH4}}_{\text{bb}} + \sum_i \Delta E^i_{\text{wet}} + \Delta F^{\text{CH4}}_{\downarrow}. \tag{47}$$

This equation implicitly assumes that all the natural sources of methane but natural wetlands remain unchanged since the preindustrial. Here, we also note that the anthropogenic emissions $E_{\text{CH4}}$ do include emissions from rice paddies – i.e. from anthropogenic wetlands.

The radiative forcing induced by the increase in atmospheric $CH_4$ follows a square-root formula to which an *ad hoc* function ($\mathcal{F}_{\text{over}}$) is added to account for the overlap between the absorption bands of methane and nitrous oxide (N2O), following Myhre et al. (1998). It gives:

$$\Delta\text{RF}^{\text{CH4}} =$$

$$+\,\alpha^{\text{CH4}}_{\text{rf}}\,\sqrt{\text{CH4}_0}\,\left(\sqrt{1 + \frac{\Delta\text{CH4}}{\text{CH4}_0}} - 1\right)$$

$$-\,\left(\mathcal{F}_{\text{over}}\left[\Delta\text{CH4}, \Delta\text{N2O}\right] - \mathcal{F}_{\text{over}}\left[\Delta\text{CH4}=0, \Delta\text{N2O}\right]\right); \tag{48}$$

where $\alpha^{\text{CH4}}_{\text{rf}} = 0.036$ W m$^{-2}$ ppb$^{-0.5}$ and the analytical expression of $\mathcal{F}_{\text{over}}$ are given by Myhre et al. (2013a, table 8.SM.1). In addition to the RF induced by methane itself, we have to account for the RF induced by the increase in stratospheric water vapor caused by the oxidation of methane. To do so, as others (e.g. Meinshausen et al., 2011), we assume it is equal to 15% of the direct methane RF, but calculated with its lagged concentration:

$$\Delta\text{RF}^{\text{H2Os}} = \alpha^{\text{H2Os}}_{\text{rf}}\,\sqrt{\text{CH4}_0}\,\left(\sqrt{1 + \frac{\Delta\text{CH4}_{\text{lag}}}{\text{CH4}_0}} - 1\right); \tag{49}$$

where $\alpha_{\mathrm{rf}}^{\mathrm{H2Os}} = 0.15 \times \alpha_{\mathrm{rf}}^{\mathrm{CH4}} = 0.0054$ W m$^{-2}$ ppb$^{-0.5}$. For the preindustrial atmospheric concentration, we take $\mathrm{CH4_0} = 722$ ppb (IPCC, 2013, table AII.1.1a).

**2.6 Nitrous oxide**

In OSCAR, the stratospheric sink of nitrous oxide is included, and a particular attention is paid to how it varies with anthropogenic and natural external factors. However, no other natural processes are endogenous to the model, meaning that no change in natural sources or sinks of nitrous oxide (e.g. ocean, natural soils, biological fixation) is assumed.

**2.6.1 Stratospheric sink**

The oxidation of nitrous oxide follows the same modelling approach as that of methane, with only one sink in the stratosphere that has a varying lifetime. The law used to make the stratospheric lifetime vary, however, is recent and different from the previous version of the model.

The flux of oxidized $N_2O$ ($F_{\downarrow}^{\mathrm{N2O}}$) is driven by the preindustrial lifetime of nitrous oxide with regard to stratospheric oxidation ($\tau_{\mathrm{h}\nu}^{\mathrm{N2O}}$). The transient change in this stratospheric lifetime is a function ($\mathcal{F}_{\mathrm{h}\nu}$) of: the lagged $N_2O$ concentration ($\mathrm{N2O_{lag}}$); the equivalent effective stratospheric chlorine (EESC; see section 2.8.2); and global surface temperature change ($T_G$). The dependency on $N_2O$ and the EESC is meant to model the impact of a change in stratospheric ozone that changes the actinic flux, which in turn changes the stratospheric sink (e.g. Prather, 1998). We have:

$$
\alpha_{\mathrm{atm}}^{\mathrm{N2O}\,-1} \Delta F_{\downarrow}^{\mathrm{N2O}} =
$$
$$
-\tfrac{\mathrm{N2O_0}}{\tau_{\mathrm{h}\nu}^{\mathrm{N2O}}} \left( \left(1 + \tfrac{\Delta\mathrm{N2O_{lag}}}{\mathrm{N2O_0}}\right) \mathcal{F}_{\mathrm{h}\nu}\left[\Delta\mathrm{N2O_{lag}}, \Delta\mathrm{EESC}, \Delta T_G\right] - 1 \right). \tag{50}
$$

The formulation of $\mathcal{F}_{\mathrm{h}\nu}$ is inspired by that used for methane and the study by Prather et al. (2015). It has three chemical sensitivities ($\chi_{\mathrm{N2O}}^{\mathrm{h}\nu}$, $\chi_{\mathrm{EESC}}^{\mathrm{h}\nu}$ and $\chi_{\mathrm{age}}^{\mathrm{h}\nu}$). This last parameter represent the sensitivity of the sink to a change in stratospheric age of air. This age-of-air change is itself driven by a changing Brewer-Dobson circulation which is induced by a changing climate (e.g. Butchart, 2014). In the following, we consider that the inverse of the relative change in age of air is a linear function of the absolute change in global surface temperature (parameterized by $\gamma_{\mathrm{age}}$; see also figure S23). This leads to the following formula:

$$
\ln\left[\mathcal{F}_{\mathrm{h}\nu}\right] =
$$
$$
+ \chi_{\mathrm{N2O}}^{\mathrm{h}\nu} \ln\left[1 + \frac{\Delta\mathrm{N2O_{lag}}}{\mathrm{N2O_0}}\right]
$$
$$
+ \chi_{\mathrm{EESC}}^{\mathrm{h}\nu} \ln\left[1 + \frac{\Delta\mathrm{EESC}}{\mathrm{EESC_0}}\right]
$$
$$
+ \chi_{\mathrm{age}}^{\mathrm{h}\nu} \ln\left[\frac{1}{1 + \gamma_{\mathrm{age}} \Delta T_G}\right]. \tag{51}
$$

The preindustrial stratospheric lifetime $\tau_{\mathrm{h}\nu}^{\mathrm{N2O}}$ is taken as 123 years (Prather et al., 2015). As we do with methane, we introduce variation in the $N_2O$ lifetime by having the option to rescale the default value by a factor equal to the lifetime

simulated by any of the eight models of Prather et al. (2015, table 2) over the multi-model mean estimate. The first two chemical sensitivities of the stratospheric sink (i.e. $\chi_{\text{N2O}}^{\text{h}\nu}$ and $\chi_{\text{EESC}}^{\text{h}\nu}$) are taken as one of the four sets of values from the study by Prather et al. (2015). Three sets of value are given in their table 3, and the fourth is the  recommendation in their text. Also, to translate their table 3 into our parameters, we assume that the preindustrial EESC in the models were 420

5    ppt – from IPCC (2013, table AII.1.1b) and Newman et al. (2007, table 1). Alternatively, for backward compatibility, these parameters can also follow (Prather et al., 2012, table A1), in which case the sensitivity to EESC is zero.

Regarding the chemical sensitivity to the age of air, we assume it is not zero only when the other sensitivities are deduced from the "G2d" model, therefore following the results by Prather et al. (2015, table 3) and their discussion pointing out the experimental aspect of such a parameterization. Nevertheless, in this specific case we need further information about the "G2d"

10    model which we take from Fleming et al. (2011, figure 12) where one can see that the age of air at an altitude of 25 km changed from about 4.5 to 4.0 between the preindustrial and present-day periods. This is enough to deduce the $\chi_{\text{age}}^{\text{h}\nu}$ parameter. And then, the $\gamma_{\text{age}}$ parameter can be calibrated on seven CCMVal2 chemistry-transport models (Morgenstern et al., 2010). To do so, we use outputs from the "REF-B2" experiment which is a fully transient simulation over 1961–2099: we use the "mean_age" output at a pressure-level of 25 hPa ($\sim$25 km) and the temperature at the surface level. We then fit the parameter following

15    our inversed linear relationship, defining the preindustrial conditions as the averaged first ten years of the simulations. The CCMVal2 fits are shown in figure S23.

**2.6.2   Atmospheric N$_2$O and RF**

The incremental change in atmospheric N$_2$O follows:

$$\alpha_{\text{atm}}^{\text{N2O}} \frac{\text{d}}{\text{d}t}\Delta\text{N2O} = E_{\text{N2O}} + \Delta E_{\text{bb}}^{\text{N2O}} + \Delta F_{\downarrow}^{\text{N2O}}; \tag{52}$$

20    noting again that this implicitly assumes natural emissions remain unchanged since the preindustrial.

Similarly to methane, the radiative forcing induced by the increase in atmospheric N$_2$O follows a square-root formula to which the *ad hoc* overlap function is added:

$$\Delta\text{RF}^{\text{N2O}} =$$

$$+ \alpha_{\text{rf}}^{\text{N2O}} \sqrt{\text{N2O}_0} \left( \sqrt{1 + \frac{\Delta\text{N2O}}{\text{N2O}_0}} - 1 \right)$$

25    $$- \left( \mathcal{F}_{\text{over}}\left[\Delta\text{CH4}, \Delta\text{N2O}\right] - \mathcal{F}_{\text{over}}\left[\Delta\text{CH4}, \Delta\text{N2O} = 0\right] \right); \tag{53}$$

where $\alpha_{\text{rf}}^{\text{N2O}}$ = 0.12 W m$^{-2}$ ppb$^{-0.5}$ and $\mathcal{F}_{\text{over}}$ are given by Myhre et al. (2013b, table 8.SM.1). For the preindustrial atmospheric concentration, we take N2O$_0$ = 270 ppb (IPCC, 2013, table AII.1.1a).

**2.7   Halogenated compounds**

OSCAR accounts for many halogenated species. These are grouped into three categories: eleven hydrofluorocarbons (HFC-23,

30    HFC-32, HFC-125, HFC-134a, HFC-143a, HFC-152a, HFC-227ea, HFC-236fa, HFC-245fa, HFC-365mfc, HFC-43-10mee)

noted together {HFC}; eight perfluorocarbons ($CF_4$, $C_2F_6$, $C_3F_8$, c-$C_4F_8$, $C_4F_{10}$, $C_5F_{12}$, $C_6F_{14}$, $C_7F_{16}$) to which we add $SF_6$ and $NF_3$, and noted together {PFC}; and sixteen ozone depleting substances (CFC-11, CFC-12, CFC-113, CFC-114, CFC-115, $CCl_4$, $CH_3CCl_3$, HCFC-22, HCFC-141b, HCFC-142b, Halon-1211, Halon-1202, Halon-1301, Halon-2402, $CH_3Br$, $CH_3Cl$) noted together {ODS}. These are the same as in previous version 2.1.

**2.7.1 Atmospheric sinks**

Conceptually, the modelling approach of the halogenated compounds' sinks is similar to that used for methane. Each of these species X is affected by three sinks, each sink with its specific preindustrial lifetime: a tropospheric oxidation by the hydroxyl radical ($\tau_{OH}^X$), a stratospheric oxidation ($\tau_{h\nu}^X$), and another sink which encloses all other processes such as oxidation in dry soils or in the oceanic boundary layer ($\tau_{othr}^X$). Note that a given oxidation process may not actually affect a given species; in this case the associated lifetime is set to a value of infinity ($\infty$) – or equivalently its loss frequency ($\nu^X = 1/\tau^X$) is set to zero. Mathematically, similarly to equation (40), we have for any species X being a HFC, PFC or ODS:

$$
\alpha_{atm}^{X}{}^{-1} \Delta F_{\downarrow}^X =
$$
$$
-\frac{1}{\tau_{OH}^X} \left( (\Delta X + X_0)\, \mathcal{F}_{OH}\left[\Delta CH4, \Delta O3s, \Delta T_G, \mathcal{F}_{prec}\right] - X_0 \right)
$$
$$
-\frac{1}{\tau_{h\nu}^X} \left( (\Delta X_{lag} + X_0)\, \mathcal{F}_{h\nu}\left[\Delta N2O_{lag}, \Delta EESC, \Delta T_G\right] - X_0 \right)
$$
$$
-\frac{1}{\tau_{othr}^X} \Delta X; \tag{54}
$$

[revised manuscript text omitted]

$$+\,\tau_{\mathrm{BC,bb}}\sum_i \Delta E_{\mathrm{bb}}^{\mathrm{BC},i}$$
$$+\,\Gamma_{\mathrm{BC}}\,\Delta T_G. \tag{65}$$

In the case of nitrate aerosols, inspired by Shindell et al. (2009), we assume their formation is driven by nitrogen oxides and ammonia emissions, and therefore we uncouple the nitrate and sulphate chemistries while they are coupled in reality (Boucher et al., 2013). Hence, the change in burden of nitrate aerosols (NO3) is parameterized by the apparent lifetime of nitrogen oxides ($\tau_{\mathrm{NOx}}$), the apparent lifetime of ammonia ($\tau_{\mathrm{NH3}}$), and their sensitivity to global surface temperature ($\Gamma_{\mathrm{NO3}}$). So we have:

$$\Delta\mathrm{NO3} =$$
$$+\,\tau_{\mathrm{NOx}}\left(E_{\mathrm{NOx}}+\sum_i \Delta E_{\mathrm{bb}}^{\mathrm{NOx},i}\right)$$
$$+\,\tau_{\mathrm{NH3}}\left(E_{\mathrm{NH3}}+\sum_i \Delta E_{\mathrm{bb}}^{\mathrm{NH3},i}\right)$$
$$+\,\Gamma_{\mathrm{NO3}}\,\Delta T_G. \tag{66}$$

And finally, the change in burden of secondary organic aerosols (SOA) is parameterized by the apparent lifetime of anthropogenic NMVOCs ($\tau_{\mathrm{VOC}}$), the apparent lifetime of biogenic NMVOCs ($\tau_{\mathrm{BVOC}}$), and their sensitivity to global surface temperature ($\Gamma_{\mathrm{SOA}}$). Here, the dependency of SOA on other factors such as atmospheric $\mathrm{NO_x}$ or POA (Boucher et al., 2013) is neglected. So we have:

$$\Delta\mathrm{SOA} =$$
$$+\,\tau_{\mathrm{VOC}}\left(E_{\mathrm{VOC}}+\sum_i \Delta E_{\mathrm{bb}}^{\mathrm{VOC},i}\right)$$
$$+\,\tau_{\mathrm{BVOC}}\,\Delta E_{\mathrm{BVOC}}$$
$$+\,\Gamma_{\mathrm{SOA}}\,\Delta T_G. \tag{67}$$

Finally, here it must be noted that, despite being used for the calibration (see below) and being shown in equations (63) and (67), DMS and BVOC emissions are constant in this version of OSCAR. In other words, in any simulation with OSCAR v2.2 we have $\Delta E_{DMS} = 0$ and $\Delta E_{BVOC} = 0$. Also in this version, we do not model any change in natural aerosols, i.e. in mineral dust and sea salt.

[revised manuscript text omitted]

To estimate global warming, however, we have to account for the so-called "efficacy" of these forcings, i.e. we have to introduce new parameters ($\kappa_{warm}^{X}$) that measure the relative efficiency at warming the Earth of a given RF when compared to the RF of $CO_2$ (see e.g. Hansen et al., 2005; Forster et al., 2007). In OSCAR, we assume all efficacies are equal to 1 – although accounting for the semi-direct effect of BC could be defined as using an efficacy – except for the two surface albedo forcings and for volcanic aerosols. Therefore, the RF used to calculate warming ($RF_{warm}$) is:

$$\Delta RF_{warm} =$$
$$+ \Delta RF^{WMGHG} + \Delta RF^{NTCF} + RF_{con} + RF_{solar}$$
$$+ \kappa_{warm}^{BCsnow} \Delta RF^{BCsnow} + \kappa_{warm}^{LCC} \Delta RF^{LCC} + \kappa_{warm}^{volc} RF_{volc}. \tag{76}$$

Here, $\kappa_{warm}^{BCsnow}$ can take three values: a median value of 3.0, a low value of 2.0, and a high value of 4.0, all from Boucher et al. (2013, section 7.5.2.3); $\kappa_{warm}^{LCC}$ can take one of the four values given by Bright et al. (2015, table 7); and $\kappa_{warm}^{volc}$ is set to an arbitrary value of 0.6 based on Gregory et al. (2016). However, regarding volcanic aerosols, we note that since the forcing is normalized to zero over the historical period in section 2.2.3, its efficacy only influences the variability of our results and not the trend.

Now, to estimate global precipitations change, we also need to estimate how much of this top-of-the-atmosphere RF is actually  occurring within the atmosphere – thus creating a local energy imbalance – by opposition to the RF  occurring at the Earth's surface. To do so, we introduce new parameters that quantify this atmospheric fraction for several groups of forcers: carbon dioxide alone ($\pi_{atm}^{CO2}$); all the other long-lived greenhouse gases, i.e. methane, nitrous oxide and the halogenated compounds ($\pi_{atm}^{noCO2}$); tropospheric ozone alone ($\pi_{atm}^{O3t}$); stratospheric greenhouse gases, i.e. stratospheric water vapor and ozone ($\pi_{atm}^{strat}$); scattering aerosols, i.e. sulphate, primary organic, nitrate, secondary organic and volcanic aerosols ($\pi_{atm}^{scatter}$); absorbing aerosols, i.e. black carbon ($\pi_{atm}^{absorb}$); cloud-related forcings ($\pi_{atm}^{cloud}$); forcings from surface albedo change

[revised manuscript text omitted]